# Changes in the simulation of atmospheric instability over the Iberian Peninsula due to the use of 3DVAR data assimilation

Santos J. González-Rojí[1,2], Sheila Carreno-Madinabeitia[3,4], Jon Sáenz[5,6], and Gabriel Ibarra-Berastegi[7,6]

[1]Oeschger Centre for Climate Change Research, University of Bern, Bern, Switzerland.
[2]Climate and Environmental Physics, University of Bern, Bern, Switzerland.
[3]Department of Mathematics, University of the Basque Country (UPV/EHU), Vitoria-Gasteiz, Spain.
[4]TECNALIA, Basque Research and Technology Alliance (BRTA), Parque Tecnológico de Álava, Vitoria-Gasteiz, Spain.
[5]Department of Physics, University of the Basque Country (UPV/EHU), Leioa, Spain.
[6]Plentziako Itsas Estazioa (BEGIK), University of the Basque Country (UPV/EHU), Plentzia, Spain.
[7]Department of Energy Engineering, University of the Basque Country (UPV/EHU), Bilbao, Spain.

**Correspondence:** Santos J. González-Rojí (santos.gonzalez@climate.unibe.ch)

**Abstract.** The ability of two downscaling experiments to correctly simulate thermodynamic conditions over the Iberian Peninsula (IP) is compared in this paper. To do so, three parameters used to evaluate the unstable conditions in the atmosphere are evaluated: TT index, CAPE and CIN. The WRF model is used for the simulations. The N experiment is driven by ERA-Interim's initial and boundary conditions; The D experiment has the same configuration as N, but the 3DVAR data assimilation step is additionally run at 00, 06, 12 and 18 UTC. Eight radiosondes are available over the IP, and the vertical temperature and moisture profiles from the radiosondes provided by the University of Wyoming and the Integrated Global Radiosonde Archive (IGRA) were used to calculate three parameters commonly used to represent atmospheric instability by our own methodology using the R package *aiRthermo*. According to the validation, the correlation, Standard Deviation (SD) and Root Mean Squared Error (RMSE) obtained by the experiment D for all the variables at most of the stations are better than those for N. The different methods produce small discrepancies between the values for TT, but these are larger for CAPE and CIN due to the dependency of these quantities on the initial conditions assumed for the calculation of a lifted air parcel. Similar results arise from the seasonal analysis concerning both WRF experiments: N tends to overestimate or underestimate (depending on the parameter) the variability of the reference values of the parameters, but D is able to capture it in most of the seasons. In general, D is able to produce more reliable results due to the more realistic values of dew point temperature and virtual temperature profiles over the IP. The heterogeneity of the studied variables is highlighted in the mean maps over the IP. According to those for D, the unstable air masses are found along the entire Atlantic coast during winter, but in summer they are located particularly over the Mediterranean coast. The convective inhibition is more extended towards inland at 00 UTC in those areas. However, high values are observed near the southeastern corner of the IP (near Murcia) also at 12 UTC. Finally, no linear relationship between TT, CAPE or CIN was found, and consequently, CAPE and CIN should be preferred for the study of the instability of the atmosphere as more atmospheric layers are employed during their calculation than for TT index.

# 1 Introduction

Precipitation is one of the most important variables involved in the water balance, and its variability determines the water resources of the planet. Following the definitions of regional models, precipitation can be separated in two categories: large-scale and convective precipitation. In general, convective precipitation is frequently associated with precipitation extreme events due to high intensity over a short duration. However, the simulation of these events is a well-known problem in the modelling community (Sillmann et al., 2013) due to restrictions in the resolution, poor representation of complex topography, insufficient assimilated observations, forecast errors or deficiencies in the microphysics schemes in the numerical models. In order to avoid these problems, as previously done in the literature (Viceto et al., 2017), this paper focuses on the evaluation of the atmospheric conditions favourable for the development of convective precipitation rather than the validation of the simulation of extreme events.

The evaluation of the atmospheric conditions is typically based on the calculation of some instability indices such as Lifted Index (LI) (Galway, 1956), K-Index (George, 1960), Total Totals index (TT) (Miller, 1975) or Showalter Index (S) (Showalter, 1953). These conditions can be also evaluated by Convective Available Potential Energy (CAPE) (Moncrieff, 1981) or Convective INhibition (CIN) (Moncrieff, 1981). All of these variables are commonly used in the literature for this kind of studies (e.g. Ye et al., 1998; DeRubertis, 2006; Viceto et al., 2017). CAPE and CIN are based on the adiabatic lifting of a parcel, while most of the others are based on differences in the values of several variables at different pressure levels. The deep convection is caused by three ingredients: high levels of moisture in the planetary boundary layer (PBL), potential instability and forced lifting (Johns and Doswell, 1992; McNulty, 1995; Holley et al., 2014; Gascón et al., 2015). CAPE and CIN provide information about the first two ingredients (Holley et al., 2014), and both can give details about the genesis and intensity of the atmospheric convection (Riemann-Campe et al., 2009). However, previous studies (Angus et al., 1988; López et al., 2001) suggest that CAPE should not be used alone, but should be combined with other indices. The final ingredient, which is the forced lifting, is usually caused by the orography (Doswell et al., 1998; Siedlecki, 2009), the convergence of horizontal moisture fluxes (McNulty, 1995) or the breezes in coastal regions (van Delden, 2001). Thus, the high spatial and temporal resolution is important for this kind of studies focusing on the atmospheric convection, and that is why regional simulations are needed (Siedlecki, 2009).

The probability of occurrence of convective precipitation is not the same along the day, and previous studies support that the maximum convection takes place in the afternoon and evening (Siedlecki, 2009; Virts et al., 2013; Piper and Kunz, 2017; Enno et al., 2020). According to van Delden (2001), the preferred time in most of Western Europe is between 18 and 24 UTC, with the exception of the island of Corsica where the sea breeze causes convection usually between 6-12 UTC. In open sea areas, the lightning activity peaks in the morning (Enno et al., 2020), associated to thunderstorms caused by land breezes at night (Virts et al., 2013). A regional study focusing over the UK (Holley et al., 2014) suggests that the reduction overnight of CAPE is over 500 J/kg.

On the global scale, CAPE follows the spatial pattern of surface specific humidity and air temperature, which means that it increases from pole to Equator (Riemann-Campe et al., 2009). The minimums are obtained in arid regions and over areas

with cold water up-welling. Focusing on Europe, convective storms develop for lower values than the U.S. (Graf et al., 2011), and several studies tried to determine the most active regions. Amongst them, Romero et al. (2007) found that the region with highest instability is located along a zonal belt over the south-central Europe, particularly over the west Mediterranean sea and the surrounding areas. This agrees with Brooks et al. (2003), who found that the favourable environment for thunderstorms is developed in southern Europe, and that the highest number of days in such a regime are located over the Iberian Peninsula (hereafter, IP), south of the Alps, and Northern Balkans. However, van Delden (2001) found that the southwestern France and the Basque Country seem to be a preferred region for the formation of severe storms that drift towards the northeast. More recent studies based on lightning data (Enno et al., 2020) and regional climate models using higher resolution (Mohr et al., 2015; Rädler et al., 2018) highlighted the same areas with favourable environments for thunderstorms in Europe, which are located in particular over northern Italy (Po Valley), east of the Adriatic Sea (Albania, Bosnia and Serbia) and in the northeastern IP and Southern France (near the Gulf of Lyon).

Over the IP, the seasonality of precipitation is determined by different sources of moisture due to seasonal variations of the global atmospheric circulation and contrasting climatic regions (influenced by the strong topography). Northern and western IP are mainly affected by stratiform precipitation during winter, while eastern and southern IP receive great amounts of precipitation during autumn due to convective activity (Rodríguez-Puebla et al., 1998; Esteban-Parra et al., 1998; Romero et al., 1999; Iturrioz et al., 2007). Maximum precipitation amounts over central IP are measured in early spring (Tullot, 2000).

Previous studies over the IP (Viceto et al., 2017) suggest that CAPE shows a high spatiotemporal variability: the values in winter and spring over land are small due to the reduced surface temperature, and the differences between Atlantic and Mediterranean regions are remarkable during summer. According to Siedlecki (2009), the mean values range from below 50 J/kg in the north to between 100 and 200 J/kg at the Mediterranean coast (some events can even reach 1000 J/kg). As Romero et al. (2007), Viceto et al. (2017) also stated that CAPE is low during autumn in the Atlantic and continental regions, but high in the areas surrounding the Mediterranean sea. This seasonality was also observed for other indices such as K-index or TT, which show maximum values during summer (Siedlecki, 2009). Observations proved that annual precipitation over eastern stations is mostly accumulated during autumn, as a result of the cumulative warming of the Mediterranean sea due to summer insolation (Romero et al., 2007; Iturrioz et al., 2007), and later entry of very hot and humid air into the IP while cold air is present at higher levels (Dai, 1999; Eshel and Farrell, 2001; Correoso et al., 2006). Additionally, September and October are the months with highest frequency of waterspouts and tornadoes near the Balearic Islands (Gayà et al., 2001). Over the northwestern IP, the mean values of the CAPE when hailstorms occur is 360 J/kg, while for thunderstorms it is only 259 J/kg (López et al., 2001). The dispersion of these values is really high (almost 350 J/kg over the whole sample), which is similar to that found in previous studies (Alexander and Young, 1992; Lucas et al., 1994). The values are similar to those observed in other regions of Europe, but lower than those values obtained in studies based on synoptic or lightning data for severe hailstorms (around 500 J/kg) (Kunz, 2007; Púčik et al., 2015; Taszarek et al., 2017). Due to global warming, the conditions necessary for the development of extreme precipitation events will be enhanced (Brooks, 2013; Rädler et al., 2019). The frequency and intensity of climate extremes will be magnified (Diffenbaugh et al., 2013), projecting larger values of CAPE at the Mediterranean coast during summer and autumn (Marsh et al., 2009; Viceto et al., 2017).

The main objective of this paper is to evaluate the performance of two simulations created by using the Weather and Research Forecasting (WRF) model (Skamarock et al., 2008) (including or not the extra 3DVAR data assimilation step) at reproducing the atmospheric conditions that can cause convective precipitation over the IP if the third ingredient (e.g. lifting) is fulfilled. We are not restricting our analysis only to convective situations, and the entire period from 2010-2014 will be considered. For the evaluation, the comparison of pseudo-soundings extracted from the model against real observations will be carried out. Additionally, the seasonal patterns of different variables commonly used to represent atmospheric instability will be studied. Moreover, this study will also help us to accurately determine the regions of the IP more prone to develop unstable thermodynamic conditions. If the condition of the forced lifting is also fulfilled, convective precipitation may occur in those areas. As shown before, atmospheric instability is a highly demanding feature in model simulations and a topic with great importance nowadays due to the large damage that extreme convective events can cause to society, and which frequency will be increased in the future. Thus, it is of great interest to diagnose the ability of particular configurations of a model to properly simulate the structure of temperature and moisture at low levels, which lead to atmospheric instability.

The novelty of this study lies in the inclusion of data assimilation step in the downscaling experiment used for the analysis of some instability indices, as most of the previous studies are mainly based on simulations driven by boundary conditions after its initialization (e.g. Marsh et al., 2009; Holley et al., 2014; Mohr et al., 2015). To those not familiar with the data assimilation process, its main objective is to produce more reliable and accurate initial conditions for regional models. This is achieved once the effect of the assimilated observations is used to modify the fields of temperature, wind and pressure in order to make them closer to the observations. The impact of the data assimilation is not restricted only to the location of the observations being assimilated. First, the improvements due to the analysis are propagated zonally, meridionally and vertically to the nearby grid points of the domain by means of the background error covariance matrix (Barker et al., 2004, 2012). Second, after the simulation in the new cycle is performed from the initial conditions achieved through assimilation, they propagate in the next six hours by means of advection, thus affecting areas distant from the original observations.

This paper is organised as follows: The details of the configuration of the WRF model used in both experiments are presented in section 2, along with a brief outline of the methodologies used in the study. The main results are presented in section 3, while they are compared against previous studies presented in the introduction. Finally, we conclude with some remarks about our research in section 4.

## 2 Data and Methodology

### 2.1 WRF model configuration

Two experiments were carried out using version 3.6.1 of the WRF model for the period 2010-2014. In both simulations, ERA-Interim provides the initial and boundary conditions (Dee et al., 2011). Six-hourly data at 0.75 degrees were downloaded from the Meteorological Archival and Retrieval System (MARS) repository at European Centre for Medium-Range Weather Forecasts (ECMWF). Analyses of temperature, relative humidity, both horizontal wind components and geopotential height at 20 pressure levels (5, 10, 20, 30, 50, 70, 100, 150, 200, 250, 300, 400, 500, 600, 700, 800, 850, 900, 925, 950, 1000 hPa) were

used to feed WRF. Both simulations were started on the 1st of January, 2009 from a cold start. Following similar methodologies to previous studies (Argüeso et al., 2011; Zheng et al., 2017), the entire year 2009 was selected as the spin-up for the land surface model included in WRF, and consequently, it was omitted in the study presented here.

One of the experiments (hereafter, N) was nested inside ERA-Interim as usual in numerical downscaling experiments, which means that the model is driven by the boundary conditions after its initialization. It is generated running 6-hour long segments that are restarted from the restart file produced at the end of previous segment, which is similar to a continuous WRF run where the boundary conditions are provided to the model every 6-hours after the initialization of the model. The other experiment (D) relies on the same setup, but with the additional 3DVAR data assimilation step (Barker et al., 2004, 2012) that is run every 6 hours (at 00, 06, 12 and 18 UTC). In this case, 12-hour long segments starting at every analysis time (00, 06, 12 and 18 UTC) are used. The analyses are generated from the outputs of the model at a 6-hour forecast step from the previous segment as first guess in a 3DVAR data assimilation scheme. In both experiments, the outputs are saved every 3 hours, which means that analyses (00, 06, 12 and 18 UTC) and 3-hour forecasts (at 03, 09, 15 and 21 UTC) are included in our results. In the data assimilation step, quality controlled temperature, moisture, pressure and wind observations in PREPBUFR format from the NCEP ADP Global Upper Air and Surface Weather Observations dataset (referenced as $ds337.0$ at NCAR's Research Data Archive) were included. Only those observations included in a time-window of two-hours centered in the analysis times were assimilated.

As Figure 1 shows, the domain focuses over the IP, but it also includes parts of Europe, Africa and the Atlantic ocean. As stated by previous studies (Jones et al., 1995; Rummukainen, 2010), the set-up of the domain used in this study prevents border-effects affecting our results as mesoscale systems can develop freely. The spatial resolution of both experiments is 15 km, and they include 51 vertical levels up to 20 hPa in eta ($\eta$) coordinates.

Apart from the ERA-Interim data, sea surface temperature (SST) of the model was updated on a daily basis using the high-resolution dataset *NOAA OI SST v2* (Reynolds et al., 2007) developed by the National Oceanic and Atmospheric Administration (NOAA). Additionally, the following parameterizations for the physics of the model were included in both WRF simulations: five-class microphysics scheme (WSM5) (Hong et al., 2004), MYNN2 planetary boundary layer scheme (Nakanishi and Niino, 2006), Tiedtke cumulus convection scheme (Tiedtke, 1989; Zhang et al., 2011), RRTMG scheme for both long and shortwave radiation (Iacono et al., 2008), and NOAH land surface model (Tewari et al., 2004).

The background error covariance matrices were created before running the simulation with 3DVAR data assimilation. To do so, the CV5 method included in WRFDA (Parrish and Derber, 1992) was used. A separate simulation initialized at 00 and 12 UTC and spanning 13 months (from January 2007 to February 2008) was necessary for the calculation of these matrices. Independent matrices were created for each month, and each of them was calculated taking into account a 90 days period centered on each month.

Both simulations were already presented and validated in previous studies by the authors. Integrated water vapor, precipitation and evaporation over the IP were validated against station measurements and gridded datasets including independent satellite data in González-Rojí et al. (2018), and the outputs produced by D were always superior to N and the driving reanalysis ERA-Interim (for the latter, at least comparable for some variables). The closure of the water balance was also better

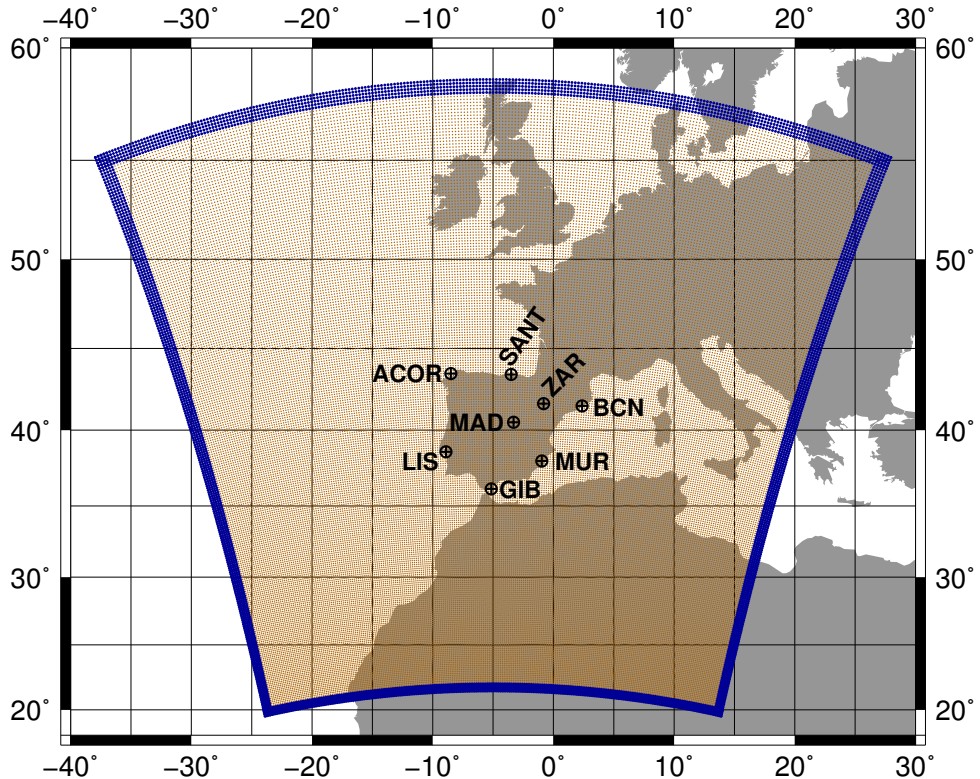

**Figure 1.** The domain used in both WRF simulations is presented with dark orange dots, while the dark blue region highlights the relaxation zone. The location of all the radiosondes available over the IP is also presented with quartered circles.

for D. Additionally, the precipitation from D exhibited similar capabilities to the one downscaled with statistical methods (González-Rojí et al., 2019). Furthermore, the wind field from D showed also improvements compared to ERA-Interim, and consequently, that data were used for the calculation of the offshore wind energy potential in the west Mediterranean (Ulazia et al., 2017). Afterwards, that study was extended to every coast of the IP (Ulazia et al., 2019). The moisture recycling over the IP was also evaluated in González-Rojí et al. (2020), highlighting the reliable results produced by the experiment including data assimilation and the importance of the moisture recycling at the Mediterranean coast during spring and summer.

The effect of data assimilation on moisture and temperature was measured by the analysis increments (analysis minus background) in González-Rojí et al. (2018). The effect of the data assimilation is more intense at 12 UTC compared to the other times, and particularly for summer (see their Figure 13). The spatial analysis of these values highlights that the effect of data assimilation is not homogeneous over the IP, and it concentrates mainly in the southeastern IP and both Guadalquivir and Ebro basins. Southern IP has been already highlighted by previous studies as a region where cold biases are observed in WRF simulations during summer (Fernández et al., 2007; Argüeso et al., 2011; Jerez et al., 2012). The fact that the effect of data assimilation is concentrated in that region in our WRF simulations is not a coincidence, and thus, the data assimilation helps to reduce that bias to some extent.

## 2.2 Radiosonde data

Atmospheric radiosonde data were downloaded from the server of the University of Wyoming (freely accessible at http://weather.uwyo.edu/upperair/sounding.html). Even if the University of Wyoming does not apply any quality control to the data, this dataset was already used in previous studies by the authors and none of the values were taken as erroneous. Moreover, data from the Integrated Global Radiosonde Archive (IGRA) created by NOAA was also included in this study. This dataset is available online after several quality control procedures.

Only eight radiosondes are available over the IP: A Coruna (ACOR), Santander (SANT), Zaragoza (ZAR), Barcelona (BCN), Madrid (MAD), Lisbon (LIS), Gibraltar (GIB) and Murcia (MUR). The location of each station is presented in Figure 1. Measurements are carried out every day at midday and midnight (00 and 12 UTC, corresponding to 02 and 14 LT summer time, respectively), with the exception of Lisbon where they are only available at 12 UTC (13 LT summer time). Additionally, the amount of data available for Gibraltar is extremely scarce since August 2012.

Temperature and mixing ratio were retrieved at all the available pressure levels at each location from the University of Wyoming database and from the IGRA dataset. Moreover, the values of TT, CAPE and CIN as calculated directly by the creators were also retrieved. However, only the values computed from the IGRA dataset were assumed as the reference in our analysis. Additionally, vertical profiles of temperature and mixing ratio downloaded from the University of Wyoming were also used to calculate TT, CAPE and CIN following our own methodology using the *aiRthermo* R package (further details can be found in the next subsection). The comparison between the original values of the indices retrieved and our results can give us information about whether their discrepancies are only due to differences in the calculation procedure.

It must be said that all the radiosondes presented here were assimilated during the 3DVAR data assimilation step in WRF. However, we do not assimilate directly any of the evaluated parameters or precipitation, as we mainly assimilate pressure, temperature, humidity and wind. Additionally, as already stated, only eight radiosondes are available over the IP. The validation of the results against the assimilated radiosondes (even if we don't assimilate directly the studied variables) can be seen as biased, but we cannot exclude some of these measurements from the data assimilation process only to be able to validate the simulation afterwards with such a reduced amount of data available (e.g. assimilate only four radiosondes, and validate the simulation with the remaining four radiosondes). Thus, in order to get the most accurate results as possible from the model, all the available measurements are used.

Moreover, we also assume that these radiosondes were very likely assimilated in ERA-Interim. Nevertheless, the impact of this is insignificant in our simulations as we only used ERA-Interim data as boundary conditions for our regional model after the initial run (1st of January, 2009). We only analysed the outputs from the model after one year of spin-up, so the results are only taking into account the variability corresponding to the regional climate model.

## 2.3 Methodology

### 2.3.1 Calculation of parameters representing atmospheric instability

For both simulations, the nearest grid point to the real latitude and longitude of each radiosonde was determined, and the corresponding pseudo-sounding (pressure, temperature and mixing ratio) at 00 and 12 UTC was obtained at WRF's original $\eta$ levels. We did not consider the averaged value of several grid points as we would be considering an extremely large area to be compared against radiosonde data. For example, if we consider an array of $3 \times 3$ grid points, we would be taking into account an area of 2025 km$^2$ (45 km $\times$ 45 km, as the horizontal resolution of our domain is 15 km), which is not suitable to

be compared against in-situ data. Additionally, according to Xu et al. (2015), for every sounding balloon, the vertical profile of the atmosphere up to 6 km is already measured for a drifting distance of 7.5 km (half the spatial resolution we use) even if the samples are taken during a clear or cloudy day (see their Figure 6). That means that our spatial resolution is suitable for the direct comparison of the nearest points against radiosonde data, as we do not neglect the horizontal drift of the sounding balloons. Thus, averaging the neighbour grid points would be suitable for the validation of results when convection-permitting

scales are used, but not in our case as the spatial resolution of our model run is 15 km.

Extracting pseudo-sounding from reanalysis or model data is nothing new, and Lee (2002) or Molina et al. (2020) amongst others showed that these pseudo-soundings are able to reproduce reasonably well the atmospheric conditions measured by real soundings. However, as highlighted by Holley et al. (2014), this procedure takes into account a stationary column at a fixed time, which can influence the comparison to real radiosonde data as these measurements are not instantaneous and not in a

220 straight vertical line as the balloons used deviate because of wind.

In order to calculate TT, CAPE and CIN using the pseudo-soundings from the model, the R package *aiRthermo* was employed (Sáenz et al., 2019). The most recent version was selected (version 1.2.1), which is publicly available in the CRAN repository (https://cran.r-project.org/package=aiRthermo). Both CAPE and CIN are calculated by means of the vertical integrals using discrete slabs defined by the resolution of pressure in the soundings (using all the available levels). The integrals for

each of the slabs enclosed by linear profiles are computed analytically, and the energy corresponding to each slab is accumulated, producing the final value of CAPE or CIN. The virtual temperature was used in every integral (Doswell and Rasmussen, 1994). Further details about the functions used for the calculation of the vertical evolution of the air parcels can be found in Sáenz et al. (2019) and also in the manual of the R package *aiRthermo* associated to that publication.

Additionally, in order to calculate CAPE and CIN in the most similar way to the University of Wyoming with the aim of

230 reducing the differences between the values due to different calculation procedures, the average of the lower vertical levels was set as the initial representative parcel (Craven et al., 2002; Siedlecki, 2009; Letkewicz and Parker, 2010). As in Siedlecki (2009), the averaged values from the lowest 500 m were used in this study. Furthermore, in order to avoid that the averaged initial parcel state is still too hot compared to the ambient conditions (in that case, CIN will never be computed as the parcel is already artificially buoyant), an isobaric precooling was applied if needed. To do that, the parcel is cooled along an isobar until

it crosses the sounding so that it is not buoyant at the initial state.

The TT index was calculated following the definition from Miller (1975). It is defined as:

$$TT = (T_{850} - T_{500}) + (D_{850} - T_{500}) \qquad (1)$$

where $T_{850}$ and $T_{500}$ are the temperatures at 850 and 500 hPa, and $D_{850}$ is the dew point temperature at 850 hPa. According to the ECMWF (Owens and Hewson, 2018), thunderstorms are likely when the values for this index are above 44°C. However, other values can be found in the literature: 48.1°C for southern Germany (Kunz, 2007), 46.7°C for the Netherlands (Haklander and Van Delden, 2003) or 46°C for Switzerland (Huntrieser et al., 1997).

It can be seen that this index is not highly dependant on the initial conditions for its calculation as it only depends on temperature at two discrete pressure levels, while CAPE and CIN are very sensitive to the initial conditions used for the simulated ascent. TT avoids this problem, but the results can suffer from errors due to inversion layers (Siedlecki, 2009). It must be pointed out that the dew point temperature is needed for TT, and that it is highly important for the calculation of the Lifting Condensation Level (LCL) while calculating CAPE and CIN. In the case of the radiosonde data, the indices are calculated using the measured dew point temperature at 850 hPa when is needed, while in our method, this variable is calculated from the temperature and mixing ratio at that pressure level. This can cause small differences in the results even if the same original radiosonde data are used.

Further indices could be calculated from the pseudo-soundings obtained from the outputs of the model or real observations. However, keeping in mind that the main objective of this study is to evaluate the difference in the performance of two simulations at reproducing the unstable atmospheric conditions that can cause convection, we needed to restrict our study to a small set of the indices calculated and provided directly by the radiosonde data holders used in this study: IGRA and University of Wyoming. By doing that, we can compare our results to those obtained by them, and infer which simulation performs better (that including data assimilation or the one without). In both cases, CAPE, CIN, TT index, LI, S index or K index are provided. In the case of the University of Wyoming, SWEAT is also included, but not in IGRA. Then, only six indices were available for us for the validation of our data. The R-package *aiRthermo* also allows the calculation of these six indices, so it was not a restricting feature in our analysis. However, previous studies reported a strong correlation between CAPE and LI (Blanchard, 1998; López et al., 2001), and K-index is also based on temperature at different pressure levels, so it suffers from the same problems as TT. Consequently, in order to avoid these connections between indices, we restricted this study to TT, CAPE and CIN.

### 2.3.2 Analysis

Once TT, CAPE and CIN are calculated at the nearest grid points to radiosonde locations of both simulations (N and D), and also those using the original sounding data from the University of Wyoming (labelled as 'aiRthermo' in the results), we obtain a time series with a 12-hourly temporal resolution for each index. These values can be compared against the reference values of the indices retrieved directly from the University of Wyoming and those computed from the IGRA dataset (labelled as 'Wyoming' and 'Reference' respectively in the next figures). The comparison between 'Wyoming' and 'aiRthermo' aims to achieve an estimation of the error/differences due to the different methods applied by both sources of results. This comparison

was based on independent locations over the IP (separated by several kilometers), so a Taylor diagram was chosen as the best option to show Pearson's correlation (r), Root Mean Squared Error (RMSE) and Standard Deviation (SD) of each experiment in the same plot. In order to determine which experiment is doing the best job at simulating the reference values of the variables, the procedure explained by Taylor (2001) was followed: the dots that lie nearest to the reference on the $X$ axis represent variables that agree well with observations (high correlations and low RMSEs), and those lying near the highlighted arc will present comparable standard deviations to the observations.

Additionally, the bootstrap technique with resampling was applied to the results in order to represent an estimation of the sampling errors from each experiment (Efron and Gong, 1983; Wilks, 2011). In our case, the original time series used in the Taylor diagrams consist of 60 values, each of them for the corresponding month along the period 2010-2014 (12 months $\times$ 5 years). For the bootstrap, we created 1000 perturbed time series taking into account different samples of the data. 67 % of the new time series (2/3 of the length of the original time series – 40 values in our case) is made from the original data, and the remaining 33 % (1/3 – 20 values) is chosen from those values already taken from the original data. For each correlation calculated, the same samples are taken from all datasets and experiments. The variability of the Pearson's and Spearman's correlations obtained with these synthetic time series were shown by means of Box-Whiskers plots.

Then, the seasonal analysis of each parameter at each location was carried out. In this case, the variability of the results is showed by different Box-Whiskers plots. Each season was defined as follows: winter is defined from December to February (DJF), spring from March to May (MAM), summer from June to August (JJA) and autumn from September to November (SON).

The main objective of this paper was to analyze the ability of the model to properly simulate atmospheric conditions by means of TT, CAPE and CIN. Thus, the calculation of TT, CAPE and CIN was extended to every grid point included in a mask defined for the land points of the IP over model's domain. The spatial distribution of the mean values of them at 00 and 12 UTC during winter and summer was calculated. These maps show the spatial distribution of TT, CAPE and CIN over the land grid points in the IP which are more prevalent in each season. However, from the point of view of the applicability of these results to the evaluation of unstable atmospheric conditions, it is also important to analyze the joint distribution of CAPE and CIN limited to those days characterized by high values of CAPE. In order to select those days, the 75th percentile of CAPE at every grid point, season of the year and time (00 and 12 UTC) was used as a threshold. Only those days in which CAPE was above this percentile (labeled as P75) were considered to calculate the mean value of CAPE and CIN.

## 3   Results

Taylor diagrams for the TT index calculated for each radiosonde of the IP are shown in Figure 2. The Box-Whiskers associated to the correlations (both Pearson's and Spearman's) obtained for each of the 1000 time-series created with the bootstrap technique are also included. According to the Taylor diagrams, the best experiment reproducing the reference values is Wyoming, followed by aiRthermo (the real measurements of temperature, mixing ratio and pressure from the sounding were used to calculate TT with our methodology), D and later by N. Wyoming obtains the closest values to the observations at all the stations.

The results for aiRthermo are quite similar to Wyoming, except for Murcia where D is better reproducing the reference data. The correlations are always above 0.99 for Wyoming, 0.98 for aiRthermo, 0.97 for D and 0.75 for N. The observed SD is really well simulated by Wyoming, aiRthermo and D, but N underestimates it at most of the stations as it is only able to reproduce

the one in Santander and A Coruna. The RMSE is below 0.6 °C for aiRthermo, below 1 °C for D and below 2.5 °C for N.

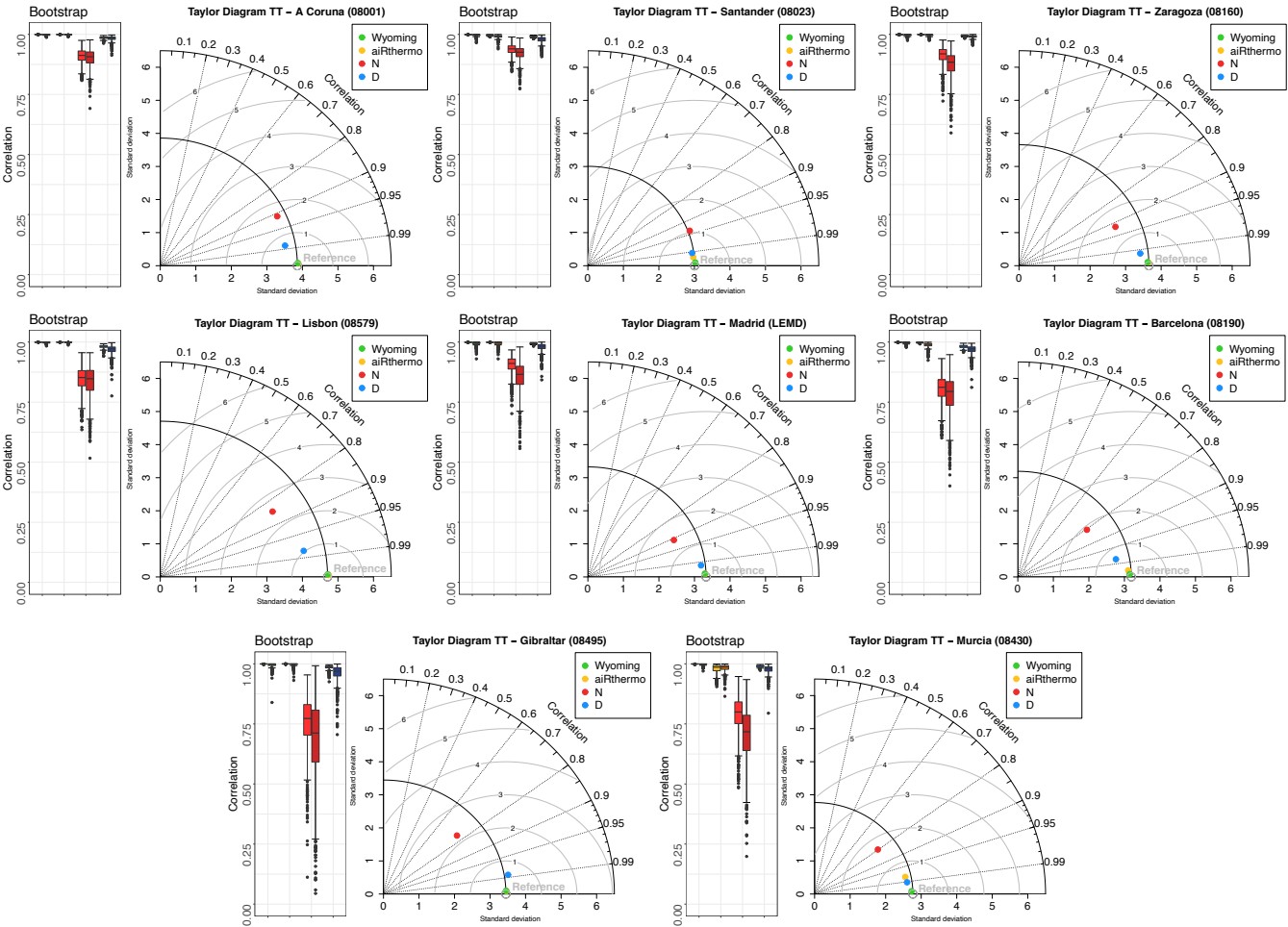

**Figure 2.** Taylor diagrams showing the r, RMSE and SD values for Wyoming, aiRthermo, N and D compared to TT values computed from IGRA (Reference). On the left side of each Taylor diagram, a Box-Whiskers plot is added in order to show Pearson's and Spearman's correlations between each experiment and the reference data (lighter and darker colours, or first and second columns of the Box-Whiskers respectively). The bootstrap technique with resampling was used to create 1000 synthetic time-series. Wyoming, aiRthermo, N and D are plotted in green, orange, red and blue respectively.

The bootstrap analysis is consistent with the results obtained in the Taylor diagrams, and it shows that the Pearson's correlations are always above 0.99 for Wyoming and 0.98 for aiRthermo (again, with the exception of Murcia where they are above 0.9). The correlations are always above 0.95 for D. In the case of N, the spread of the values is much larger than for aiRthermo

and D, and their median values are obtained between 0.8 and 0.9. If we change to Spearman's correlations, we can see that values are similar but with a small decrease of the values (particularly in Gibraltar, Murcia and Madrid).

Thus, as expected, we obtain the most similar results to those calculated from IGRA (Reference) with the values from Wyoming and those calculated with the real measurements from the soundings (that is, aiRthermo). However, we can still detect small differences between the values of the datasets due to the use of measured dew point temperature in Wyoming, whilst it is computed from temperature and mixing ratio in IGRA and aiRthermo. These differences are more remarkable in Murcia. Between both WRF experiments, it is clear that the experiment including the 3DVAR data assimilation is able to outperform the standard simulation only driven by the reanalysis data at the boundaries of the domain (N). The differences between both WRF simulations are highlighted, particularly at those stations located at the Mediterranean coast (Barcelona, Murcia and Gibraltar) and in Lisbon.

In the case of CAPE, the validation results are presented in Figure 3. The best experiment reproducing the results is aiRthermo, followed by Wyoming, D and finally by N. The correlations are at all the stations above 0.99 in aiRthermo and 0.95 for Wyoming, while for D they are above 0.9 and above 0.7 for N. A similar behaviour is observed for SD and RMSE. The largest RMSEs are obtained for Barcelona and Murcia (both in the Mediterranean region).

The bootstrap analysis shows that the highest Pearson's correlations are obtained by aiRthermo and Wyoming, but followed really closely by D. As for TT index, N presents the worst performance and the largest spread. If we consider, instead, the use of Spearman's correlations, we can see that the values are similar at most of the stations and only in A Coruna and Santander there is a strong decrease of the values.

As stated before in Section 2.3.1, the calculation of CAPE is more sensitive to subtle differences in the methodology than that of TT, and this is highlighted in the validation of these results. Even if the same data are used for the calculation of CAPE (Wyoming and aiRthermo used the same measurements as input), it is clear that small differences in the initial conditions can result in serious discrepancies between both methods as stated by Siedlecki (2009). The largest RMSEs for aiRthermo can be found at Barcelona and Murcia. As for the TT index, two stations at the Mediterranean coast present the largest differences between the experiments. However, while the computation of TT from both WRF simulations produces standard deviations similar to the observed ones, the results for CAPE substantially overestimate the variance of Atlantic sites (A Coruna, Santander and Lisbon) and Madrid, or underestimate it at the Mediterranean coast (Barcelona, Murcia and Gibraltar). Anyway, it can be seen that data assimilation improves the simulation of CAPE over the IP.

Finally, the validation of CIN is presented in Figure 4. As for CAPE, the best results are obtained again by aiRthermo, followed by Wyoming, D and N (with the exception in Gibraltar where D and N are really similar). aiRthermo obtains in every station correlations above 0.97, followed by Wyoming, D and N with correlations above 0.93, 0.85 and 0.65 respectively. Both WRF experiments overestimate or underestimate it depending on the station (particularly N in Lisbon, Madrid, Murcia and Zaragoza). The RMSE is always larger for N, and particularly in Murcia and Gibraltar where the values exceed the 40 J/kg.

The bootstrap analysis presents the same results as for CAPE (Figure 3). However, for Gibraltar, as shown in the Taylor diagram, both WRF experiments produce similar Pearson's correlation values during the bootstrap. If we consider, instead, Spearman's correlations, the worse performance of N is perceptible in A Coruna, Santander, Murcia and Gibraltar. However, in

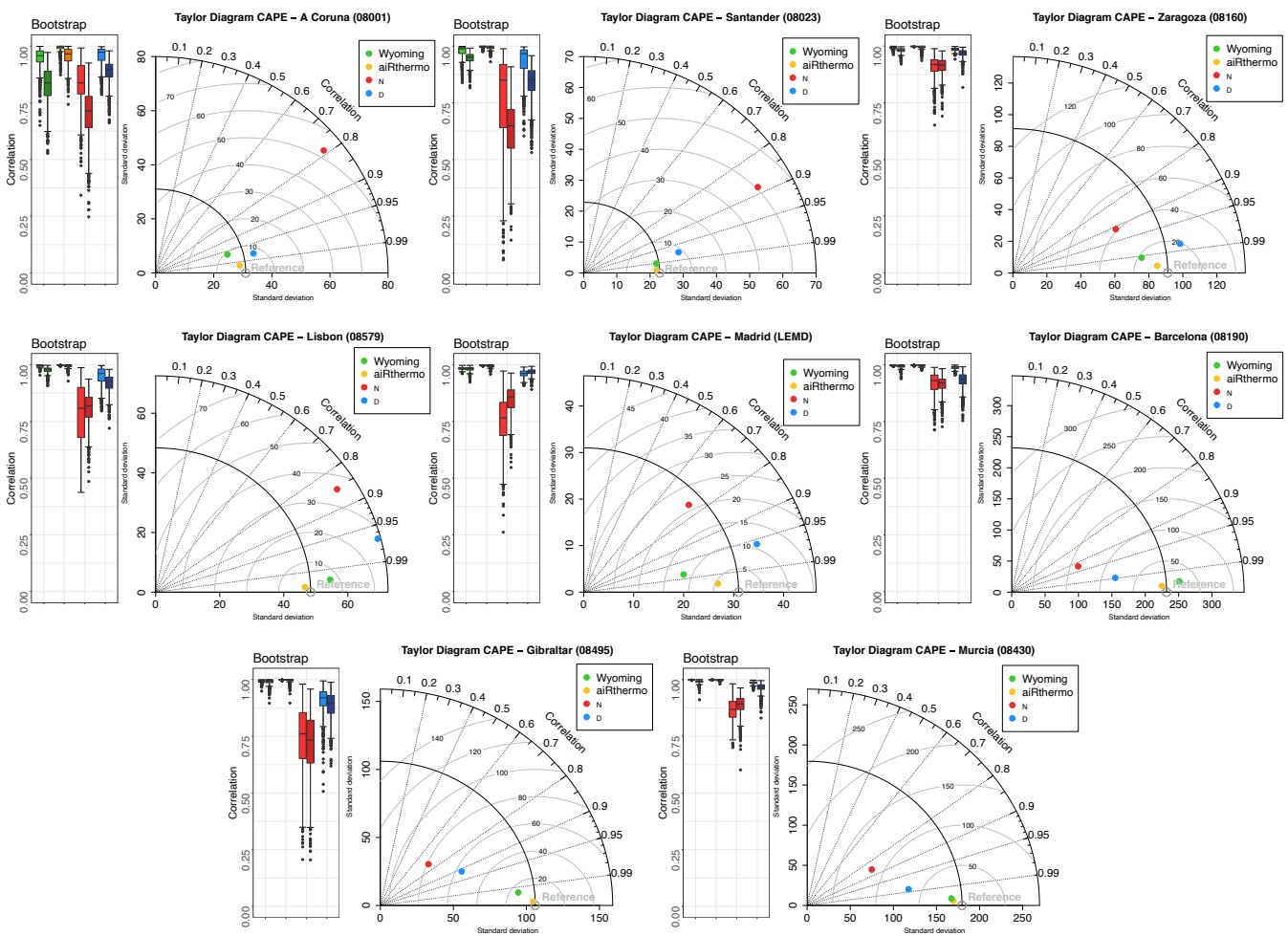

**Figure 3.** Same as Figure 2 but for CAPE.

Gibraltar, differences between both WRF experiments arise: WRF D obtained better correlations than WRF N as in the other
stations. In contrast to previous results, the poorest correlations for CIN are obtained at stations located at the Atlantic coast as
Lisbon and A Coruna.

As for CAPE, the differences between aiRthermo and Wyoming are highlighted here. This result supports the idea that small
differences in the initial conditions of the lifted air parcel and the determination of the LCL due to differences in the dew point
temperature can cause large differences in the values of CIN even if the same vertical profile of temperature and mixing ratio
are used for its calculation. Again, the differences between both WRF experiments are important and the experiment including
data assimilation (D) presents generally closer results to the observed ones.

The seasonal analysis of the five datasets (Reference, Wyoming, aiRthermo, N and D) for TT index is presented in Figure 5.
In this case, Wyoming, aiRthermo and D are able to correctly simulate the reference seasonal variability of TT index at all the

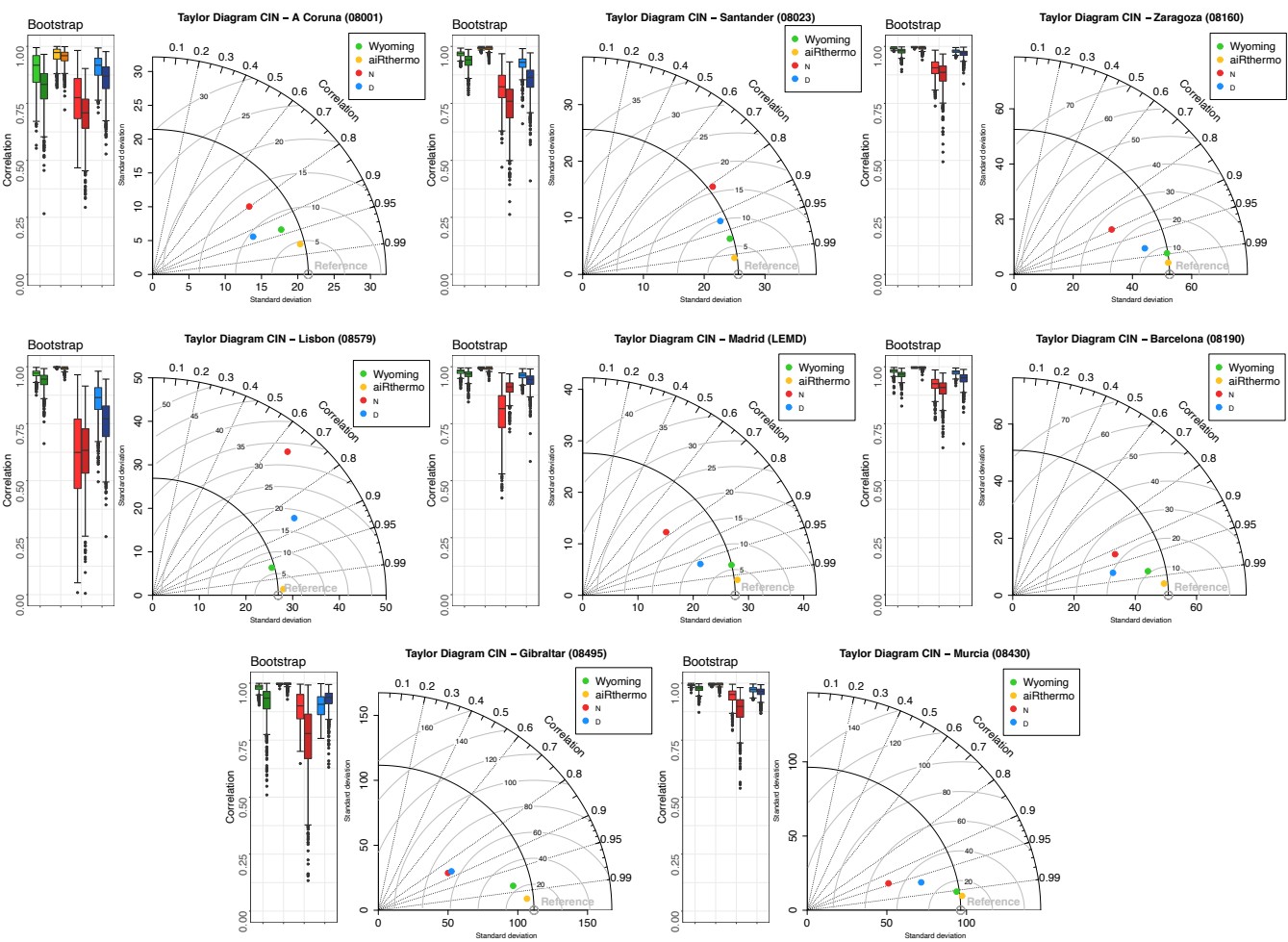

**Figure 4.** Same as Figures 2 and 3 but for CIN.

stations and all the seasons. However, N tends to overestimate the values of TT in every season and for most of the stations over
355 the IP. The difference in TT is most important during winter months as a severe overestimation by the simulation N without
data assimilation can be seen. It can be tracked on the basis of Figures S1-S3 (from the supplementary material) to an improved
representation of temperature at 500 hPa. As shown by Figure S1, the winter temperature at 850 hPa is higher for D than for
N, but this would lead to higher values of TT, so that it does not explain the observed discrepancies for the N simulation. For
the case of dew point temperature at 850 hPa (Figure S2), it is higher for the N simulation than for D, which leads to higher
values of TT for N. Additionally, the temperature at 500 hPa is higher for D than for N, and this also leads to higher values of
TT for N than for D. It is thus, clear, that the improvement in the simulation of TT during winter for the D simulation is due
to an improved simulation of moisture and temperature at the low (850 hPa) and mid-troposphere (500 hPa) derived from the
assimilation of soundings. The same diagnostic can be done for spring, another season during which N overestimates TT in

many of the soundings (A Coruna, Santander, Lisbon and Barcelona). They are located in areas where the difference between
365 the dew point temperature from simulation D minus the one from N is negative (see Figure S2).

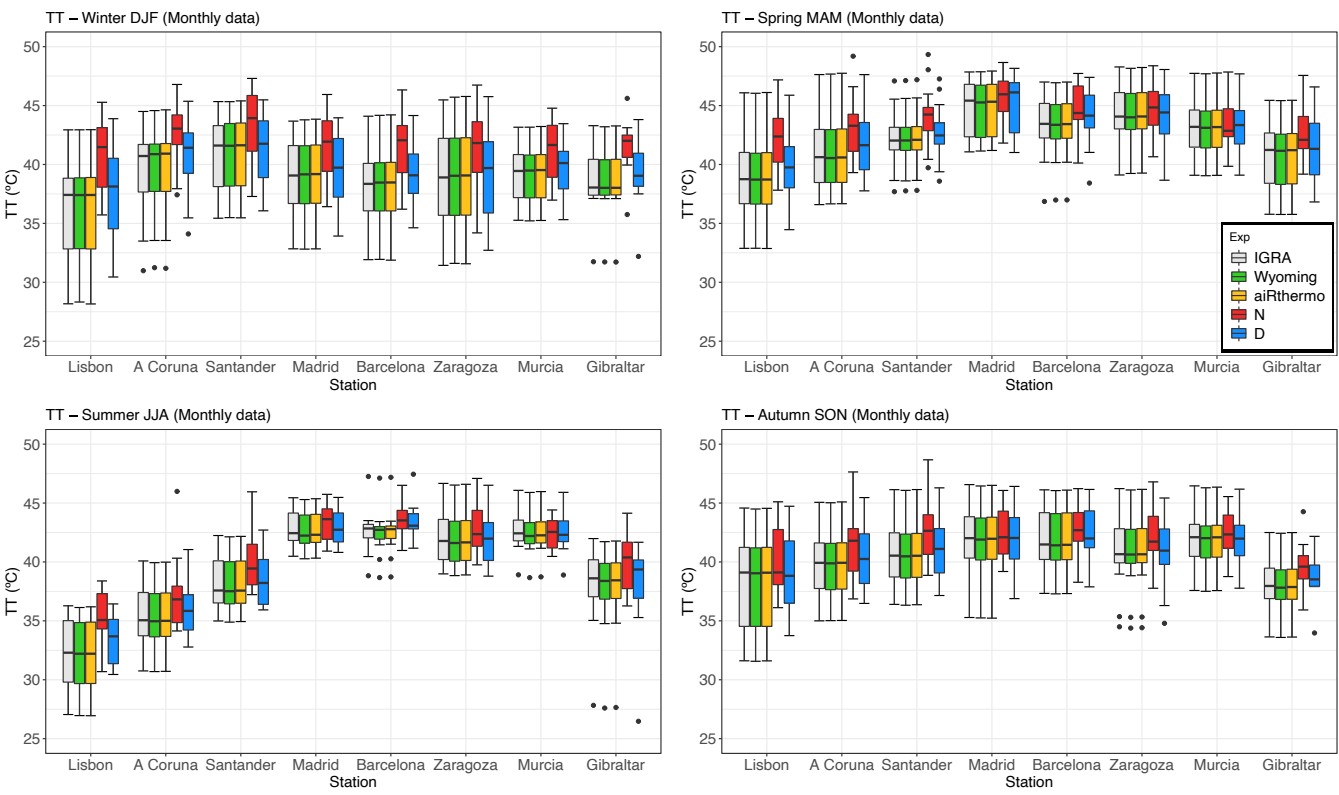

**Figure 5.** TT index for the reference data (grey), Wyoming (green), aiRthermo (orange), N (red) and D (blue) computed at each station for every season: winter (top panel, left), spring (top panel, right), summer (bottom panel, left) and autumn (bottom panel, right).

A Coruna and Santander present the largest values during winter. Higher values than in winter are observed during spring, and the maximum is recognizable in Madrid, which is the only station located over central IP. Even if the maximum is found there, the other stations also present values above 38°C. During summer, central, eastern and southern stations (Madrid, Barcelona, Zaragoza and Murcia) are the ones presenting higher values. In that season, the Atlantic stations (A Coruna and Santander)
and Gibraltar present values below 40°C. The values of TT in summer at those stations are smaller than the ones in winter, which occurs mainly due to the combined effect of the high increasing values of temperature at 850 and 500 hPa (about 15 and 10 degrees respectively) and the smaller increase of dew point temperature (only a few degrees) in those regions from winter to summer (see Figures S1-S3 from the supplementary material). Finally, all the stations show similar values in autumn, with the exception of Gibraltar where the values are smaller.
The seasonal analysis for CAPE is presented in Figure 6, and it highlights the spatial and temporal heterogeneity of the areas where unstable air masses can be observed over the IP, as also shown by Holley et al. (2014). Wyoming and aiRthermo are able

to reproduce (as expected) the variability of the reference values, and D is able to capture the spread of the values at most of the stations during winter, summer and autumn. However, both WRF experiments (particularly D) overestimate CAPE at most of the stations in spring due to the differences in the virtual temperature in lower levels compared to reference data (colder near surface and warmer near 800 hPa) and with lifted parcels for D slightly warmer than the reference ones and those for N (see Figure S4).

The experiment without data assimilation (N) tends to overestimate CAPE in winter and to underestimate it in summer. In winter, this overestimation is caused mainly by colder conditions in the 850-750 hPa pressure levels and warmer lifted air parcels (particularly in Lisbon, A Coruna and Santander - See Figure S5). A detailed analysis of the vertical structure of the differences of both simulations against IGRA for virtual temperature and mixing ratio (Figure S6) shows that the vertical structure of moisture is improved in the D simulation, thus leading to a better estimation of CAPE through the troposphere due to improved estimations of virtual temperature.

On the contrary, in summer, the underestimations of CAPE by the N simulation are caused by warmer conditions in the lower pressure levels compared to the reference, which produces that the lifted parcel crosses the sounding in a lower pressure level than D, and consequently, underestimating CAPE (particularly for Barcelona, Murcia and Gibraltar - See Figure S7). A detailed analysis for Barcelona (Figure S8) shows that there exists a substantial underestimation of moisture at lowest levels by the N simulation, something which is consistent with findings in González-Rojí et al. (2018) in a verification with independent non-assimilated MODIS integrated water vapour. This is also observed in Murcia (Figure not shown). During spring and autumn, the underestimations or overestimations of N depend on the station and a clear pattern is not observed.

The lowest values of CAPE are obtained during winter (below 50 J/kg at all the stations), and the largest ones are observed in summer (reaching 500 J/kg at some stations). However, as stated before, the distribution of CAPE is not homogeneous and different regions are prone to higher values during each season. During winter, the three Atlantic stations (A Coruna, Santander and Lisbon) and Gibraltar present the highest values of CAPE over the IP. In general, the values are below 50 J/kg, but some events can exceed 100 J/kg. During spring, the distribution of CAPE is quite homogeneous over the IP and only stations such as Lisbon, Madrid or Gibraltar present slightly higher values of CAPE than the other stations. In summer, only the stations located in the eastern and southern parts of the IP present remarkable values of CAPE. Particularly, the highest CAPE values are located at the Mediterranean coast (Barcelona and Murcia). Finally, during autumn, the regions with high CAPE are extended towards the inland of the IP, such as Madrid and Zaragoza. During this season, some extreme events can reach values over 1000 J/kg over the Mediterranean coast. This feature was already observed by Siedlecki (2009). All these seasonal changes in CAPE also agree with previous studies based on CAPE (Romero et al., 2007; Viceto et al., 2017).

Finally, the seasonal analysis for CIN is presented in Figure 7, and it highlights the stations where the inhibition is important. In general, Wyoming tends to underestimate the values of CIN at most of the stations and in every season, while aiRthermo is able to capture it. Both WRF simulations (but particularly the experiment without data assimilation) tend to underestimate the observed variability.

The values of CIN are smaller in winter and spring, and the maximum is observed in summer. During winter, CIN is higher in Gibraltar and at the Atlantic stations (Lisbon, A Coruna and Santander) than at the other stations from the IP. However,

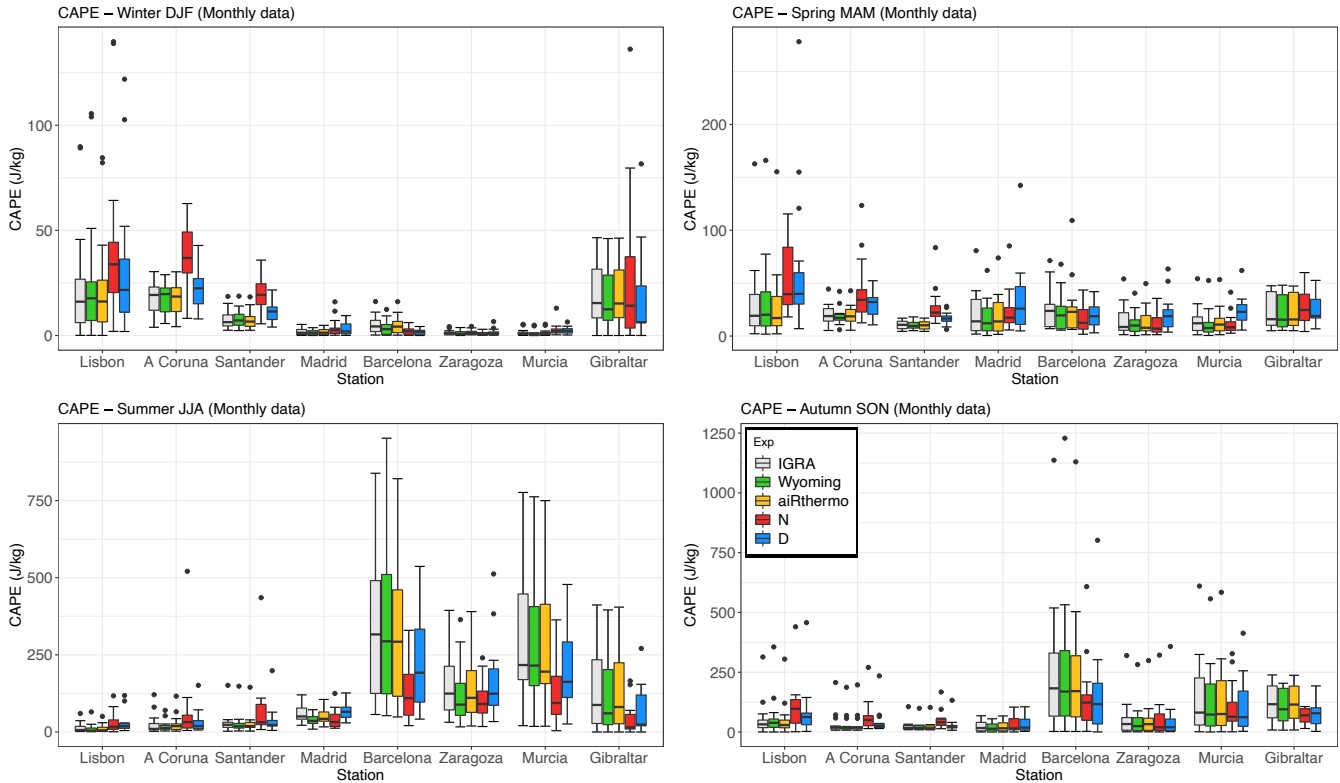

**Figure 6.** Same as Figure 5 but for CAPE.

these values are small compared to those for other seasons. During spring, the values are higher than in winter, and similar values are observed at most of the stations (around 10 J/kg), with the exception of Barcelona where the CIN reaches values of 20 J/kg. In summer, the values are higher at all the stations, but particularly in those for the eastern and southern IP (Barcelona,

Zaragoza, Murcia and Gibraltar). The same regime is observed for autumn, but the values are smaller than in summer. These values during summer and autumn are in agreement with Siedlecki (2009), who found CIN means above 100 J/kg in the west Mediterranean sea and surrounding countries.

As stated before, in the final phase of this study, the same procedure for the calculation of TT, CAPE and CIN at each station was extended to each grid point included in the IP. The mean winter and summer spatial patterns at 00 and 12 UTC were

420 calculated for both WRF experiments. In addition, CAPE and CIN limited to those days characterized by high values of CAPE (based on the 75th percentile of CAPE at each grid point, season of the year and time - 00 and 12 UTC) were also evaluated. These maps were added to the corresponding figures for CAPE and CIN as a third column. However, these results are only shown for the D experiment, the one that was shown to be the most accurate one according to the Taylor diagrams and seasonal Box-Whiskers plots in previous results.

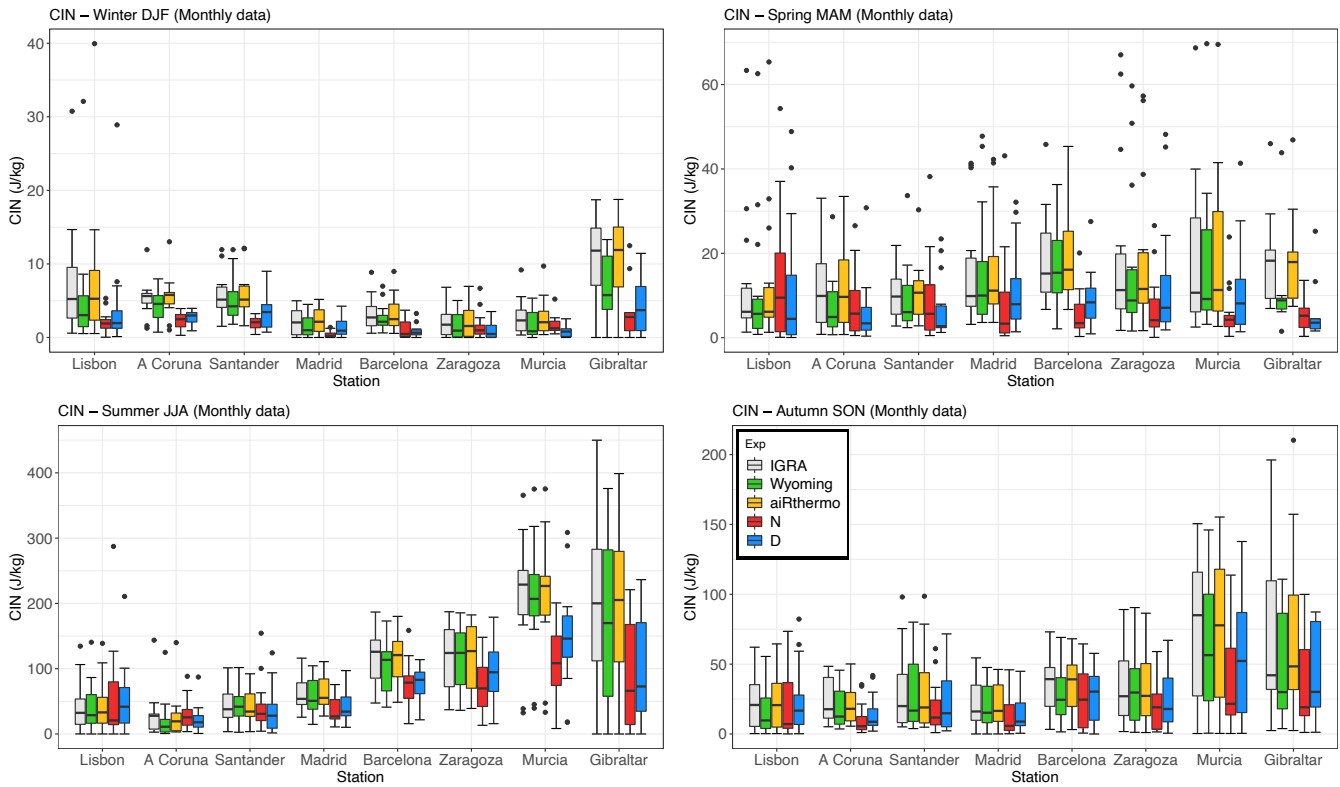

**Figure 7.** Same as Figures 5 and 6 but for CIN.

The spatial distribution for TT is shown in Figure 8, which highlights the heterogeneity of the results. The differences between both simulations are observable, but also those between day and night. Additionally, it can be seen that TT cannot be calculated in most of the mountain regions of the IP because the 850 hPa layer is near the surface or below ground.

During winter, the maps of TT show that N yields higher values than D, which is in agreement with the overestimation observed in Figure 5. At 00 UTC, according to D, the regions where unstable air masses are observed are those at the Cantabrian coast and in the southeastern IP. Both regions are surrounded by remarkable mountainous systems such as the Cantabrian Range and the Baetic system, which can trigger convection by orographically induced lifting. For N, these areas are also extended to the rest of the IP, with the exception of the southwestern corner where the values are small. At 12 UTC, after solar irradiance has started heating up the land, the regions extend towards inland areas. In the experiment with data assimilation (D), most of the northern plateau presents high values of TT, and the lowest values are observed near the coastal valleys of the southwestern corner and the Mediterranean coast (like the Ebro basin or Murcia). In the case of N, the lowest values are observed mainly in the southwestern IP near the Guadalquivir valley. According to Figures S1-S3, the lowest values of TT observed near the coastal valleys of both WRF experiments are a consequence of low dew point temperatures there mainly due to dry air.

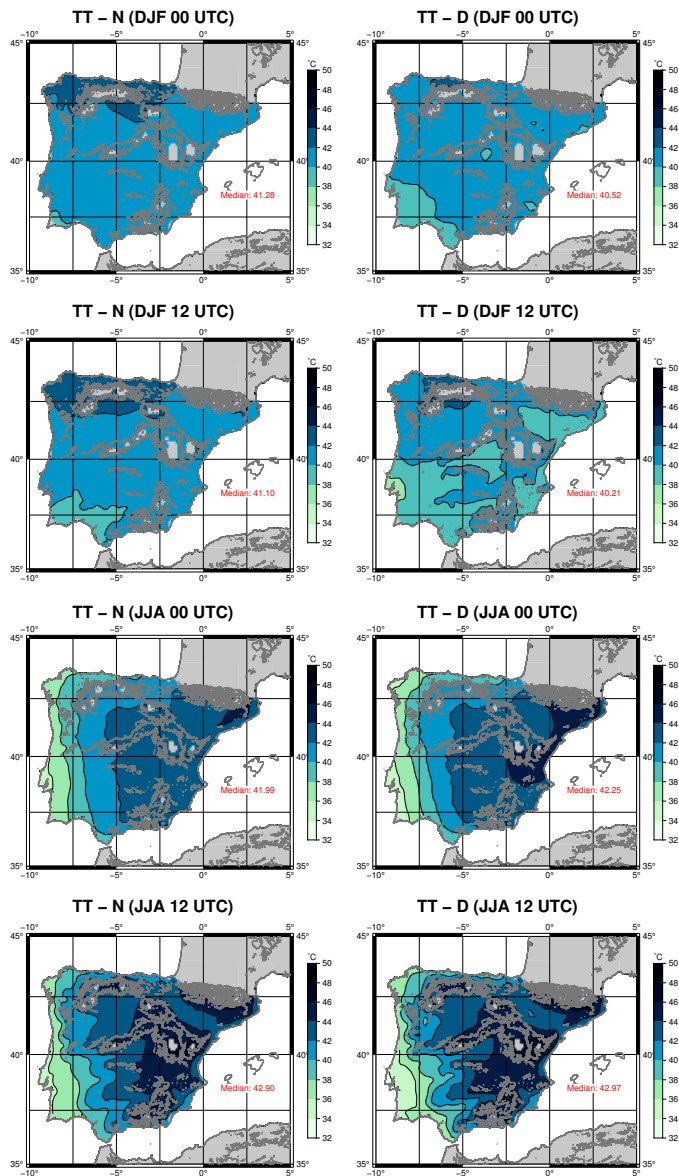

**Figure 8.** Spatial distribution of mean TT for period 2010-2014 over the IP as computed from N (first column) and D (second column) for winter (rows 1 and 2) and summer (rows 3 and 4) at 00 (rows 1 and 3) and 12 UTC (rows 2 and 4). The median value (°C) of each map is presented in the bottom right corner of the plots.

As in Figure 5, much higher TT values are obtained during summer over the IP, particularly at 12 UTC. At 00 UTC, a west-east gradient is observed in both WRF simulations. However, the values depicted at the Mediterranean coast are higher for the experiment including data assimilation (D). At 12 UTC, the regions with unstable air masses extend towards the central area. In this case, they are located near the Pyrenees, and in the proximity of the Iberian, Central and Baetic systems. The minimum

TTs are observed in the western part of the IP, but particularly near Lisbon. The intensity of the most extreme values of TT is higher in D (with data assimilation).

The maps of CAPE at 00 and 12 UTC during winter and summer are presented in Figure 9, together with the mean values of
CAPE that are larger than the 75th percentile at each point of the domain for the D experiment (P75 column). During winter, as shown in Figure 6, the N experiment presents higher values than D. At 00 UTC, the patterns are really similar for both WRF experiments. The main difference between them is observed at the western Atlantic coast of the IP, where higher CAPE values are obtained for N. At 12 UTC, the unstable air masses are found at the western coast of the IP in both simulations, and they extend further inland than at 00 UTC, particularly near the Tagus and Guadalquivir rivers. Again, the values are higher for
450    N, but the pattern is similar in both experiments. If the analysis is limited to values of CAPE beyond the third quartile during winter, the values are higher, but the spatial distribution at both 00 and 12 UTC is very similar to the average one from the D experiment.

Compared to what is observed during winter, CAPE is higher during summer for the experiment including data assimilation. This is in agreement with the station analysis from previous Figure 6. At 00 UTC, the area with higher CAPE is observed in
the northern and eastern IP, but particularly near the Mediterranean coast. However, at 12 UTC, this area with high values (over 250 J/kg) extends towards the interior and in the experiment including data assimilation it also covers the southern part of the Pyrenees. Additionally, high values are observed in most of the IP (except the southwestern corner for N), but particularly in the simulation including data assimilation. The patterns of CAPE obtained for both winter and summer are in agreement with those from Viceto et al. (2017), even if we differentiate between 00 and 12 UTC and we studied different periods (1986-2005
in their study, 2010-2014 in ours). If the focus is set on the days characterized by CAPE higher than the third quartile (third column of Figure 6), it can be seen that, as expected, the CAPE field is intensified but the changes in its spatial distribution are not relevant. The Mediterranean region is still the one showing the highest values of CAPE, particularly at 12 UTC, even though there is a general small increase of CAPE over the entire IP.

Finally, regarding CIN, the maps for the mean values at 00 and 12 UTC during winter and summer are presented in Figure 10.
In reverse to what we found for CAPE, CIN is usually higher at 00 UTC than at 12 UTC (with the exception of Murcia in summer at 12 UTC), as it could be expected due to the stabilizing effect of nocturnal radiation. During winter, at 00 UTC, both simulations show small values over the IP, and only some high values are observed in the western and southwestern corners of the IP (and particularly for N) because of higher atmospheric stability due to surface cooling. At 12 UTC, the areas are confined to those coastal regions, but they also extend to the Mediterranean coast in the D experiment. For the days characterized by
CAPE higher than the third quartile (rightmost column of Figure 10) in D during winter, it can be seen that the highest values of CIN appear at night at the Mediterranean coast. This area is usually where nocturnal radiation cooling is large because of a lower cloud cover, although this result is highly dependent on the dataset used for the analysis (Calbó and Sanchez-Lorenzo, 2009), and consequently, it should be further analysed for our simulations in the future. Conversely, during day-time, the structure of CIN for the days in which CAPE is higher than the 75th percentile does not show any remarkable feature.
During summer, at 00 UTC, the most remarkable values are obtained in both simulations along the Ebro basin and near the Mediterranean coast. However, the CIN inland is higher for D. At 12 UTC, less inhibition is observed in the eastern valleys

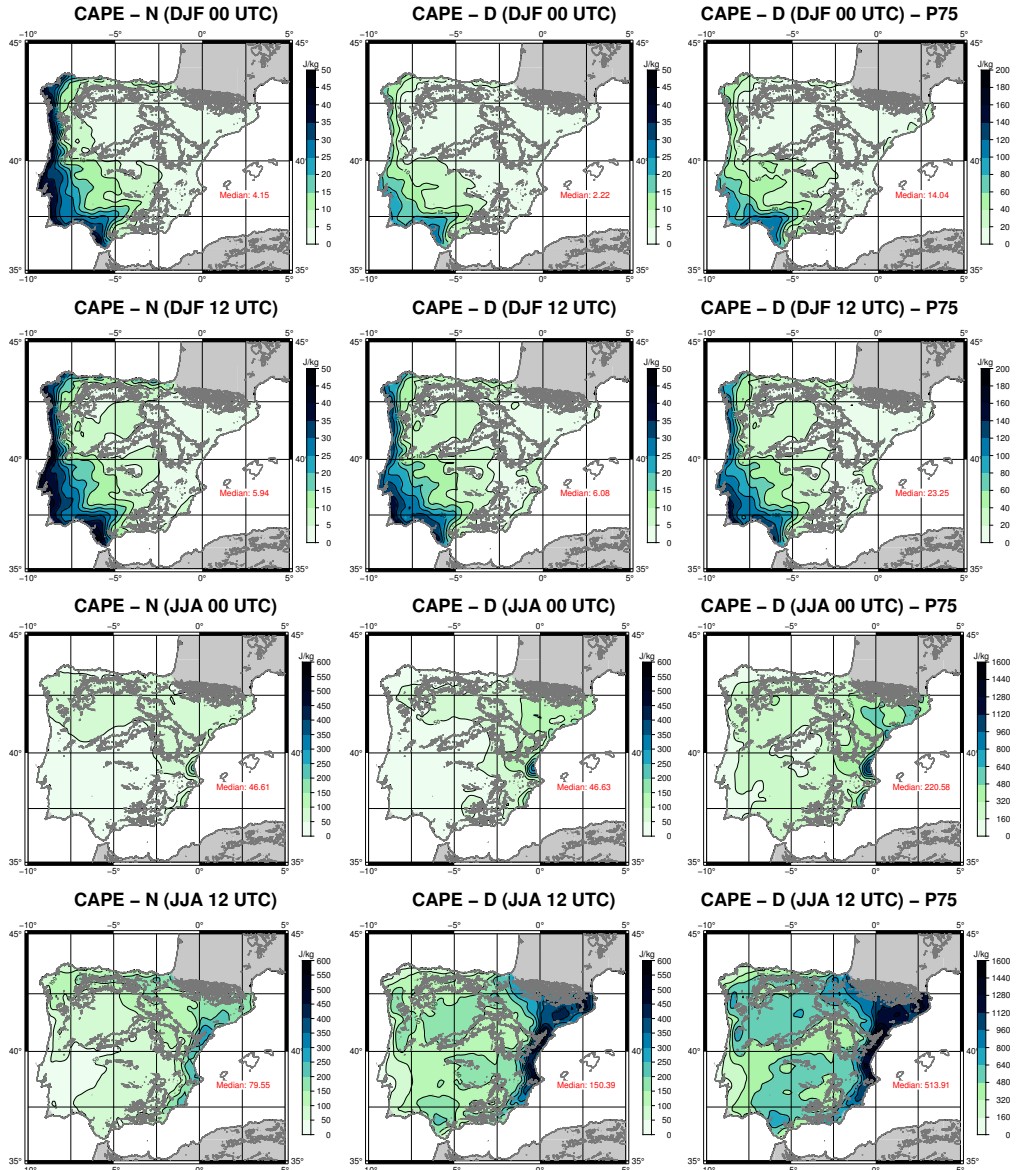

**Figure 9.** Same as Figure 8 but for CAPE. The rightmost column (P75) shows the spatial distribution of the mean values of CAPE in the D experiment for those days in which CAPE is higher to the third quartile of the sample of CAPE at every grid point.

of the IP (with the exception of Murcia, which presents extremely high values of CIN). At the same time, an increase in the convective inhibition over the Guadalquivir basin is shown. The extension towards the interior is again higher for D (including data assimilation). During summer, the selection of the days with the highest values of CAPE implies again the change in

the distribution of nocturnal CIN so that large areas of the southern IP are affected by the stabilizing effect due to radiational surface cooling during the frequent clear nights beyond the Mediterranean regions which were also apparent in the average

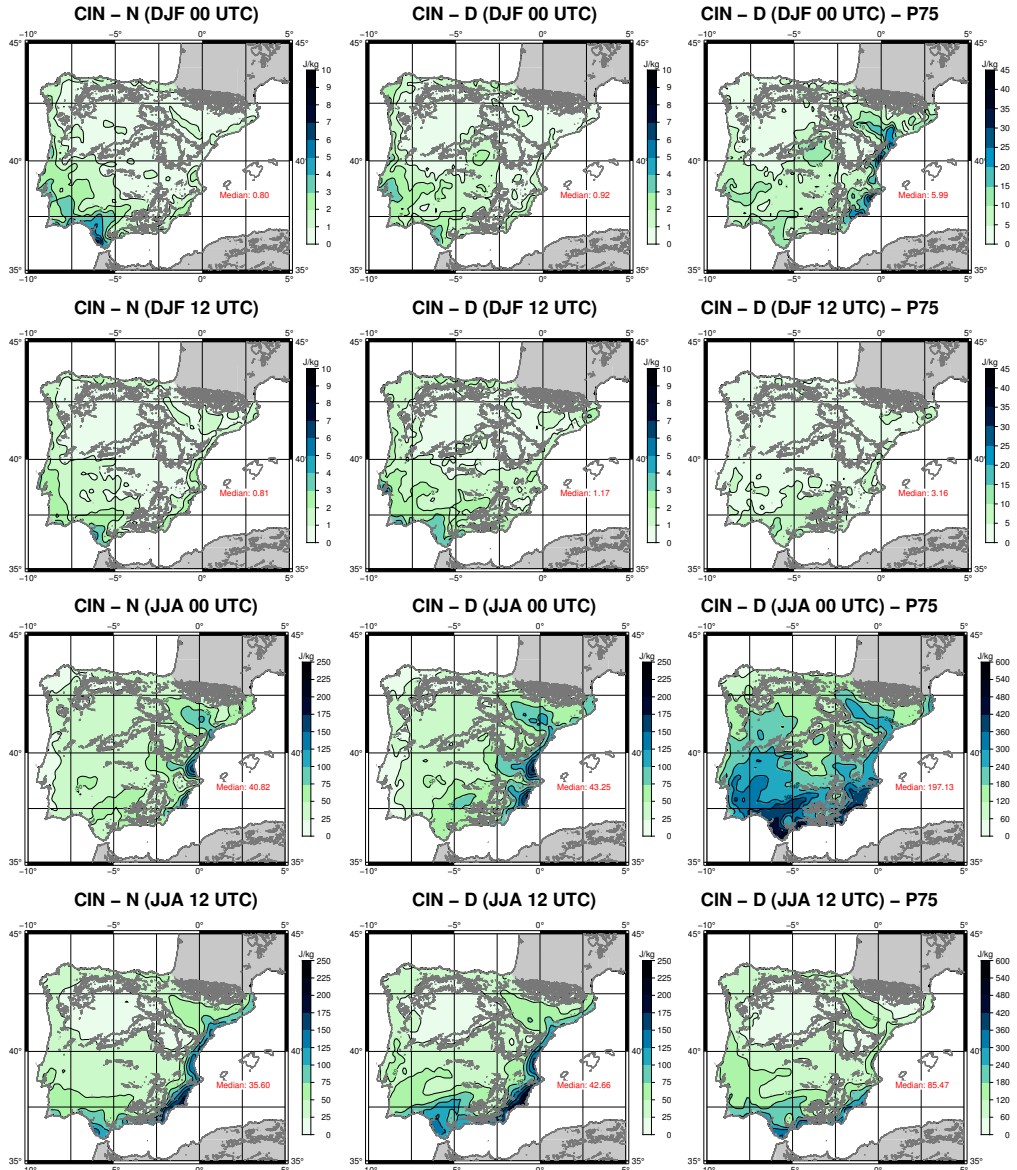

**Figure 10.** Same as Figures 8 and 9 but for CIN. The rightmost column (P75) shows the mean of CIN in the D experiment during those days in which CAPE is higher or equal than its third quartile at every grid point.

night (00 UTC) map. During midday (12 UTC), the values of CIN for the days in which CAPE is above the 75th percentile are larger, but the spatial distribution does not change compared to the map showing the mean CIN.

Comparing the results shown in Figures 5 and 6 (or in Figures 8 and 9), it seems that there is a discrepancy between the results for TT and CAPE since maximal values of these indices are not observed in the same regions. However, it must be

taken into account that these results for CAPE (and CIN) are obtained from the entire series of 12-hourly values obtained during 2010-2014. Thus, Figures 6 and 7 (also Figures 9 and 10) must be compared in combination to the values of TT.

Additionally, atmospheric conditions can be analysed using TT, CAPE or CIN, but the relationship between them is not linear: the $R^2$ between TT and CAPE is below 0.2 and the $R^2$ between CAPE and CIN is below 0.1 for all the stations and seasons, particularly for stable or neutral atmospheres (not shown here). Thus, since the calculation of CAPE and CIN takes into consideration the vertical profile of the atmosphere until the Level of Neutral Buoyancy, they should be considered more reliable than TT, which only takes into account two pressure levels.

Taking in consideration the information presented above, some clear patterns arise from these results. During winter, the areas with unstable air masses are located at the Atlantic coast of the IP, and the instability is larger during the afternoon as CIN is really high in those regions until 12 UTC. However, during summer, the unstable areas are located to the north of the Mediterranean coast. CAPE is even larger during the afternoon in those regions, but some unstable areas can also appear before 12 UTC even if the inhibition is high during that period (CAPE is also high).

## 4 Conclusions

The main purpose of this paper is to evaluate the ability to simulate the atmospheric conditions over the IP with two high-resolution simulations created with the state-of-the-art model WRF. One of these simulations is driven by the boundary conditions provided by ERA-Interim reanalysis (N experiment), while the second one presents the same configuration but including the additional 3DVAR data assimilation step every 6 hours (D experiment). Three parameters were evaluated: TT index, CAPE and CIN. All of them were calculated from the outputs of the model using the publicly available R package *aiRthermo*, also developed by the authors.

In order to validate these parameters, their values were downloaded from the University of Wyoming server for the eight radiosondes available over the IP. Additionally, temperature, mixing ratio and height at all the available pressure levels from the radiosondes were also retrieved. In that case, also from IGRA from NOAA. These variables were used to calculate again these three parameters with *aiRthermo*. Comparing these new values with the ones retrieved directly from the University of Wyoming, the small differences which can only be attributed to different methodologies can be obtained.

The correlation coefficients according to Pearson and Spearman, SD and RMSE show small differences between the different methods used for their computation, which are caused by the use of measured or calculated dew point temperatures in their calculation. However, these differences are more important for CAPE and CIN than for TT because they are highly dependant on the initial conditions for the calculation of the vertical integrations, while TT index is only dependant on two discrete pressure levels. Between both WRF simulations, the most accurate results are produced by the experiment with data assimilation (D), as it is able to correctly capture their temporal evolution as shown by the correlations and RMSEs. The bootstrap analysis with resampling supports this result. Additionally, this is also in agreement with previous studies by the authors where these two simulations were validated against in-situ or gridded observational datasets of integrated water vapour, precipitation, evaporation or wind over coastal areas (Ulazia et al., 2017; González-Rojí et al., 2018). Particularly for integrated water vapour, a

variable closely related to the parameters analyzed in this study, our verification against an independent satellite-based dataset that has not been assimilated (MODIS) showed that this quantity was substantially better simulated across the whole IP for the D simulation (González-Rojí et al., 2018).

The seasonal analysis carried out for TT showed really small differences between methods in every season. Between both WRF experiments, N tends to overestimate the reference variability, while D is able to capture it. In the case of CAPE and CIN, the differences between methods are larger, but not as those within both WRF experiments. However, D is able to produce closer values to the reference than N, only due to the effect of data assimilation as it is the only difference between the two WRF experiments. The experiment including data assimilation is able to produce more realistic virtual temperature profiles during summer and winter, while the experiment without data assimilation produces colder or warmer conditions at the pressure levels in winter or summer respectively, along with warmer lifted air parcels in winter. The seasonal values of the indices showed that the unstable air masses are located mainly over the stations from the Atlantic coast of the IP during winter. All the stations present remarkable values during spring. However, during summer, the most remarkable values are obtained over the Mediterranean coast. Also these stations showed the maximum values during autumn.

All the three indices agree highlighting the heterogeneity of the patterns observed over the IP during winter and summer and at 00 and 12 UTC. The D experiment, which is the most accurate one according to the previous analysis, shows that during winter the unstable areas are found along the entire Atlantic coast, but particularly in the southwestern corner of the IP when the instability is extended towards inland regions. During summer, this feature is reversed, and the regions most prone to unstable air masses are located at the Mediterranean coast and the Ebro basin. The convective inhibition (high values of CIN) is strong at 00 UTC in those regions, but that is highly reduced at 12 UTC. If we restrict the values of CAPE and CIN only to the days with the highest values of CAPE (above the 75th percentile of each grid point), the spatial distribution of CAPE does not change substantially and the values are simply increased. However, in the case of CIN, the largest differences in its distribution appear at night, particularly at the Mediterranean coast during winter and in the southern IP during summer.

If we assume that convection can be triggered orographically or due to the effect of the breezes in those regions (something that is perfectly feasible as the highlighted regions are found in mountainous areas near the coasts), these patterns are in agreement with the precipitation patterns observed in previous studies over the IP (Rodríguez-Puebla et al., 1998; Esteban-Parra et al., 1998; Romero et al., 1999; Iturrioz et al., 2007). The patterns for CAPE observed during winter and summer are similar to those obtained by the regional analysis performed by Viceto et al. (2017). However, their values are comparable to those obtained by our experiment without data assimilation (N). In this case, the data assimilation (D) produces higher values, but much more realistic than the ones from those simulations (N) without it (according to Figure 6).

Finally, no linear relationships were found between the studied parameters as the $R^2$ is always below 0.2, independently of the station and the season. Thus, since the calculation of CAPE and CIN takes into consideration the vertical profile of the atmosphere until the Level of Neutral Buoyancy, these two variables should be considered first to evaluate atmospheric conditions and not TT, as this index only takes into account two pressure levels (500 and 850 hPa).

*Data availability.* These results can be reproduced using the postprocessed outputs from the model available in https://doi.org/10.5281/ zenodo.3611343.

*Author contributions.* The methodology and the software was developed by S.J.G.-R., S.C.-M. and J.S.; The conceptualization, preparation
of datasets and analysis was carried out by all the authors; The original draft of the paper was written by S.J.G.-R., but all the authors took part in the edition and revision of it.

*Competing interests.* The authors declare no conflict of interest.

*Acknowledgements.* S.J.G.R. is now funded by the Oeschger Centre for Climate Change Research (OCCR), but during his PhD he was supported by a FPI Predoctoral Research grant (MINECO, BES-2014-069977). This study was also supported through MINECO project
CGL2016-76561-R from the Spanish Government (MINECO/ERDF, UE) and grant GIU 20/008 from the University of the Basque Country (UPV/EHU). The computational resources were provided by I2BASQUE, and the authors thank the creators of WRF/ARW and WRFDA systems. Authors thank the anonymous reviewers for their comments, which have helped to improve the paper. Finally, most of the calculations were carried out with R (R Core Team, 2018), and the authors want to thank all the authors of the packages used for it.

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
