# Peer review of "Changes in the simulation of atmospheric instability over the Iberian Peninsula due to the use of 3DVAR data assimilation"

_Hydrology and Earth System Sciences, 2020_

## Short Comment (SC1) · 30 Mar 2020

Dear authors,

I find this paper very interesting and well written. Nevertheless, from my point of view clarification about the vertical levels of ERA-Interim used as input to WRF is needed.

Authors write that they used 20 levels without providing further details. In González-Rojí et al, 2018 we find the information that levels range from 5 hPa to 1000 hPa. I conclude from this that authors work with data on pressure levels available from the ECMWF servers.

[Figure]

Counting the available pressure levels between 5 hPa and 1000 hPa from ECMWF, I find 34 levels. When data on model levels are used, 60 levels are available (terrain following and thus with different pressure levels at each grid point).

The calculation of CAPE and CIN is sensitive to the vertical of resolution of the data.

If authors used pressure level data, information about the used levels and why they did not use all available pressure levels is needed. Furthermore, I would like to read why authors did not use model level data, since they provide much more information about the temperature and humidity profiles especially in regions with high topography.

If authors used model level data, I also would like to read information about the used levels and why not the full set of available levels have been used. From my point of view some information provided by the assimilation could already be enclosed when using all available model levels.

A minor issue: as far as I know, there is no quality control for the Wyoming radiosondes. Authors should provide information about their own QC routines. Waiving the high vertical resolution of Wyoming radiosondes (from my point of view not a good idea when analyzing CAPE and CIN), IGRA quality controlled radiosondes could be used.

---

## Referee Comment (RC1) · Anonymous Referee #1 · 28 Apr 2020

General comments:

This manuscript is very well-written and presents the results in a straight-forward manner. It is clear that assimilating various retrievals improves model representation of the three instability variables described. I only have a couple comments outlined below regarding the methodology details

Comments:

Section 2.3.2: This section is very brief on the details of the bootstrapping technique. Perhaps include more details and some references?

[Figure]

What is exactly being verified? Model analyses after assimilation cycling or forecasts? This is not abundantly clear. If these are forecasts being verified, how do the statistics vary with lead time?

---

## Referee Comment (RC2) · Anonymous Referee #2 · 29 Apr 2020

**Review „Changes in the simulation of instability indices over the Iberian Peninsula due to the use of 3DVAR data assimilation "**
**by Santos J. González-Rojí and co-authors submitted to HESS**

In their study, González-Rojí et al. investigate three different convective parameters obtained from two dynamically downscaled WRF model runs over the Iberian Peninsula. Over a 5-year period, the convective parameters from the WRF runs are quantitatively evaluated with sounding data and spatially investigated for different seasons. In addition, the spatial distribution and variability of the convective parameters is investigated and related to certain precipitation characteristics from the literature. The authors found that WRF runs with 4Dvar assimilation best reflect the convective situation.

Overall, the work is well structured and written with a good balance of text and figures. My main concern is that large parts of the paper are rather descriptive in the sense that mainly the figures are described and not interpreted. Reasons for the discrepancies found between the data sets are not given - although that would be most interesting and would increases the scientific value of the paper. In the current version, the benefit of the work for a larger community remains unclear. In the following you find a list of major and minor points as well as some suggestions for editing.

**Major revision points:**

1.) After reading the paper, more questions arise than answers or new scientific insights are given. This is because the paper mainly describes the figures, but does not provide explanations. Questions are: Why do the assimilation runs perform better compared to the simple WRF downscaling? Since the convective parameters considered depend on both temperature gradient and moisture, hat is better reproduced? On which levels/layers? Depending on the location (sounding station) and the season? Why are the differences between the models greater at some stations than at others (depending on the parameter)? What is the relation between CAPE and TT index?

2.) The main conclusion of the paper is that the assimilation run performs better compared to the run without assimilation. But is this not to be expected if soundings are assimilated for which the comparison is made afterwards? What would be the result if you left out some of the soundings for the assimilation and made the comparison for these locations?

3.) Are you sure that ERA-Interim did not originally assimilate the eight soundings you considered? It does not make sense to assimilate any data set twice.

4.) *Either there is a general misunderstanding of convection triggering or the formulations are clumsy. Convective instability and sufficient moisture at lower levels are necessary but not sufficient conditions for the development of convective storm. Convection initiation requires additionally a lifting mechanisms that either reduces CIN or lift a parcel to the level of free convection (LFC). High CAPE/TT values neither trigger convection nor can they directly be related to precipitation as written several times throughout the manuscript.*

5.) CIN works only in conjunction with CAPE. In case of zero CAPE, CIN doesn't matter for convective initiation or development. Analyses of the mean values or the spatial distribution of CIN are useful only when considering days with a certain amount of CAPE (or instability in general).

6.) Using only the nearest grid point to a sounding station neglects the horizontal drift of the radiosoundings. A better choice would be to consider the average value of several grid points.

7.) No reference is made on the original ERA-Interim fields. Thus it is not possible to assess the added value of the downscaled model runs and the need for higher resolutions of the data.

8.) The last section "Conclusions" is only a summary without any (general) conclusions. Tell us what other scientists may learn from your study.

9.) A thorough language check is necessary (e.g., "…observations **in** the stations…" or "obtained **in** stations" or similar formulations used throughout the manuscript are incorrect/weird).

**Minor revision points:**

1. Explain why you have selected CAPE, CIN (note my comment above), and TT and not others, in particular indices that either estimate potential or conditional instability or dynamical properties (deep layer shear, storm-relative helicity; or an index combining thermodynamical and dynamical properties). Is there any cross-correlation between those parameters (CAPE vs. TT)? Also explain why you have only considered a 5-year period, which is far from being representative for the general climate.

2. It's very difficult to compare the different sub-figures due to different axis ranges. I suggest to using the same scaling within one figure.

3. When describing the general convective situation over the IP / over Europe, you should consider also more recent literature.

4. Why have you created your virtual WRF soundings only from one grid point? As correctly stated in the text, the soundings may drift over some distance during the ascent. Using an array of 3 x 3 grid points or so would have been a better choice. Please add a comment on that.

5. L1 (see major point above): Instability does not trigger convection.

6. L2 (also L29-30): CAPE/CIN are measures of the energy and not instability indices.

7. Shorten the abstract and focus on the essentials.

8. L14: "the ingredients for the development of convective precipitation": As alluded to previous, you investigated only the convective environment, thus only one part of the ingredients.

9. L22-23: Do you mean warm fronts? Note that cold fronts especially during summer frequently trigger convective storms by cross-circulations. Thus, classifying precipitation into frontal and convective does not make sense. Convective precip is not triggered by convective instability (see major point 4).

10. L24: "The latter is usually associated with extreme events due to their intensity and short duration". Convection is per se not extreme. And you may add here "high intensity". But the short duration is not the reason why convection may become extreme (or rather related precip and wind).

11. L24-26: The limited skill of NWP models to reliably simulate convective precipitation is not because of their low resolution (note that several European weather services run their models already at 1 km resolution), but partly caused by forecast errors on the synoptic scale, which drive the predictability of convection initiation, and various sources of uncertainty on small scales such as limitations in the assimilated observables or microphysical schemes. There exist a bunch of literature on that.

12. L37: It is impossible to estimate the life cycle or the intensity of convective storms from thermodynamic quantities solely. For organized convective storms, which represent the most intensive storms, you require sufficient vertical wind shear – speed shear (crosswise vorticity) for multicells, and directional shear (streamwise vorticity) for supercells.

13. L41-42: You state that "a (high; include) spatial and temporal resolution is important" for resolving vertical lifting, and thus regional simulations are needed. But in your study, you

investigate only the convective environment and not the mechanisms relevant for convective initiation. So I do not see why you need higher resolved met. fields.

14. L43-44: The reason why convection peaks in the afternoon is related to solar irradiation. This is a fact and not "suggested by previous studies".
15. L44-45: Van Delden used only Synoptic stations with a 6-hourly resolution for their statistics. He found that "most thunderstorms occur at 18 and 24 UTC". 18 means the period from 12 to 18. Thus thunderstorms are most frequent between 12 and 18 UTC! But: It would be better to cite more recent studies based on lightning detections such as, for example, Piper and Kunz, 2017 (Nat. Haz. Earth Syst. Sci.; Fig. 4), Enno et al., 2020 (Atmos. Res.; Fig. 9), or also Lopez et al., (2001), the latter already cited. Not also that Corsica is not the only exception showing a different diurnal convection cycle (e.g., Fig. 4 in Piper and Kunz, 2017).
16. L50: Kaltenböck et al. (2009) investigate the relation between convective environment, lightning data and severe storms reports only for Europe, but not for the USA. So replace the citation or delete this statement.
17. L51-56: The discussion of the convective environment should consider more recent publications based on lightning or high-resolution climate models (e.g., Mohr et al., 2015 (GRL); Sanchez et al., 2017 (Atmos. Res.); Rädler et al., 2018 (JAMC); Enno et al., 2020 (Atmos. Res.)).
18. L57: Explain how the seasonality of precipitation is determined by topography?
19. L64-65: These are not very high values for CAPE. On single days, they can be much higher in the interior of IP (note that according to Fig. 6. monthly mean has a maximum at 1250 J kg$^{-1}$, which implies that at single days much higher values than 1000 J kg$^{-1}$ are reached).
20. L72-73: These are very low values. Other studies (e.g., Kunz, 2007, NHESS; Pucik et al., 2015 (MWR), Taszarek et al., 2017 (MWR)) found much higher CAPE values (also for different version of the mixed-layer CAPE). This should at least be mentioned.
21. Paragraph 57-75: Separate between precipitation and convective environment (CAPE).
22. L76-80: I do not see how climate change is related to this work. I propose to delete this paragraph.
23. L81-86: Please better explain the objectives of the work. Evaluation is not an aim, but a method. Why is the evaluation of the convective field of interest?
24. L110: Give some more details on the levels: spacing, highest level, which ones are used to compute CAPE/CIN.
25. P5, 1$^{st}$ paragraph: This part is a bit out of context in the section "Data and Methods". Consider to move it to the introduction.
26. Section 2.2: Why haven't you considered IGRA sounding data?
27. L135: For readers outside of Europe it would be helpful to include here also local times (approximately).
28. L146: what is meant by "…the analysis increments are stronger at 12 UTC…"? And by "Strong increments are observed during summer…" in L148? Also the relation to the cold-bias in L149 is unclear.
29. L150: "…the effect of the assimilation is not restricted only to the station location". This is a very crucial point. Unfortunately, you did not show that. → see major point 2
30. Sect. 2.3.1: Please explain briefly how you compute the lifting curve from the surface/mixed level to the LCL to the LFC and to the LNB (including quantification of $\theta_e$).
31. L158: Do you have any reference for the statement that soundings "take many minutes to measure the profile of the atmosphere"? The multiplicity of soundings I performed in the past took ~ ½ hour to reach the LNB.
32. L169: What is "an isobaric precooling" and why was it applied?
33. L172: As TT relies on temperature differences, the unit (°C, K) does not matter.

34. L173-174: You should mention that other authors found other values (e.g., Huntrieser et al., 1996: 46 K; Haklander and Van Delden, 2003: 46.7 K; Kunz, 2007: 48.1 K).

35. L174-176: "…not highly dependent on the initial conditions…" correct (but even absolutely independent), but why differ the values you compute by your own from those provided by Wyoming – and only at Murcia (strongest), Santander, Zaragoza, Barcelona? Prevailing inversion layers as stated in L176 cannot be the reason as TT is based on main pressure levels which are always provided by Wyoming. Considering the initial values, you stated that you used the same mixing over the lowest 500 m, thus your method must be identical to that of Wyoming. How have you determined the LCL/LFC?

36. Sect. 2.3.2 / Results: The statistical distribution of CAPE is highly skewed. The product moment correlation coefficient according to Pearson, however, require a normal distribution. A better choice would be the rank correlation coefficient according to Spearman.

37. Sect. 2.3.2, last sentence: please delete the statement about precipitation (cf. major point 4).

38. L188: "by independent locations": Independent in which sense? The locations are not independent right now.

39. Section 3. Results: To facilitate direct comparison of the subfigures in a panel, it would be very helpful if they have the same axis range.

40. L203: Please explain how you selected a model as the "best" model: by the highest correlation coefficient, the lowest rmes, a similar SD, or a combination thereof?

41. L215-216: What is the reason of the small differences between Wyoming and aiRthermo both relying on the same data – in particular for TT which does not involves any assumption about lifting? Why are the differences largest at Murcia?

42. L223: Again, are there any reasons why the two stations of Murcia and Barcelona show the largest differences (rmse) compared to the other stations?

43. L231: "…small differences in initial conditions…"; can you be more specific here (also with regard to TT, as already alluded above)?

44. Figure 3 (CAPE) shows very large differences of the standard deviation between the different models and for some of the stations. Any idea on that?

45. L250: Could you be more specific?

46. L254: "N tends to overestimate the variability in every season and for most of the stations…" Why?

47. L255: "..presents the largest values during winter, which agrees with the fact that the northern and northwestern IP receives greats amount of rain during that season". Is winter rainfall really dominated by convective precipitation? I cannot find any statements in the cited literature. Which of the Atmospheric patterns AP1-19 defined by Romero et al., 1999 are convective patterns? Rodriguez-Puebla et al., 1998, considers only the relation to teleconnection patterns, but did not classify precipitation.

48. L255-260 and Fig. 5: Why does Lisbon show higher TT values in winter than in summer? You may also mention that the differences between the models at Lisbon, La Coruna, and Santander are larger in winter compared to summer (why?).

49. L261-262: Again I miss a reference that shows not only total precipitation, but a classification among the types (stratiform/convective).

50. L268-270: Why does D overestimate CAPE at most stations in spring? And why does the N experiment overestimate CAPE in winter and underestimate it in summer?

51. Figs. 5/6: At Barcelona, Zaragoza, and Murcia, CAPE is highest in summer, whereas TT reaches highest values in spring at these stations. What is the reason of the obvious discrepancy between CAPE and TT?

52. L281 and following: As already mentioned above (see major comment 5), CIN is relevant for convection only in combination with CAPE (An example: imagine a day with zero CAPE and zero CIN; another day with CAPE = 3000 J/kg and CIN = 300 J/kg. None of the days would have the right conditions for deep moist convection to occur. The average of the two days would give CAPE = 1500 J/kg and CIN = 150 J/kg. Fair values for DMC). You could simply fix that by considering CIN only on days for CAPE in excess of 50 or 100 J/kg.
53. L305: "…lowest values are observed near the coastal valleys…" why?
54. Figure 8/9: The spatial distribution of TT and CAPE in most of the cases is contrary, i.e. regions with higher CAPE have lower TT values and vice versa. Any explanation of this apparent contradiction?
55. L326 and following: See major comment 5 and minor 52.
56. L336-345: The relation to "dynamics" does not fit here as the paper solely has a thermodynamical perspective. Be careful with the relation between convective conditions and precipitation.

**Edits:**

1. L5: explain "IP" at first use; the same applies to "SD" in L8
2. L6-7 "measured variables at different pressure levels" may be replaced by "vertical temperature and moisture profiles".
3. L9 methodologies → methods
4. L11: "an air parcel's vertical evolution" → "a lifted air parcel"
5. L13: "…reference values" of which quantity?
6. L15: "**in** the Mediterranean coast" → "**over** the…"; never use in the coast, in the station etc.
7. L16: "The chances of developing thunderstorms in those areas at 12 UTC is much higher than at 00 UTC". This generally applies to convection. Delete.
8. L18: Murcia: Don't mention a location in the abstract that is not well known.
9. L22: "in the planet"?? → "of the planet"
10. L39 "alone and it should be…" → "alone, but should be…"
11. L39: There are much more lifting mechanisms, thus include ", for example, by orography, …"
12. L48: **On** the global scale
13. L50: "convective storms develop for lower values…"
14. L62: "CAPE presents" → "CAPE shows a high…"
15. L72-73: use the plural: hailstorms and thunderstorms
16. L74: "similar to those observed in Europe.." Where in Europe? Is the IP not part of Europe? What is meant by "dispersion of the values"?
17. L100: exercises → experiments
18. P4: please explain all abbreviations at first use
19. L109-110: ".., and they use 51 levels" → "and with 51 levels"
20. L111: form → from
21. LL123: Both simulations, **N and D**, …" (note that you refer to another simulation in the previous section).
22. L136: either delete "..where they only measure it at…" or rewrite this sentence
23. L141: suggestion: "Additionally, vertical profiles of pressure, temperature, and mixing ratio obtained…"
24. L142: delete "indices"
25. L147: "This pattern…" which pattern do you refer to? What is meant by intensity?
26. L154: better use WRF's original eta levels instead of models's to avoid any misunderstanding that you quantified CAPE/CIN only from the 20 ERA-Interim levels.

27. L187: "…of the error/differences due to the different methods applied"
28. L203: be careful with the wording, aiRthermo is not an experiment but simple a quantification.
29. L231: trigger → result in
30. L233: "…between **the** experiments"
31. L255: " Coruna and Santander…"
32. L273: What do you mean by "some stations are more important than others…"?
33. L295-296: "Figure 8 shows…" and "The heterogeneity…" reformulate these two sentences
34. L301: important → higher
35. L317: "…the unstable  **air mass** is found…" An area cannot be unstable
36. L318: For readers outside of the IP, can you give a hint about the location of Tagus and Guadalquivir rivers? And later, L331: the Ebro basin?
37. L320: " **Compared** to what…" Or to which contradiction you refer?
38. L321-322: "where tunderstorms can be developed" → "the area with higher CAPE values…"

---

## Author Response (AR1)

Dear Editor,

Please find enclosed the revised version of the manuscript entitled "Changes in the simulation of instability indices over the Iberian Peninsula due to the use of 3DVAR data assimilation" by S. J. González-Rojí, S. Carreno-Madinabeitia, J. Sáenz and G. Ibarra-Berastegi, that we resubmit to the journal *Hydrology and Earth System Sciences.*

All the major points raised by both reviewers (and by Dr. Klemens Barfus) during the open review have thoroughly been addressed in the current version of the manuscript. Additionally, 6 extra figures were included as annexes in the new version in order to support some new statements made in the manuscript and to clarify some points that were raised by Reviewer#2. Attached to this cover letter you will find the tracked changes version of it, where all the thorough modifications are highlighted.

As explained in the detailed response to the reviewers published online, we consider that we successfully addressed all the points raised by the reviewers and, as such, we hope that the manuscript could be accepted this time.

Yours faithfully,
Santos J. González-Rojí

**Comment made by: Klemens Barfus (klemens.barfus@tu-dresden.de)**

Reply by authors is shown in blue and starts with the symbol >>.

Dear authors,

I find this paper very interesting and well written. Nevertheless, from my point of view clarification about the vertical levels of ERA-Interim used as input to WRF is needed.

>> Thanks for your kind words and we appreciate these insightful comments.

Authors write that they used 20 levels without providing further details. In González-Rojí et al, 2018 we find the information that levels range from 5 hPa to 1000 hPa. I conclude from this that authors work with data on pressure levels available from the ECMWF servers.

Counting the available pressure levels between 5 hPa and 1000 hPa from ECMWF, I find 34 levels. When data on model levels are used, 60 levels are available (terrain following and thus with different pressure levels at each grid point).

>> Yes, we feed the WRF model with **analyses** of temperature, relative humidity, both wind components and geopotential at 20 pressure levels downloaded from the MARS repository. The exact pressure levels are: 5, 10, 20, 30, 50, 70, 100, 150, 200, 250, 300, 400, 500, 600, 700, 800, 850, 900, 925, 950, 1000 hPa.

>> The data in the original model levels from ERA-Interim are represented in spherical harmonics, and we feed the inputs to WRF model in a regular longitude/latitude grid. The reason is that all the variables needed by WRF are obtained only in that regular grid. Only temperature is available in the original model levels from ERA-Interim (see Table 3 from Berrisford et al., 2011). Those are the reasons that we didn't follow the path suggested by Dr. Barfurs.
* * *
Berrisford, P., Dee, D., Poli, P., Brugge, R., Fielding, K., Fuentes, M., Kallberg, P., Kobayashi, S., Uppala, S. and Simmons, A., 2011. The ERA-Interim archive, version 2.0. https://www.ecmwf.int/en/elibrary/8173-era-interim-archive

The calculation of CAPE and CIN is sensitive to the vertical of resolution of the data.

If authors used pressure level data, information about the used levels and why they did not use all available pressure levels is needed. Furthermore, I would like to read why authors did not use model level data, since they provide much more information about the temperature and humidity profiles especially in regions with high topography.

If authors used model level data, I also would like to read information about the used levels and why not the full set of available levels have been used. From my point of view some information provided by the assimilation could already be enclosed when using all available model levels.

>> We know that both CAPE and CIN are sensitive to the number of vertical levels used in their calculation, but concerning ERA-Interim, we have not calculated the values of these indices. Neither in the original model data nor in the downloaded pressure levels. We have not used in our paper the CAPE values available in the forecast stream (not the analyses stream) of ERA-Interim (see Table 8 from Berrisford et al., 2011). Since the objectives of our paper do not include a comparison of WRF versus ERA-Interim, we do not see necessary to add this information.

>> Regarding the indices calculated with the measured data from the radiosondes from Wyoming (aiRthermo in the manuscript), all the available pressure levels in the soundings were used for the calculation of TT, CAPE and CIN.

>> In the case of both WRF simulations, the calculations were done using all the available eta model levels from our configuration. As stated in the manuscript, 51 vertical $\eta$ levels up to 2000 hPa (value at the top of the atmosphere) are available.

>> All this information about the calculation of the indices with the pressure levels available for each option (aiRthermo and both WRF simulations) is already stated in the subsection 2.3.1. However, in the next version of the manuscript, all this information will be stated in a clearer way.
* * *
Berrisford, P., Dee, D., Poli, P., Brugge, R., Fielding, K., Fuentes, M., Kallberg, P., Kobayashi, S., Uppala, S. and Simmons, A., 2011. The ERA-Interim archive, version 2.0. https://www.ecmwf.int/en/elibrary/8173-era-interim-archive

A minor issue: as far as I know, there is no quality control for the Wyoming radiosondes. Authors should provide information about their own QC routines. Waiving the high vertical resolution of Wyoming radiosondes (from my point of view not a good idea when analyzing CAPE and CIN), IGRA quality controlled radiosondes could be used.

>> The radiosondes from Wyoming University were already used for the validation of the precipitable water from our both experiments in González-Rojí et al. 2018, and none of the values were taken as erroneous. For the continuity of the study related with both simulations, we decided to use the same radiosondes for the calculation of the indices. Additionally, since the values of CAPE are very sensitive to methodological factors, such as the computation of the initial parcel or the vertical spacing in pressure levels, we have estimated CAPE and CIN from the values of temperature and relative humidity at pressure levels in IGRA soundings using the same methodology that we have used in our paper (see Section 2.3.1).

>> In order to validate the CAPE, CIN and TT indices calculated by Wyoming University, they were compared with the ones calculated by IGRA as suggested by Dr. Barfurs. The comparison shows that the results are not sensitive to the selection of the dataset, (see enclosed Taylor diagrams). We expected these results, since the use of homogenous data is particularly important for long-term trends, and we are simply analyzing five years of data.

>> Figure 1 shows by means of Taylor diagrams the comparison of previous results included in the paper against the TT index calculated from the IGRA soundings (all levels), which will be chosen as the reference. It is clearly seen that the closest points are always without almost no exception the green ones (values of TT reported by the Wyoming archive), and the yellow ones (those computed by our package aiRthermo from the pressure levels from Wyoming archives) with the only exception of Murcia. The next best determination of the TT index is achieved by the D experiment (WRF run including data assimilation, blue points) and, finally, the red points corresponding to the N simulation (WRF run without data assimilation) shows the worst agreement in all cases. This implies that the use of IGRA data as the reference instead of the Wyoming soundings does not change the conclusions of our study (even the exception for Murcia was observed in our previous results). It also shows that the error due to homogeneity in such a short interval of time (five years) is very small (at least, in this observational record).

>> Figure 2 shows a similar result for CAPE. The estimation of CAPE from Wyoming and the one estimated with aiRthermo using pressure levels from Wyoming soundings are always quite close to the value estimated from IGRA soundings (RMSE smaller than 20 J/kg in all cases). As in the case of TT, the main results of

our paper are not affected by switching the reference dataset from Wyoming to IGRA: the D experiment shows better agreement with observed CAPE than the N experiment.

>> Figure 3 shows the same result for CIN. The RMSE of CIN values computed from Wyoming soundings and IGRA soundings is also small (smaller than 20 J/kg). With the only exception of Gibraltar, in which N and D behave exactly the same (this was also observed in the paper), D always produces better results than the N simulation.

>> Thus, we find that our results are robust to the selection of the observational dataset. However, the figures that we show here will be incorporated to the final version of the paper, since we feel this comment leads to a better paper.
* * *
González-Rojí, S.J., Sáenz, J., Ibarra-Berastegi, G. and Díaz de Argandoña, J., 2018. Moisture balance over the Iberian Peninsula according to a regional climate model: The impact of 3DVAR data assimilation. *Journal of Geophysical Research: Atmospheres*, *123*(2), pp.708-729.

[Figure]

Fig.1 Taylor diagrams of TT index if the values from IGRA are taken as reference.

[Figure]

Fig. 2. Taylor diagrams of CAPE if the values from IGRA are taken as reference.

[Figure]

Fig.3. Taylor diagrams of CIN if the values from IGRA are taken as reference.

**Comment made by: Anonymous Referee #1**

Reply by authors is shown in blue and start with the symbol >>.

General comments:
This manuscript is very well-written and presents the results in a straight-forward manner. It is clear that assimilating various retrievals improves model representation of the three instability variables described. I only have a couple comments outlined below regarding the methodology details.

>> Thank you for these supportive words.

Comments:
Section 2.3.2: This section is very brief on the details of the bootstrapping technique. Perhaps include more details and some references?

>> The bootstrap operates by constructing the artificial data batches using sampling with replacement from the original data. Conceptually, the sampling process is equivalent to writing each of the N data values on separate slips of paper and putting all N slips of paper in a hat. To construct one bootstrap sample, N slips of paper are drawn from the hat and their data values recorded, but each slip is put back in the hat and mixed (this is the meaning of "with replacement") before the next slip is drawn. This process is repeated a large number of times yielding, for example, to 1000 samples of size N that are slightly perturbed versions of the original data set. The 95% confidence intervals for the different statistical indicators can then be derived from their P975 and P025 observed percentiles of their distribution as obtained from the 1000 perturbed series. More mathematical details can be found in Wilks (2011), Efron & Gong (1983) and Downton & Katz (1993).

>> Coming back to our manuscript, the bootstrap technique was applied to the temporal analysis of each index. In our case, the original time series used in the Taylor diagrams consist of 60 values, each of them for the corresponding month along the period 2010-2014 (12 months x 5 years). For the bootstrap, we created 1000 perturbed time series taking into account different samples of the data. 67% of the new time series (2/3 of the length of the original time series - 40 values in our case) is made from the original data, and the remaining 33% (1/3 - 20 values) is chosen from those values already taken from the original data. For each correlation calculated, the same samples are taken from the observed and model data.

>> In order to clarify how the bootstrap is performed in our analysis, some extra lines will be added to section 2.3.2 (Analysis) of the paper. This new lines will sum up the information presented above, and they will include the citations.
* * *
>> Wilks, D. S. (2011). *Statistical methods in the atmospheric sciences* (Vol. 100). Academic press.

>> Efron, B., & Gong, G. (1983). A leisurely look at the bootstrap, the jackknife, and cross-validation. The American Statistician, 37(1), 36-48.
DOI: 10.1080/00031305.1983.10483087

>> Downton, M. W., & Katz, R. W. (1993). A test for inhomogeneous variance in time-averaged temperature data. Journal of climate, 6(12), 2448-2464.
DOI: 10.1175/1520-0442(1993)006<2448:ATFIVI>2.0.CO;2

What is exactly being verified? Model analyses after assimilation cycling or forecasts? This is not abundantly clear. If these are forecasts being verified, how do the statistics vary with lead time?

>> The structure of segments used in each experiment is presented in Figure 1. The N experiment is generated running 6-hour long segments that are restarted from the restart file produced at the end of previous segment (top panel of Figure 1). This is similar to a continuous WRF run where the boundary conditions (in our case, from ERA-Interim) are provided to the model every 6-hours after the initialization of the model the 1st of January, 2009.

>> For the experiment including the data assimilation, the structure is a little bit more complex. In these case, 12-hour long segments starting at every analysis time (00, 06, 12 and 18 UTC) are used (bottom panel of Figure 1). The analysis are generated from the outputs of the model at a 6-hour forecast step from the previous segment as first guess in a 3DVAR data assimilation scheme. The data assimilation is performed using the observations in PREPBUFR format obtained from the NCEP ADP Global Upper Air and Surface Weather Observations (ds337:0) dataset generated by NOAA. Only those observations included in a 2-hour time-window centered at the analysis times were included.

>> In both cases, the outputs are saved every 3 hours, which means that analysis (00, 06, 12 and 18 UTC) and 3-hour forecasts (at 03,09, 15 and 21 UTC) are included in our results. These recording frequency is highlighted with magenta ellipses in Figure 1. These are the data that are verified in the manuscript.

>> The new version of the manuscript will include an expanded explanation about how both simulations were created in order to make it clearer to the readers.

[Figure]

Figure 1: Diagram showing the structure of the segments used for running N (top, in red) and D (bottom, in blue) experiments with the WRF model. The outputs recorded are highlighted with magenta ellipses, and the restart files used to run the segments are shown in orange.

**Comment made by: Anonymous Referee #2**

Reply by authors is shown in blue, and starts with the symbol >>.

In their study, González-Rojí et al. investigate three different convective parameters obtained from two dynamically downscaled WRF model runs over the Iberian Peninsula. Over a 5-year period, the convective parameters from the WRF runs are quantitatively evaluated with sounding data and spatially investigated for different seasons. In addition, the spatial distribution and variability of the convective parameters is investigated and related to certain precipitation characteristics from the literature. The authors found that WRF runs with 4Dvar assimilation best reflect the convective situation.

>> We point the reviewer that we have used 3DVAR data assimilation.

Overall, the work is well structured and written with a good balance of text and figures. My main concern is that large parts of the paper are rather descriptive in the sense that mainly the figures are described and not interpreted. Reasons for the discrepancies found between the data sets are not given - although that would be most interesting and would increases the scientific value of the paper. In the current version, the benefit of the work for a larger community remains unclear. In the following you find a list of major and minor points as well as some suggestions for editing.

>> Thanks for your comments.

**Major revision points:**

1.) After reading the paper, more questions arise than answers or new scientific insights are given. This is because the paper mainly describes the figures, but does not provide explanations. Questions are: Why do the assimilation runs perform better compared to the simple WRF downscaling? Since the convective parameters considered depend on both temperature gradient and moisture, hat is better reproduced? On which levels/layers? Depending on the location (sounding station) and the season? Why are the differences between the models greater at some stations than at others (depending on the parameter)? What is the relation between CAPE and TT index?

>> The simulation including data assimilation produces more reliable results than the one without it. This conclusion is extracted from the paper after the analysis of the convective indices studied, after comparing the results from both WRF experiments against the ones obtained from Wyoming University (also against IGRA radiosondes as shown in one of the comments posted in the website).

>> The differences between WRF experiments are due to the effect of data assimilation in the the vertical profiles of temperature and mixing ratio. The effect of the assimilation is not restricted to the surface, and it is propagated towards the top of the atmosphere and the nearby grid points due to the optimization of the cost function (Barker et al., 2004, 2012). Additionally, as presented in previous studies (already cited in the manuscript in sections 2.1 and 2.2), the effect is also observed in the soil moisture and both surface temperature and moisture. As presented in González-Rojí et al.

(2018), data assimilation is important at 12 UTC for moisture, and at 00 and 12 UTC for temperature, and their effects are important in the southeastern IP and both Guadalquivir and Ebro basins (see their Figure 13). This pattern is consistent along the seasons, but its intensity varies seasonally (stronger during summer than in winter). As presented in González-Rojí et al. (2020), the soil moisture content is also different in both simulations as a result of the data assimilation (this variable is not assimilated, and data assimilation is the only difference in the configuration of the model).

>> The main objective of our paper is neither to find a relation between the studied convective indices over the IP nor their performance as predictors of heavy rainfall events. We only want to evaluate how well the values of each index are simulated by comparing the results from two different configurations of the model to observational data, and to study the differences in the seasonal patterns due to the use of a data assimilation step in the numerical downscaling phase. There are not many studies analyzing this currently.

>> ------------------------------------------------------------------------

>> González-Rojí, S. J., Sáenz, J., Ibarra-Berastegi, G., & Díaz de Argandoña, J. (2018). Moisture balance over the Iberian Peninsula according to a regional climate model: The impact of 3DVAR data assimilation. *Journal of Geophysical Research: Atmospheres*, *123*(2), 708-729.

>> González-Rojí, S. J., Sáenz, J., Díaz de Argandoña, J., & Ibarra-Berastegi, G. (2020). Moisture Recycling over the Iberian Peninsula: The Impact of 3DVAR Data Assimilation. *Atmosphere*, *11*(1), 19.

>> Barker, D. M., Huang, W., Guo, Y. R., Bourgeois, A. J., & Xiao, Q. N. (2004). A three-dimensional variational data assimilation system for MM5: Implementation and initial results. *Monthly Weather Review*, *132*(4), 897-914.

>> Barker, D., Huang, X. Y., Liu, Z., Auligné, T., Zhang, X., Rugg, S., ... & Demirtas, M. (2012). The weather research and forecasting model's community variational/ensemble data assimilation system: WRFDA. *Bulletin of the American Meteorological Society*, *93*(6), 831-843.

2.) The main conclusion of the paper is that the assimilation run performs better compared to the run without assimilation. But is this not to be expected if soundings are assimilated for which the comparison is made afterwards? What would be the result if you left out some of the soundings for the assimilation and made the comparison for these locations?

>> The paper supports the idea that the experiment including data assimilation performs better than the one without, similar conclusion to what we have observed for other variables in previous studies by the authors. However, in this case, the main conclusion of the paper is that important differences arise in those patterns only due to data assimilation. The impact of data assimilation is not limited to the grid cells close to the location of the soundings. As shown in the Figures of our paper, the changes extend over large areas of the Iberian Peninsula despite the limited coverage by soundings.

>> It is true that the comparison against assimilated soundings can be biased, but we can not discard observations when preparing the simulations without performing a damage to the study that we want to perform. On the other side, as we mentioned before, we are analyzing derived variables not directly assimilated on a regional domain covering places with no observation at all. We are mainly comparing the values of different convective indices after different calculation methods (as the

method followed by Wyoming and our method included in the package aiRthermo). Additionally, as an extra way of validating our results, we always compared the values obtained in the patterns over the entire IP with previous studies focusing in the region (or at least covering it even if it is with low resolution data).

3.) Are you sure that ERA-Interim did not originally assimilate the eight soundings you considered? It does not make sense to assimilate any data set twice.

>> We did not check every cycle of six hours all the data assimilated by ERA-Interim, as we think it is pointless. We actually assume that some of these radiosondes have very likely already been assimilated in ERA Interim reanalysis. However, that is not a problem for our simulations with WRF as we only used the data from ERA-Interim as boundary conditions for our regional model after the initial run. Since the run which used ERA Interim for initial conditions (January 1st, 2009) corresponded to one year before the period that we started analyzing the output (January 1st, 2010), we can be sure that the interior of the domain is reflecting the variability corresponding to the regional climate model.

>> Moreover, the effect of assimilating one station in ERA-Interim, which has a resolution of around 80 km, cannot be comparable to the effect of assimilating a station in a domain with 15 km. Besides that, the original objective of our paper was to compare the quality of WRF simulations and ERA Interim is only used to provide initial and boundary conditions to WRF.

4.) Either there is a general misunderstanding of convection triggering or the formulations are clumsy. Convective instability and sufficient moisture at lower levels are necessary but not sufficient conditions for the development of convective storm. Convection initiation requires additionally a lifting mechanisms that either reduces CIN or lift a parcel to the level of free convection (LFC). High CAPE/TT values neither trigger convection nor can they directly be related to precipitation as written several times throughout the manuscript.

>> To some extent, we agree with the reviewer. Convective instability and moisture in low levels of the atmosphere are ingredients necessary to trigger convective storms, and consequently, convective precipitation. However, the final ingredient, which is the lifting, is provided by the instability, forced by orography, the convergence of horizontal moisture fluxes or the breezes in coastal regions. All this information is included already in the second paragraph of the introduction of our paper, so we agree with the reviewer on that.

>> In order to avoid misleading ideas by the readers, we have carefully rewritten all the sentences highlighted by the reviewer in the new version of the manuscript.

5.) CIN works only in conjunction with CAPE. In case of zero CAPE, CIN doesn't matter for convective initiation or development. Analyses of the mean values or the spatial distribution of CIN are useful only when considering days with a certain amount of CAPE (or instability in general).

>> We agree to some extent with the reviewer on that. However, the objective of this paper is not to evaluate CAPE and CIN only for extreme events as tools to predict extreme convective rainfall. The objective of this paper is to evaluate the ability of WRF simulations (including or not the 3DVAR data assimilation step) to produce reliables values of TT, CAPE and CIN over the Iberian Peninsula, irrespective of whether they produce or not rainfall events.

>> As stated already at the end of the Introduction, "the main objective of this paper is to evaluate the performance of two simulations created by using the WRF model at reproducing the atmospheric conditions that can trigger convective precipitation over the IP. To do so, the comparison of pseudo-soundings extracted from the model against real observations will be carried out." At the very end, what we are doing in the paper is to evaluate the Probability Density Functions (PDFs) of the three instability indices obtained in each experiment against the reference values measured by the University of Wyoming (also IGRA in the future version), but not only during extreme events.

>> This clarification was added to the new version of the manuscript.

6.) Using only the nearest grid point to a sounding station neglects the horizontal drift of the radiosoundings. A better choice would be to consider the average value of several grid points.

>> That is true to some extent. We agree that considering the nearest grid point for the comparison against a sounding is not always the best option. However, this depends on the spatial resolution of the domain of the simulations. Averaging several points can be a good idea when convection-permitting scales are used (below 5-3km), but not when the spatial resolution of the experiments is 15 km (as in our case). If we consider the average of the nearest grid points, we would be taking into account an area of 2025 km2 (45km x 45km), and that is too much for a comparison against station data.

>> Additionally, according to recent studies (Xu et al., 2015), most of the vertical levels up to 6 km are already measured for a drifting distance of 7.5 km, independently of a clear or cloudy day (see their Figure 6). As also mentioned by the reviewer, both convective instability and sufficient moisture at lower levels are necessary for developing a convective storm, and these lower levels are already measured below 6km. Thus, taking into account our spatial resolution, we stand by our decision to use the nearest point to the station for comparison against station data.

>> ---------------------------------------------------------------------------------------------------------

>> Xu, G., Xi, B., Zhang, W., Cui, C., Dong, X., Liu, Y., and Yan, G. (2015), Comparison of atmospheric profiles between microwave radiometer retrievals and radiosonde soundings, *J. Geophys. Res. Atmos.*, 120, 10,313– 10,323, doi:10.1002/2015JD023438.

7.) No reference is made on the original ERA-Interim fields. Thus it is not possible to assess the added value of the downscaled model runs and the need for higher resolutions of the data.

>> The information about the original ERA-Interim fields was also asked by the comment published by a reader. As stated in his reply available online, we used 20 pressure levels downloaded from the MARS repository to feed the WRF model, which are: 5, 10, 20, 30, 50, 70, 100, 150, 200, 250, 300, 400, 500, 600, 700, 800, 900, 925, 950 and 1000 hPa. Our set-up, as stated in the manuscript, uses 51 vertical levels, so there is a relevant increase in the number of vertical levels compared to the data from ERA-Interim. Additionally, the spatial resolution of ERA-Interim is around 80 km, and our domain has 15 km resolution. Thus, we also improve the spatial resolution of the data.

>> Taking into account this information already stated in the manuscript, we sincerely consider our simulations provide extra information to the one present in the Reanalysis. Additionally, these two

experiments have been already validated against observational datasets (both for stations and grids) in previous studies by the authors, and in some cases, particularly for the experiment including data assimilation, they are able to outperform the driving reanalysis ERA-Interim. All these studies are cited at the end of section 2.1. We have not performed any quantitative analysis of the added value of these simulations since, as we have already stated before, we do not compare the performance of the WRF runs with the original data (see Figures 2 to 8 of the original manuscript, for instance). We are interested in comparing the performance of a run using 3DVAR with a different one which does not use it.

8.) The last section "Conclusions" is only a summary without any (general) conclusions. Tell us what other scientists may learn from your study.

>> We do not agree with the reviewer. It includes all the important information extracted from the analysis performed, and it includes details about the comparison of both experiments regarding the indices TT, CAPE and CIN, not only in the location of the radiosondes but also for the entire IP.

9.) A thorough language check is necessary (e.g., "…observations **in** the stations…" or "obtained **in** stations" or similar formulations used throughout the manuscript are incorrect/weird).

>> A detailed revision and edition of the language has been carried out in the new version of the manuscript.

**Minor revision points:**

1. Explain why you have selected CAPE, CIN (note my comment above), and TT and not others, in particular indices that either estimate potential or conditional instability or dynamical properties (deep layer shear, storm-relative helicity; or an index combining thermodynamical and dynamical properties). Is there any cross-correlation between those parameters (CAPE vs. TT)? Also explain why you have only considered a 5-year period, which is far from being representative for the general climate.

>> As stated in the manuscript, we considered some of the most commonly used convective indices, which can give information about the regions where more unstable conditions are met over the IP. This is not something weird or new, and that is why several studies focus only on some of these indices, or only even in one of them. Some examples of these papers, and particularly focusing in the IP, can be found in the Introduction of the manuscript.

>> About the length of our simulations, in any case we say in the manuscript that we want to show a climatology of these indices, as we also agree that it would be impossible only with 5 years. As presented in the paper, we only want to evaluate the differences triggered by the use of data assimilation in those patterns for a limited period of time. Since the same period of time is used for both simulations, and since the same model, parameterizations and boundary conditions are used for both runs, the differences identified must be clearly assigned to the use of 3DVAR data assimilation. The period of time is shorter than the estimated 30 years needed to robustly resolve the climatology, but it is long enough (five years is not a week) to draw robust conclusions across the behaviour in different seasons of the year, for instance.

2. It's very difficult to compare the different sub-figures due to different axis ranges. I suggest to using the same scaling within one figure.

>> We agree on this comment with the reviewer because having the same scaling in the figures is easier to interpret the results, particularly for the intercomparison of results. However, that is not possible in our case because the values show a really large range. Here are some examples concerning each of the figures included in the manuscript:

>> 1) Figures 3 and 4 (Taylor diagrams for CAPE and CIN): A Coruna presents standard deviations of around 25 J/kg for CAPE, but Barcelona presents values around 250 J/kg. If we set the axis to the maximum, the results from many stations will not be recognizable. The same happens with CIN, as Gibraltar and Murcia present values around 100 J/kg, and A Coruna around 18 J/kg.

>> 2) Figure 5 (Box and Whiskers for TT) already has the same axis range.

>> 3) Figures 6 and 7 (Box & Whiskers for CAPE and CIN): most of the values for CAPE in winter are below 75 J/kg (and some stations show values around 0 J/kg), but in summer some stations reach the 750 J/kg. Same happens to CIN, in which all the stations obtained values below 20 J/kg in winter, and below 400 J/kg in summer.

>> 4) Figures 9 and 10 (patterns for CAPE and CIN): Same thing as before happens. For CAPE, the values in winter are below 50 J/kg, but in summer are below 600 J/kg. For CIN, the values in winter are always below 10 J/kg, and in summer below 250 J/kg. However, in these Figures, the same axis range has been selected for each season, independently of the time of the day in order to clarify the results.

>> In order to set the same range of values for the TT index, which results do not vary as much as for CAPE or CIN, Figures 2 and 8 will be created again. However, the other figures remained the same in the new version of the manuscript as we truly believe that setting the same axis for all the plots in each Figure will complicate the visualization and interpretation of the results.

3. When describing the general convective situation over the IP / over Europe, you should consider also more recent literature.

>> Some of the most recent papers focusing ONLY the IP were presented in the introduction, and that is why even the ones focusing in future scenarios were commented in there.

4. Why have you created your virtual WRF soundings only from one grid point? As correctly stated in the text, the soundings may drift over some distance during the ascent. Using an array of 3 x 3 grid points or so would have been a better choice. Please add a comment on that.

>> As stated in the mayor comment number 6 of the reviewer, that would be necessary if convection-permitting scales were used in the simulation. However, the spatial resolution that we used is 15 km resolution, and most of the levels measured by a balloon are already measured when the drifting distance is below 7.5 km (the height of the balloon is around 6 km) independently of the conditions in the sky. This distance is less than the distance covered by our grids, and increasing the grid to the nearest points (15km x3 grids) will not be a good option to evaluate the performance of the model.

5. L1 (see major point above): Instability does not trigger convection.

>> We have edited that sentence as suggested by the reviewer.

6. L2 (also L29-30): CAPE/CIN are measures of the energy and not instability indices.

>> True. CAPE and CIN represent the Convective Available Potential Energy and the Convective Inhibition energies in a column of the atmosphere, but they are related to atmospheric instability. See, for instance, Tsonis (page 155), in which CAPE and CIN are discussed as problems in Chapter 8, entitled "Vertical stability of the atmosphere". In Djuric (1994), Chapter 5 is entitled "Analysis of vertical soundings" and section 5-6 is entitled "Instability Indices" and Section 5-8 is entitled "Integrated indicators if instability". These sections describe the indices we present in our paper. Bohren and Albrecht (1998) write (we quote): "A sounding (not just a layer) is often said to be conditionally unstable or in the conditional state if a parcel from any level of the sounding has positive CAPE" in page 317.

>> --------------------------------------------------------------------

>> A. Tsonis, (2007), "An Introduction to Atmospheric Thermodynamics", 2nd Ed., Cambridge University Press.

>> D. Djuric (1994), "Weather Analysis", Prentice Hall

>> C. F. Bohren and B. A. Albrecht (1998) "Atmospheric Thermodynamics", Oxford University Press.

7. Shorten the abstract and focus on the essentials.

>> The abstract was shortened according to the suggestion by the reviewer.

8. L14: "the ingredients for the development of convective precipitation": As alluded to previous, you investigated only the convective environment, thus only one part of the ingredients.

>> The sentence was edited to incorporate what the reviewer said.

9. L22-23: Do you mean warm fronts? Note that cold fronts especially during summer frequently trigger convective storms by cross-circulations. Thus, classifying precipitation into frontal and convective does not make sense. Convective precip is not triggered by convective instability (see major point 4).

>> As stated already in those lines, we follow the definition by the WRF model of considering precipitation separated in two components: large-scale and convective precipitation. We state explicitly the frontal systems only as an example of that kind of precipitation, but that does not mean that we only consider that in that category. In order to clarify that, we have rewritten those lines.

10. L24: "The latter is usually associated with extreme events due to their intensity and short duration". Convection is per se not extreme. And you may add here "high intensity". But the short duration is not the reason why convection may become extreme (or rather related precip and wind).

>> We agree with the reviewer that convection is per se not extreme, and that is why we have already stated that usually convective precipitation can end up in a extreme event.

11. L24-26: The limited skill of NWP models to reliably simulate convective precipitation is not because of their low resolution (note that several European weather services run their models

already at 1 km resolution), but partly caused by forecast errors on the synoptic scale, which drive the predictability of convection initiation, and various sources of uncertainty on small scales such as limitations in the assimilated observables or microphysical schemes. There exist a bunch of literature on that.

>> We have added these other limitations to the text as suggested by the reviewer.

12. L37: It is impossible to estimate the life cycle or the intensity of convective storms from thermodynamic quantities solely. For organized convective storms, which represent the most intensive storms, you require sufficient vertical wind shear – speed shear (crosswise vorticity) for multicells, and directional shear (streamwise vorticity) for supercells.

>> That line highlights the ability of both CAPE and CIN to provide some information about the potential development and intensity of convective precipitation. As stated later in the same paragraph, we stated that another extra ingredient is needed to trigger it, such as the lifting.

13. L41-42: You state that "a (high; include) spatial and temporal resolution is important" for resolving vertical lifting, and thus regional simulations are needed. But in your study, you investigate only the convective environment and not the mechanisms relevant for convective initiation. So I do not see why you need higher resolved met. fields.

>> That line is not related only to the fact that high resolution is needed for resolving vertical lifting (that is obvious). As stated there already, high resolution data is needed to carry out similar studies to those presented in the paragraph, which evaluate different convective indices (as our study).

>> Additionally, the spatial resolution is important because in order to calculate variables as CAPE and CIN, these are more reliable when the resolution is finer. Particularly when you want to validate those results against radiosonde data or in areas of complex topography. As already mentioned, it does not make sense to calculate these variables with the mean values over huge areas or several grid-points. As shown in our paper (Figures 8 to 10), the spatial variability of TT, CAPE and CIN strongly resemble the features of terrain.

14. L43-44: The reason why convection peaks in the afternoon is related to solar irradiation. This is a fact and not "suggested by previous studies".

>> We agree on that, and that is what is expected. However, as stated later in the same paragraph, we show that some regions show those peaks in the morning. Thus, as it is not true everywhere, we used "suggested" to introduce that feature. We will change it to "backed-up" in the new version of the manuscript.

15. L44-45: Van Delden used only Synoptic stations with a 6-hourly resolution for their statistics. He found that "most thunderstorms occur at 18 and 24 UTC". 18 means the period from 12 to 18. Thus thunderstorms are most frequent between 12 and 18 UTC! But: It would be better to cite more recent studies based on lightning detections such as, for example, Piper and Kunz, 2017 (Nat. Haz. Earth Syst. Sci.; Fig. 4), Enno et al., 2020 (Atmos. Res.; Fig. 9), or also Lopez et al., (2001), the latter already cited. Not also that Corsica is not the only exception showing a different diurnal convection cycle (e.g., Fig. 4 in Piper and Kunz, 2017).

>> Some of the papers highlighted by the reviewer were added to the new manuscript, and the lines addressing the foundings by van Delden were adapted.

16. L50: Kaltenböck et al. (2009) investigate the relation between convective environment, lightning data and severe storms reports only for Europe, but not for the USA. So replace the citation or delete this statement.

>> That is true, Kaltenböck et al. (2009) only focuses on Europe. However, when he focuses on CAPE (in section 3.4 of his paper), he does the next statement: "Reasons could be the small sample or an underreporting of F2 and F3 events, the synchronous occurrence of different severe events (e.g. tornado accompanied by hail) and standard values of CAPE and SRH, which seem to be lower for Europe than in the US." And that is why that paper is cited in that line.

17. L51-56: The discussion of the convective environment should consider more recent publications based on lightning or high-resolution climate models (e.g., Mohr et al., 2015 (GRL); Sanchez et al., 2017 (Atmos. Res.); Rädler et al., 2018 (JAMC); Enno et al., 2020 (Atmos. Res.)).

>> As already stated in the reply to comment 15, some of the papers highlighted by the reviewer were added to the new manuscript.

18. L57: Explain how the seasonality of precipitation is determined by topography?

>> The reviewer is right. Precipitation is not determined by topography. In the new version of the manuscript this line was modified in order to highlight the fact that the seasonal precipitation patterns are affected by several factors, including: Different sources of moisture due to seasonal variations of the global atmospheric circulation  and contrasting climatic regions (influenced by the strong topography of the Iberian Peninsula).

19. L64-65: These are not very high values for CAPE. On single days, they can be much higher in the interior of IP (note that according to Fig. 6. monthly mean has a maximum at 1250 J kg-1, which implies that at single days much higher values than 1000 J kg-1 are reached).

>> In that paragraph we are not evaluating if the values are high or not. We are just presenting to the reader the mean values of CAPE obtained over the Iberian Peninsula for a season, and the differences between the north and the south. We have rewritten that sentence in order to clarify it.

20. L72-73: These are very low values. Other studies (e.g., Kunz, 2007, NHESS; Pucik et al., 2015 (MWR), Taszarek et al., 2017 (MWR)) found much higher CAPE values (also for different version of the mixed-layer CAPE). This should at least be mentioned.

>> As we said in the previous comment, we only present the mean values obtained by previous studies in those lines. We have also included in the new version some of the other values included in the papers suggested by the reviewer.

21. Paragraph 57-75: Separate between precipitation and convective environment (CAPE).

>> Those lines were separated in two different paragraphs in the new version of the manuscript.

22. L76-80: I do not see how climate change is related to this work. I propose to delete this paragraph.

>> We wanted to show that indices like CAPE have been also investigated under future climate change scenarios. That is why we added that paragraph. We think that it is important to show that, so we have reduce it and merge it with previous paragraph instead of completely deleting it as suggested by the reviewer.

23. L81-86: Please better explain the objectives of the work. Evaluation is not an aim, but a method. Why is the evaluation of the convective field of interest?

>> We do not agree on that with the reviewer. We want to evaluate the performance of both WRF downscaling experiments at simulating some of the commonly used convective indices, and observe the differences that arise due to the use of data assimilation in one of these experiments. Taking into account all the information given in the introduction, it is clear that this topic is quite important to evaluate the regions more prone to develop unstable thermodynamic conditions that can end up in convective precipitation. Additionally, the importance of this topic is backed-up by all the papers that can be found in the literature, and particularly in our paper, by all the mentioned studies focusing over the Iberian Peninsula. Additionally, we want to stress that most of the WRF runs being currently run do not use the 3DVAR assimilation step, and we think that showing that it allows a better estimation of CAPE or CIN is an important contribution to the literature.

24. L110: Give some more details on the levels: spacing, highest level, which ones are used to compute CAPE/CIN.

>> This is something already asked by one of the readers of the paper, who posted a comment during the open discussion. As we stated in our reply to him, 51 vertical levels are available in our WRF experiments, and they go up to 20 hPa. In WRF, these vertical levels are in $\eta$ coordinates, so they follow the terrain of the domain. Thus, the spacing between them is not constant. Explicitly, these are the values:

>> 0.9965, 0.988, 0.9765, 0.962, 0.944, 0.9215, 0.8945, 0.8649009, 0.8347028, 0.8045048, 0.7743067, 0.7316024, 0.6780097, 0.6275734, 0.5801385, 0.5355568, 0.4936861, 0.4543901, 0.4175383, 0.3830059, 0.350673, 0.3204254, 0.2921534, 0.2657521, 0.2411216, 0.218166, 0.1967937, 0.1769174, 0.1584536, 0.141405, 0.1258691, 0.1118248, 0.09912901, 0.0876521, 0.07727711, 0.06789823, 0.05941983, 0.05175545, 0.04482694, 0.03856365, 0.0329017, 0.02778335, 0.02315643, 0.01897375, 0.01519264, 0.01177457, 0.00868467, 0.005891433, 0.003366379, 0.001083758.

>> For the calculation of the indices in both WRF simulations, all the available pressure levels were used. As we replied to the reader, all this information will be clarified in the corresponding sections of the paper: 2.1 and 2.3.1.

25. P5, 1st paragraph: This part is a bit out of context in the section "Data and Methods". Consider to move it to the introduction.

>> We do not agree with the reviewer on that. All the information given in this paragraph is related to the previous analyses and validations of the simulations that are going to be used in the study,

that is, experiments N and D in the paper. That is why this paragraph is presented here after the short introduction of both experiments.

>> We believe that if we move this paragraph to the Introduction, it will be completely out of context as the Introduction is mainly focusing on previous studies about the topic of the paper, that is, instability indices.

26. Section 2.2: Why haven't you considered IGRA sounding data?

>> This is something also suggested by one of the readers of the paper, who posted a comment during the open discussion. As presented in our reply to him, we used the data from the University of Wyoming because we used already the data in González-Rojí et al. 2018 for the validation of the precipitable water, and in that case, none of the values were taken as erroneous. Additionally, we wanted to keep a consistency between all the studies carried out with both WRF simulations, and that is the reason why the same radiosondes were used for the calculation of the instability indices in this paper.

>> In the reply to the reader, we also validated the indices calculated by Wyoming against the ones calculated by IGRA. As CAPE and CIN are very sensitive to methodological factors such as the computation of the initial parcel or vertical spacing in pressure levels, we estimated them from the values of temperature and relative humidity at pressure levels in IGRA soundings using the same methodology that we have used in our paper (see Section 2.3.1). The comparison shows that the results are not sensitive to the selection of the dataset (see enclosed Taylor diagrams). We expected these results, since the use of homogenous data is particularly important for long-term trends, and we are simply analyzing five years of data.

>> Finally, as stated in the reply to the reader's comment, our results are robust to the selection of the observational dataset. However, as we feel that comment leads to a better paper, the figures included in it will be incorporated to the final version of the paper.

>> -------------------------------------------------

>> González-Rojí, S. J., Sáenz, J., Ibarra-Berastegi, G., & Díaz de Argandoña, J. (2018). Moisture balance over the Iberian Peninsula according to a regional climate model: The impact of 3DVAR data assimilation. *Journal of Geophysical Research: Atmospheres*, *123*(2), 708-729.

27. L135: For readers outside of Europe it would be helpful to include here also local times (approximately).

>> The local times were added to that line.

28. L146: what is meant by "…the analysis increments are stronger at 12 UTC…"? And by "Strong increments are observed during summer…" in L148? Also the relation to the cold-bias in L149 is unclear.

>> The effect of the data assimilation is measured by the analysis increments (analysis minus background) at the analysis times (00, 06, 12 and 18 UTC). We analysed these quantities In González-Rojí et al. (2018), and we showed that the effect of the data assimilation was more intense at 12 UTC compared to the other times, and particularly for summer. The spatial analysis of these

values highlighted that the effect of data assimilation is not homogeneous over the Iberian Peninsula, and it concentrates mainly in the southeastern IP and both Guadalquivir and Ebro basins. This is related to the well-known cold bias observed in the IP in summer in WRF simulations, as the data assimilation is able to make it much smaller.

>> All these lines were edited to clearly state what is explained in the previous lines.

>> -------------------------------------------------

>> González-Rojí, S. J., Sáenz, J., Ibarra-Berastegi, G., & Díaz de Argandoña, J. (2018). Moisture balance over the Iberian Peninsula according to a regional climate model: The impact of 3DVAR data assimilation. *Journal of Geophysical Research: Atmospheres*, *123*(2), 708-729.

29. L150: "…the effect of the assimilation is not restricted only to the station location". This is a very crucial point. Unfortunately, you did not show that. see major point 2

>> That line is not related at all with the conclusions of this paper. As stated there, we are highlighting the fact that the data assimilation not only affects the nearest point to the observation, but it also is able to affect the nearest points as its effect is propagated zonally, meridionally and vertically. That is something expected since the main goal of the background error covariance matrices is to define how the effect of the data assimilation is propagated to the near cells.

>> Regarding the major point 2 from the reviewer, as stated already there, it is true that the comparison against assimilated soundings can be biased, but we are analyzing derived variables not directly assimilated on a regional domain covering places with no observation at all. Additionally, comparing station data against the mean of the 3x3 grids over the Iberian Peninsula would be ideal for convection permitting scales, but not for our 15 km resolucion domain.

30. Sect. 2.3.1: Please explain briefly how you compute the lifting curve from the surface/mixed level to the LCL to the LFC and to the LNB (including quantification of e).

>> All these information is already included in Sáenz et al. (2019) and also in the manual of the R package aiRthermo associated to that publication (available in CRAN). As stated there, "To compute CAPE and convective inhibition (CIN), the vertical integrals are computed in pressure levels by adding the energy corresponding to discrete slabs defined by linear or logarithmic vertical profiles, which are defined by the soundings. The integrals for each of the slabs enclosed by linear profiles are computed analytically, and the energy corresponding to each slab is accumulated, producing the final value of CAPE or CIN. The integrals are always calculated using the virtual temperature (Doswell and Rasmussen, 1994).

>> There are different methods of accurately determining the lifting condensation level (LCL) or the equivalent potential temperature of an air parcel in aiRthermo. In the first case, the package calculates these variables by computing their vertical evolutions and numerically solving the ordinary differential equation representing their ascent from the initial conditions given by their temperature, pressure, and mixing ratio. For compatibility, functions that allow these variables from well known alternative equations to be computed, such as the approximate method presented by Bolton (1980) to compute LCL, are also provided."

>> This information was already summarized in lines 162-165 of the previous version. These lines have been rewritten to make it much clearer in the new version.

>> --------------------------------------------------

>> Doswell III, C. A., & Rasmussen, E. N. (1994). The effect of neglecting the virtual temperature correction on CAPE calculations. *Weather and forecasting*, *9*(4), 625-629.

>> Bolton, D. (1980). The computation of equivalent potential temperature. *Monthly weather review*, *108*(7), 1046-1053.

31. L158: Do you have any reference for the statement that soundings "take many minutes to measure the profile of the atmosphere"? The multiplicity of soundings I performed in the past took ~ ½ hour to reach the LNB.

>> We think that 30 min could be defined as "many minutes". Anyway, that line has been edited to clearly state that the measurements made by the soundings are not instantaneous.

32. L169: What is "an isobaric precooling" and why was it applied?

>> As stated in that paragraph, in order to follow a similar methodology to that used by the University of Wyoming, the averaged values from the lowest 500 m were considered and an isobaric precooling was applied to the initial parcel state.

>> As shown by references provided in the paper, the computation of CAPE and CIN is very sensitive to the characteristics of the initial parcel that is lifted. We decided to follow the procedure of averaging the lower levels recommended by Craven et al. (2002). However, During the development of aiRthermo, we realized that the use of a parcel averaged for the low layers of the atmosphere could very often lead to underestimations of CIN. The reason is that it can happen that after averaging the lowest levels of the sounding to compute the initial parcel, it is still too hot compared to the ambient conditions. In that case, CIN will never be computed, since the initial parcel is already (artificially) buoyant. Thus, a cooling must be applied to the parcel if it is warmer than the environment. In aiRthermo, two options are available: adiabatic precooling (adiabatic ascent until the lifting parcel crosses the sounding) or isobaric precooling (the parcel is cooled along an isobar until it crosses the sounding so that it is not buoyant)). In our study, the isobaric precooling was chosen.

>> --------------------------------------------------

>> Craven, J. P., Jewell, R. E., and Brooks, H. E. (2002). Comparison between Observed Convective Cloud-Base Heights and Lifting Condensation Level for Two Different Lifted Parcels, Weather and Forecasting, 17, 885–890.

>>Sáenz, J., González-Ro jí,  S.J., Carreno-Madinabeitia, S., & Ibarra-Berastegi, G. (2018). Manual for the R package aiRthermo. https://cran.r-project.org/web/packages/aiRthermo/aiRthermo.pdf

>> Siedlecki, M. (2009): Selected instability indices in Europe, Theoretical and Applied Climatology, 96, 85–94.

33. L172: As TT relies on temperature differences, the unit (°C, K) does not matter.

>> We agree on that with the reviewer. We just wanted to state the definition provided by Miller in 1975. We have edited that line in the new version.

34. L173-174: You should mention that other authors found other values (e.g., Huntrieser et al., 1996: 46 K; Haklander and Van Delden, 2003: 46.7 K; Kunz, 2007: 48.1 K).

>> These references have been added to the text.

35. L174-176: "…not highly dependent on the initial conditions…" correct (but even absolutely independent), but why differ the values you compute by your own from those provided by Wyoming – and only at Murcia (strongest), Santander, Zaragoza, Barcelona? Prevailing inversion layers as stated in L176 cannot be the reason as TT is based on main pressure levels which are always provided by Wyoming. Considering the initial values, you stated that you used the same mixing over the lowest 500 m, thus your method must be identical to that of Wyoming. How have you determined the LCL/LFC?

>> The differences are due to the different methodologies in the calculation of the TT index, mainly due to the Dew Point temperature. In the case of the University of Wyoming, the value is taken directly from the measurement. However, in our case, we do not use that measured value and we calculate it with the Pressure, Temperature and Mixing ratio at 850 hPa.

>> Figure 1 shows the scatterplots of the monthly means of the observed TT (computed from the values obtained directly from Wyoming University), and the ones computed with aiRthermo. In all of the cases, the R2 is above 0.99 for all the stations with the exception of Murcia (0.96). The worst values are obtained in the stations highlighted by the reviewer: Murcia (strongest), Santander, Zaragoza, Barcelona. Particularly in those stations, the values obtained by the University of Wyoming are larger than the ones obtained by aiRthermo from measured pressure, temperature and mixing ratio. The differences between the values are small, but they affect the correlation.

[Figure]

**Figure 1**: Scatterplots for the monthly values of the observed values of TT (directly taken from the University of Wyoming) against the monthly values of TT calculated with aiRthermo. The R-squared is presented in the bottom right corner of each plot.

>> We do not believe that this Figure should be added to the manuscript, but some details about these results will be given in the text after the Taylor diagrams for TT.

36. Sect. 2.3.2 / Results: The statistical distribution of CAPE is highly skewed. The product moment correlation coefficient according to Pearson, however, require a normal distribution. A better choice would be the rank correlation coefficient according to Spearman.

>> Tailor diagrams can only be created by means of Pearson's correlation. In order to show visually which experiment is best compared to the reference data, we decided to sacrifice the use Spearman correlation. The reason for this is that Taylor diagrams are built considering the mathematical relationship between the correlation coefficient, the RMS error and the variance of the series.

>> However, we show here the comparison of the bootstrap results for both correlation types. Figure 2 shows the results for CAPE and Figure 3 the results for CIN. If we focus only on Figure 2, we can see that the values are similar in most of the stations and only in A Coruna and Santander there is a strong worsening of the values. In any case, the same structure is observed: aiRthermo is the closest one to the values obtained by IGRA, followed by Wyoming, WRF D (experiment including data assimilation) and finally by WRF N (without data assimilation).

[Figure]

**Figure 2**: Box and Whiskers for the correlations obtained during the bootstrap for CAPE and calculated by different methods: Pearson in green, and Spearman in blue. IGRA stations are taken as Reference.

>> If we change to Figure 3, the worsening of the correlations is perceptible in A Coruna, Santander, Murcia and Gibraltar. As for CAPE, aiRthermo shows the highest correlations with IGRA dataset, followed by Wyoming, WRF D and WRF N. Particularly for Gibraltar, where the Pearson correlations were similar for both WRF experiments, differences between both of them arise if Spearman correlation is used: in that case, as in the other stations, WRF D obtained better correlations than WRF N. Thus, our conclusions still hold with Spearman correlations, but if we used it, we would lose the nice visual properties of the Taylor diagram and the associated diagnostics using RMSE or fractions of variance, which are also important.

[Figure]

**Figure 3**: Same as Figure 2 but for CIN.

>> As already said before, Taylor diagrams can only be constructed with the Pearson correlation. Thus, in the new version of the manuscript, both Figures showing the Taylor diagrams for CAPE and CIN will be updated. The Box and Whiskers will be changed to these new versions in order to show also the Spearman correlations.

37. Sect. 2.3.2, last sentence: please delete the statement about precipitation (cf. major point 4).

>> We have edited the sentence as suggested by the reviewer in comment 4 to "These maps show the regions over the IP where the unstable conditions are more prevalent in each season."

38. L188: "by independent locations": Independent in which sense? The locations are not independent right now.

>> We do not understand why the reviewer says that the locations are not independent when they are located in different regions of the Iberian Peninsula, and they are several kilometers far from each other, in different climatic areas and for some fields, such as CAPE with a horizontal scale length smaller than the distance between sites. However, this sentence was edited to highlight these facts.

39. Section 3. Results: To facilitate direct comparison of the subfigures in a panel, it would be very helpful if they have the same axis range.

>> As already stated in Major comment 2, we agree on this comment with the reviewer because having the same scaling in the figures is easier to interpret the results. However, that is not possible in our case because the values show a really large range.

40. L203: Please explain how you selected a model as the "best" model: by the highest correlation coefficient, the lowest rmes, a similar SD, or a combination thereof?

>> As we are using Taylor Diagrams to show the Pearson correlation, RMSEs and standard deviations of each experiment against the reference values, we are following Taylor's suggestions to select the "best" model. As stated by him, "the simulated variables that agree well with observations will lie nearest the point marked 'observed' on the x-axis. These models will have relatively high correlation and low RMSEs, and those lying on the dashed arc will have the correct standard deviation" (Taylor, 2001; Taylor 2005). We will add this information to the new version of the manuscript.

>> --------------------------------------------------

>> Taylor, K. E. (2001): Summarizing multiple aspects of model performance in a single diagram, J. Geophys. Res., 106( D7), 7183– 7192.

>> Taylor, K. E. (2005). Taylor diagram primer. *Published to web at:*
 *https://pcmdi.llnl.gov/staff/taylor/CV/Taylor_diagram_primer.pdf?id=96*

41. L215-216: What is the reason of the small differences between Wyoming and aiRthermo both relying on the same data – in particular for TT which does not involves any assumption about lifting? Why are the differences largest at Murcia?

>> As already stated in minor comment 35, the differences are due to the different methodologies in the calculation of the TT index, mainly due to the Dew Point temperature. In the case of the University of Wyoming, the value is taken directly from the measurement. However, in our case, we do not use that measured value and we calculate it with the Pressure, Temperature and Mixing ratio at 850 hPa. As shown in Figure 1, the values of TT obtained by the University of Wyoming are larger than the ones obtained by aiRthermo. This is observable in Santander, Zaragoza, Barcelona, and particularly in Murcia.

42. L223: Again, are there any reasons why the two stations of Murcia and Barcelona show the largest differences (rmse) compared to the other stations?

>> Again, this is due to the different methodologies followed to calculate TT index. Wyoming University uses the measured Dew Point Temperature, but in aiRthermo we calculate it with the pressure, temperature and mixing ratio at 850mb. There may exist different methods to estimate saturation pressure of water vapour implied in the calculation of Td. See minor comment 35 and 41.

43. L231: "…small differences in initial conditions…"; can you be more specific here (also with regard to TT, as already alluded above)?

>> As replied in our previous comments, the differences are due to the fact that Wyoming University uses the measured Dew Point temperatures to calculate its indices, while our methodology is only based on pressure, temperature and mixing ratio. In the case of aiRthermo, we only used those measurements and we do not include Dew Point Temperature in any case.

>> For the calculation of CAPE and CIN, the Lifted Condensation Level (LCL), the Level of Free Convection (LFC) or the Equilibrium Level (EL) must be calculated. Depending on the methodology used (the number of low levels averaged for the initial parcel, the definition of the saturated pressure as a function of temperature, truncation errors due to the number of digits used in the ascii files storing the soundings to name three examples), the location of these levels can vary and this can trigger differences in CAPE and CIN. Everything starts with the calculation of the LCL, and that is calculated using the Dew point temperature. Small differences in those values can trigger differences in the values of CAPE and CIN.

>> Even if differences are expected, thanks to Figures 6 and 7 of the current version of the manuscript, we can see that the distribution is similar for both Reference (Wyoming data) and aiRthermo in all the stations. This is also applicable to the TT index, as shown in Figure 1 of this reply, and Figure 5 of the manuscript.

44. Figure 3 (CAPE) shows very large differences of the standard deviation between the different models and for some of the stations. Any idea on that?

>> The only difference in the configuration of both WRF experiments is the data assimilation in the D experiment. Thus, it is clear that this should be the reason.

45. L250: Could you be more specific?

>> This sentence has been edited to highlight the ability of the data assimilation to produce more reliable results (similar to those derived from measured data).

46. L254: "N tends to overestimate the variability in every season and for most of the stations…" Why?

>> Again, the only differences between WRF experiments is the data assimilation scheme included in D. Thus, the data assimilation corrects the temperature, pressure or/and mixing ratio from the model.

47. L255: "..presents the largest values during winter, which agrees with the fact that the northern and northwestern IP receives greats amount of rain during that season". Is winter rainfall really dominated by convective precipitation? I cannot find any statements in the cited literature. Which of the Atmospheric patterns AP1-19 defined by Romero et al., 1999 are convective patterns? Rodriguez-Puebla et al., 1998, considers only the relation to teleconnection patterns, but did not classify precipitation.

>> In that sentence we are not referring only to convective precipitation, as can be inferred from the cited papers, which are also discussed in the Introduction in order to present the seasonal regimens of precipitation observed in the IP. As stated in that line, we only highlight that the largest TT values are located in the regions where more precipitation is measured during winter, without any comment related to convective precipitation.

>> In order to avoid wrong interpretations, this line have been rewritten in the new version of the manuscript.

48. L255-260 and Fig. 5: Why does Lisbon show higher TT values in winter than in summer? You may also mention that the differences between the models at Lisbon, La Coruna, and Santander are larger in winter compared to summer (why?).

>> Figures 4, 5 and 6 show the seasonal mean maps (winter and summer only) for the variables involved in the calculation of TT index, that is, temperature and dew point temperature at 850 hPa and temperature at 500 hPa respectively.

[Figure]

**Figure 4:** Spatial distribution of mean Temperature at 850 hPa for period 2010-2014 over the IP as computed from N (first column) and D (second column) for winter and summer. The median value (K) is in the bottom right corner of the plots.

[Figure]

**Figure 5:** Same as Figure 4 but for dew point temperature at 850 hPa.

[Figure]

**Figure 6:** Same as Figures 4 and 5 but for temperature at 500 hPa.

>> Taking into account the information from these figures, the shorter values of TT obtained in Lisbon during summer than in winter are due to the the shorter Td values obtained in that area in summer. Particularly, the values of temperature at 850 and 500 hPa increase in that region from winter to summer (about 15 and 10 degrees respectively), while the dew point temperature is only a few degrees larger. That produces the reduction of the TT values from winter to summer near Lisbon.

>> These Figures also depict the differences between both WRF experiments. In Figure 4, we can see that the data assimilation increases the temperatures over the IP in winter, while it reduces them in summer (particularly in the southeastern corner of the IP). For the dew point temperature, the reverse is observed in Figure 5: the dew point temperatures are reduced in winter, while they are increased in summer (with the exception of the western facade of the IP where they are reduced). In Figure 6 we can see that the temperatures at 500 hPa are slightly increased in winter and slightly reduced in summer. These differences are the ones shortening the differences between both WRF simulations from winter to summer.

>> We do not think that these figures should be included in the final manuscript. Thus, a short summary of these results will be included in the text and the figures will be provided as supplementary materials.

49. L261-262: Again I miss a reference that shows not only total precipitation, but a classification among the types (stratiform/convective).

>> As already said in comment 47, in those lines we are not restricting our results to convective precipitation, and we only highlight the fact that these regions are the ones obtaining more precipitation in that season.

>> The line will be rewritten to avoid misunderstandings in the new version of the paper.

50. L268-270: Why does D overestimate CAPE at most stations in spring? And why does the N experiment overestimate CAPE in winter and underestimate it in summer?

>> Figure 7 shows the median of vertical profiles of virtual temperature for the soundings and the lifted parcels until 550 hPa in spring. The dashed lines represent the 5 and 95 percentiles. If we focus only on the soundings from IGRA, WRF D and WRF N, we can see that in general, D is closer to the observed virtual temperature (particularly in A Coruna, Santander, Barcelona and Zaragoza). Additionally, both WRF experiments tend to warmer conditions between 800-750 and colder near surface (until 900 hPa). If we switch to the lifted trajectories, these are warmer for D and colder for N in most of the stations. Additionally, N tends to cross the sounding later than D (e.g. Lisbon, Santander or Gibraltar).

>> Thus, both WRF simulations overestimate CAPE during this season due to the differences in the virtual temperature in lower levels (colder near surface and warmer near 800 hPA compared to measured data). In combination with the fact that the lifted trajectories for D are slightly warmer than the observed ones and N, this experiment overestimated CAPE in most of the stations during that season.

[Figure]

**Figure 7:** Vertical profiles of virtual temperature for the sounding levels and for the lifted parcel during spring. The dashed lines represent the 5 and 95 percentiles, and the solid lines the median.

>> Figure 8 shows the results for Winter. If we focus on the soundings, we can see that in most of the stations D is much more similar to IGRA than N. During these season, the soundings tend to be colder in low levels (below 800 hPa) for N compared to IGRA. In the case of the lifted trajectories, these are warmer for N. Thus, the combination of these two factors (colder soundings and warmer lifted trajectories) cause the overestimation of CAPE for the N experiment. This is well observed in Lisbon, A Coruna and Santander.

[Figure]

**Figure 8:** Same as Figure 7 but for winter.

>> If we change to summer, Figure 9 shows the corresponding trajectories and soundings. In this season, N shows an overestimation of CAPE, which is clearly observable in Barcelona, Murcia and Gibraltar. In these three stations, N shows warmer sounding levels that IGRA, which produces that the lifted trajectory crosses earlier than the other experiments the sounding, and consequently, underestimating the CAPE.

[Figure]

**Figure 9:** Same as Figures 7 and 8 but for summer.

51. Figs. 5/6: At Barcelona, Zaragoza, and Murcia, CAPE is highest in summer, whereas TT reaches highest values in spring at these stations. What is the reason of the obvious discrepancy between CAPE and TT?

>> If we focus on those stations highlighted by the reviewer, it is clear that it seems to be a discrepancy between those indices (even if the differences between the TT values between spring and summer are below 2 degrees). However, the reviewer forgets that these values of CAPE (and also CIN - Fig 7) are calculated for the entire series of 12 hourly values obtained in each station during 2010-2014. Thus, these values are not restricted to highly convective events. Consequently, Fig 6 and 7 must be compared to the values of TT in combination.

>> If we do so, we can see that the not extremely high values of CAPE observed in Barcelona, Zaragoza and Murcia during spring are contrasted with the highest values of CIN of the season in those stations. In contrast, during summer, extremely high values of CAPE are observed in those stations (medians over 200 J/kg for Barcelona and Murcia, but they can reach values over 700 J/kg), but CIN is not comparable to those values of CAPE (below 150 for Barcelona and Zaragoza, around 200 for Murcia). Thus, Barcelona and Zaragoza present unstable conditions in summer, but not in spring.

>> Since CAPE and CIN are dependent on the entire profile of the atmosphere, CAPE and CIN should be more reliable than TT.

52. L281 and following: As already mentioned above (see major comment 5), CIN is relevant for convection only in combination with CAPE (An example: imagine a day with zero CAPE and zero CIN; another day with CAPE = 3000 J/kg and CIN = 300 J/kg. None of the days would have the right conditions for deep moist convection to occur. The average of the two days would give CAPE = 1500 J/kg and CIN = 150 J/kg. Fair values for DMC). You could simply fix that by considering CIN only on days for CAPE in excess of 50 or 100 J/kg.

>> As already stated in major comment 5, that is not the objective of our paper. The objective of this paper is to evaluate the ability of WRF simulations (including or not the 3DVAR data assimilation step) to produce reliables values of TT, CAPE and CIN over the Iberian Peninsula. Once that has been evaluated by means of Taylor diagrams, we want to show the distribution of the values of the complete time series from 2010 to 2014 for each index. We are not trying to make any prognosis or diagnosis of convective events from the data we prepared.

53. L305: "…lowest values are observed near the coastal valleys…" why?

>> Figures 10, 11 and 12 show the seasonal mean maps for 00 and 12 UTC in winter and summer for temperatures and dew point temperatures at 850 hPa, and temperature at 500 hPa. According to these results, the lowest values of TT observed near the coastal valleys are originated due to the low values observed in those regions for dew point temperature, which at the same time is originated by the low mixing ratio values in those regions. The low values are observed at 00 UTC, and they are even lower at 12 UTC. As a result, the TT values are low in those coastal regions, independently of the facade of the Iberian Peninsula. This information was added to the new version of the manuscript, but not the Figures as they look similar to the means for winter and summer included in comment 48 (which will be included to supplementary materials).

[Figure]

**Figure 10:** Spatial distribution of mean Temperature at 850 hPa for period 2010-2014 over the IP as computed from N (first column) and D (second column) for winter and summer at 00 and 12 UTC. The median value (K) is in the bottom right corner of the plots.

[Figure]

**Figure 11:** Same as Figure 10 but for dew point temperature at 850 hPa.

[Figure]

**Figure 12:** Same as Figures 10 and 11 but for temperature at 500 hPa.

54. Figure 8/9: The spatial distribution of TT and CAPE in most of the cases is contrary, i.e. regions with higher CAPE have lower TT values and vice versa. Any explanation of this apparent contradiction?

>> The apparent discrepancy highlighted by the reviewer is only observed in winter near Lisbon and the western facade of the IP. This is not observed in summer, where the highly unstable areas are mainly observed towards the Mediterranean coast in both TT and CAPE.

>> However, it must be taken into account that our results for CAPE and CIN are not restricted to highly convective events, and they represent the mean values computed during 2010-2014. As can be seen in Figures 13 and 14 (A Coruna and Gibraltar as an example), TT and CAPE are related to atmospheric instability, but they are not related through a simple linear relationship (R2 below 0.2 for all the stations and seasons), particularly for stable or neutral atmospheres. Since we are showing the results corresponding to all the observations, the relationship does not need to be simple. This is expected, since TT is a diagnostic computed from discrete levels and CAPE and CIN involve the vertical integral along the atmosphere.

[Figure]

**Figure 13**: Scatterplots for the values of CAPE and TT as included in IGRA for A Coruna. The values of CAPE over the 60th percentile are in blue, and the values below that value are in red. The value of the 60th percentile is marked with a grey line, and the linear models are also included with the corresponding colors.

[Figure]

**Figure 14**: Same as Figure 13, but for Gibraltar.

>> This apparent discrepancy was addressed in the new version of the manuscript, and the above presented details were included in the new version of the manuscript..

55. L326 and following: See major comment 5 and minor 52.

>> As already stated in those comments made by the reviewer, restricting the values of CIN and not including the entire time series for period 2010-2014 is not part of the objective of our paper. We are not restricting the evaluation of these instability indices to the extreme events, and we are evaluating the performance of the model at simulating them. In this case, in order to show the different patterns obtained by each WRF experiment, the mean values of the entire period where chosen.

56. L336-345: The relation to "dynamics" does not fit here as the paper solely has a thermodynamical perspective. Be careful with the relation between convective conditions and precipitation.

>> The term "dynamics" was used only to introduce the fact that two different patterns are observed between winter and summer. In any case, it was not related at all to the relationship between convective conditions and precipitation. Thus, those lines were edited as suggested by the reviewer.

**Edits:**

1. L5: explain "IP" at first use; the same applies to "SD" in L8

2. L6-7 "measured variables at different pressure levels" may be replaced by "vertical temperature and moisture profiles".

3. L9 methodologies methods

4. L11: "an air parcel's vertical evolution" "a lifted air parcel"

5. L13: "…reference values" of which quantity?

6. L15: "**in** the Mediterranean coast" "**over** the…"; never use in the coast, in the station etc.

7. L16: "The chances of developing thunderstorms in those areas at 12 UTC is much higher than at 00 UTC". This generally applies to convection. Delete.

8. L18: Murcia: Don't mention a location in the abstract that is not well known.

9. L22: "in the planet"?? "of the planet"

10. L39 "alone and it should be…" "alone, but should be…"

11. L39: There are much more lifting mechanisms, thus include ", for example, by orography, …"

12. L48: **On** the global scale

13. L50: "convective storms develop for lower values…"

14. L62: "CAPE presents" "CAPE shows a high…"

15. L72-73: use the plural: hailstorms and thunderstorms

16. L74: "similar to those observed in Europe.." Where in Europe? Is the IP not part of Europe? What is meant by "dispersion of the values"?

17. L100: exercises experiments

18. P4: please explain all abbreviations at first use

19. L109-110: ".., and they use 51 levels" "and with 51 levels"

20. L111: form from

21. LL123: Both simulations, **N and D**, …" (note that you refer to another simulation in the previous section).

22. L136: either delete "..where they only measure it at…" or rewrite this sentence

23. L141: suggestion: "Additionally, vertical profiles of pressure, temperature, and mixing ratio obtained…"

24. L142: delete "indices"

25. L147: "This pattern…" which pattern do you refer to? What is meant by intensity?

26. L154: better use WRF's original eta levels instead of models's to avoid any misunderstanding that you quantified CAPE/CIN only from the 20 ERA-Interim levels.

27. L187: "…of the error/differences due to the different methods applied"

28. L203: be careful with the wording, aiRthermo is not an experiment but simple a quantification.

29. L231: trigger result in

30. L233: "…between **the** experiments"

31. L255: "A Coruna and Santander…"

32. L273: What do you mean by "some stations are more important than others…"?

33. L295-296: "Figure 8 shows…" and "The heterogeneity…" reformulate these two sentences

34. L301: important higher

35. L317: "…the unstable area **air mass** is found…" An area cannot be unstable

36. L318: For readers outside of the IP, can you give a hint about the location of Tagus and Guadalquivir rivers? And later, L331: the Ebro basin?

37. L320: "On the contrary **Compared** to what…" Or to which contradiction you refer?

38. L321-322: "where tunderstorms can be developed" "the area with higher CAPE values…"

The edits suggested by the reviewer were applied to the new version of the manuscript.

[revised manuscript text omitted]

---

## Referee Report (RR1)

**2nd Review „Changes in the simulation of instability indices over the Iberian Peninsula due to the use of 3DVAR data assimilation "**
**by Santos J. González-Rojí and co-authors submitted to HESS**

The authors have done a good job in addressing the comments of all reviewers and, as a result, the manuscript has improved considerably. However, I have still have some points that have to be considered before the paper can be accepted for publication.

To clarify, my intension was not to annoy you with my critical review, but rather to improve the scientific quality of the paper and make it more useful for a broader community. This is also an issue for a more or less pure meteorological topic to be published in a journal focusing on "fundamental and applied research that advances the understanding of hydrological systems, their role in providing water for ecosystems and society, and the role of the water cycle in the functioning of the Earth system" as HESS.

I accept that according to several replies to my comments and questions, the objective is solely to show that the application of data assimilation improves the simulation of three convective variables in their region. The authors obviously do not want to dig deeper and scrutinize possible reasons for the changes nor do they intend to discuss reasons of the high spatial variability – even if this would be of interest to many readers. With this limitation, I'm not sure whether the paper is suitable to be published in HESS. I leave the decision to the editor.

Other points that have to be considered are:

1.) Your answer to my major revision point 1 is not satisfactory. At least a statement about the reasons of the increased reliability (more realistic temperature profile or humidity profile) is required. And reasons **why** the parameters at some locations show a larger difference that at other locations must be given.

2.) My former major revision point 2: Why should one expect a lower model prediction skill without assimilation? It's right that "the impact of data assimilation is not limited to the grid cells close to the location of the soundings…. The changes extend over large areas". But from that you cannot conclude that the model in general performs better when you restrict your evaluation only to points where you assimilated data.

3.) Your answer to my former 3rd revision point: even though if I'm not fully convinced, you should at least comment on that point in the paper.

4.) Your answer to my former 6nd revision point: Please give a comment also in the manuscript.

5.) Your answer to my former 8nd revision point: The conclusion is intended to help the reader understand why your research should matter to them; it should not simply reiterate your results or the discussion; recommend a specific course or courses of action; critically refer on the relevance of your research, and also refer your work to other references. Try at least to consider a few points of these points. Otherwise rename this section into "Summary".

**Minor revision points:**

1. My former minor revision point 1: The references you have cited also used other indices and considered other regions. Still: Why did you selected these three parameters, why not, e.g., SWEAT, LI (which, according to several papers, has the highest predictive skill) or others? Simply because these were available and other not? A simple statement on that in one sentence is sufficient.
(Note that the second paragraph of your answer (5 vs. 30 years) is very speculative).

2. My former point 6: CAPE is a measure of convective instability, but not a convective **index**.

3. My former point 10: it still reads "…convective precipitation is usually associated with extreme events…" and this is wrong; change "usually" by "frequently", but also define "extreme events" (do you refer to rainfall, hail, or wind?); "due to high intensity and short duration"; the short duration is related to convective processes and does not make convective precipitation an extreme event per se; change into "due to high intensity over a short duration".

4. My former point 16: I know, but this is a statement not supported by their research. Please change this reference into a more appropriate, e.g., Graf et al., 2011.
Graf M. A., Sprenger M., and R. W. Moore, 2011: Central European tornado environments as viewed from a potential vorticity and Lagrangian perspective. #are#, 101, 31–45, https://doi.org/10.1016/j.atmosres.2011.01.007.

5. My former point 23: Basically, I agree; but please explain here in one clause or sub-clause **why** you want to evaluate that.

6. My former point 47 (and 49): Even though you are not directly referring to convective precipitation here, you are discussing TT in relation to rainfall. If precipitation is dominated by other processes such as lifting associated with positive PV-anomalies, TT doesn't matter.

7. My former points 50 and 51: Have you included a short statement about this in the manuscript?

8. My former point 52: I'm not really convinced by your reply not to consider the convective situation. It would be much more interesting to see how well CIN is modeled in convective situations and not on days, where CIN has no meaning. Besides, consideration of the relation between CIN and CAPE would make the content of the paper much more interesting for a wider community (also for me as a meteorologist ☺).

9. There are still some linguistic corrections to be made, but I think these will be fixed by the journal's edit.

10. The introduction broadly describing convective activity across Europe and other features related to convection in general is well written. However, it does not really fit to the main content of the paper focusing on "the performance of two simulations created by using WRF model". At least the topic of data assimilation as the central point of the paper has to be introduced here.

Further minor points:

1. L18-19: Mean CAPE is always higher at 12 UTC that at 00 UTC; this is not worth mentioning in the abstract.

2. L30: include "…complex topography, **insufficient assimilated observations, and** forecast errors…"

3. L33: delete the words "extreme" as you do not separate among the precipitation intensity.

4. L39:  deep convection…

5. L42: of  atmospheric convection

6. L49: precipitation extreme events → convective precipitation

7. L60: most unstable region → region with highest instability

8. L67: There is something missing in this sentence (verb and subjective)

9. L86:  mean CAPE

10. L102: extreme events → convective situations

11. L115: both wind components → **horizontal** wind; geopotential → geopotential **height**

12. L125: presents the same pramameterizations → relies on the same setup (the parameterizations used are introduced later)

13. L140/163: write out NOAA at first use

14. L167: …12 UTC, 02 and… → 12 UTC, corresponding to 02 and 14 LT, respectively)

15. L168: in L167 12 UTC was 14 and not 13 LT
16. L173-179 is somewhat cumbersome (two almost identical sentences)
17. L186 highlighted → highlight
18. L187: This… refer to what?
19. L188: in **our** WRF simulations
20. L196 as we would be taking into account → as we take into account (you did that, didn't you?)
21. L197-198: This sentence is unclear. "levels are already measured for a drifting distance"? What is the distance of 4.5 km? How is the drifting related to the cloud cover??
22. L200: "This procedure…" What do you refer to?
23. L204: still, it's not minutes rather more than one hour for an entire profile of the atmosphere
24. L205: "…because of wind"; already discussed above.
25. L222: it **was** not → it **is** not
26. L233: "Lifted Condensation Level" → "**Lifting** C…" is more appropriate (cf. AMS Glossary)
27. L235 trigger → cause
28. L237-238: "provide similar information"; I fully disagree with this statement, see the bunch of literature on the various convective indices quantifying conditional, latent, potential instability, or a combination thereof!
29. L262-264 and elsewhere: Spearman / Persons's correlation **coefficient**
30. L288/L306: worsening → decrease (avoid personal assessment)
31. L293: are **most** remarkable in Murcia
32. L295-96 and **elsewhere** (e.g., 299): **in** those stations → **at** those stations; **at** the Mediterranean coast; you cannot say in a station; this was already explained in my 1st review.
33. L300 **at/for** Barcelona
34. L308: **S**ection 2.3.1; **of** TT index
35. L332 (and elsewhere): trigger → cause
36. L346: ) and Gibraltar
37. L347/48 shorter → smaller
38. L363-64: "warmer sounding levels" is strange; Reference → the reference
39. L362-64: that's of course not a trajectory! → lifted air parcel; the sentence "which produces that the lifted trajectory crosses earlier than D the sounding" is very strange
40. L374: "most active ones"?? You mean highest CAPE values?
41. L399: "…system, so the lifting that can trigger convection can appear" → "…system that can trigger convection by orographically induced lifting"
42. L406-407: replace the sentence beginning with "…are originated…" → "are a consequence of low dew point temperature mainly due to dry air."
43. L412-13: This sentence is a fragment (no Verb and Subjective); again, you cannot say "in the slope"
44. L417/418 and elsewhere: **on** the Atlantic coast; **on** the western coast
45. L419: that → than
46. L438 results **from** TT → results **for** TT
47. L440 highly convective events: this is now very confusing. You are investigation mean values, aren't you? Why are you speaking of events? And why "highly"? Did you separate among different intensity classes? I think you did not…
48. L441: "TT and CAPE are indices for atmospheric instability"; No. The one is an index (TT), the other is an integral bulk of convective energy and **not** an index. Besides, CAPE solely estimates latent instability, whereas TT combines conditional (VT), latent, and potential instability.

49. L443-44: "since CAPE and CIN are dependent on the entire profile of the atmosphere.." There are two flaws: you mean the profile below the level of neutral buoyancy and not the whole atmosphere (which is unbounded); and CIN depends only on the layers below the LFC. Furthermore, several studies have shown that – at least in Europe – the Lifted Index has a higher prediction skill than CAPE, even if considers only a parcel lifted to 500 hPa.
50. L446: on the Atlantic coast
51. L449: what do you mean by intensity? When you refer to that, you need to consider CAPE in combination with CIN (as you know, in case of high CIN, instability cannot be released and the intensity is low no matter of CAPE).
52. L451: delete §dynamics"
53. L469: the correlation → correlation coefficients according to Pearson and Spearman
54. L470: "…of them" of what?
55. L489: delete "develop"
56. L490: **convective** inhibition

---

## Author Response (AR2)

Dear Editor,

Please find enclosed the revised version of the manuscript entitled "Changes in the simulation of instability indices over the Iberian Peninsula due to the use of 3DVAR data assimilation" by S. J. González-Rojí, S. Carreno-Madinabeitia, J. Sáenz and G. Ibarra-Berastegi, that we resubmit to the journal *Hydrology and Earth System Sciences.* Please note that due to one of the comments of Reviewer#2, we have modified the title and now it is called: "Changes in the simulation of atmospheric instability over the Iberian Peninsula due to the use of 3DVAR data assimilation".

We consider that the previous version of the manuscript successfully addressed the comments by Reviewer#1, as he/she already accepted it for publication.

Regarding Reviewer#2, we consider that all the major points raised by this reviewer have thoroughly been addressed in the current version of the manuscript. Please, note that we disagree with this reviewer when he/she states that we do not want to "dig deeper". In our previous revised version of the manuscript we already included substantial new material such as Figures A1 to A6 (in the previous numbering) to answer his/her major point number one, or the justification of the selection of the nearest point and not the average to name a few without being exhaustive. In this new version, we have again included new panels to existing figures (A1 to A3) and two new figures (A6 and A8 in the current numbering), and substantial new text in order to give even more details about the causes of the differences between both experiments.

Attached to this cover letter you will find the new version of the manuscript, the version with the changes tracked, where all the thorough modifications are highlighted, and a detailed response to the reviewer's comments. Thus, we consider that we have successfully addressed all the points raised by this second reviewer and, as such, we hope that the manuscript can be accepted this time.

Yours faithfully,
Santos J. González-Rojí

**Reply to 2nd Review made by the Anonymous Referee #2**

**Received: 11 November 2020**

Reply by the authors is shown in blue, and starts with the symbol >>.

The authors have done a good job in addressing the comments of all reviewers and, as a result, the manuscript has improved considerably. However, I have still have some points that have to be considered before the paper can be accepted for publication.

To clarify, my intension was not to annoy you with my critical review, but rather to improve the scientific quality of the paper and make it more useful for a broader community. This is also an issue for a more or less pure meteorological topic to be published in a journal focusing on "fundamental and applied research that advances the understanding of hydrological systems, their role in providing water for ecosystems and society, and the role of the water cycle in the functioning of the Earth system" as HESS.

>> We never interpreted your comments as annoying and we have carefully re-read the replies that we submitted to your previous review, and we can't understand your paragraph above. It's true that we did not always agree with your suggestions but, in those cases, we always wrote the explicit reasons under our point of view (either target audience, main objective of the paper or availability of verification data, for instance).

>> Regarding the scope of the journal, we think that our study fits in HESS as it focuses on the distribution of instability indices over the Iberian Peninsula, which are variables that can be used to estimate the most unstable regions and where convective precipitation can be triggered if all the ingredients needed are fulfilled. Consequently, it falls in the one of the three points stated in the scopus of the journal, which is "the study of the spatial and temporal characteristics of the global water resources (solid, liquid, and vapour) and related budgets, in all compartments of the Earth system (atmosphere, oceans, estuaries, rivers, lakes, and land masses)."

I accept that according to several replies to my comments and questions, the objective is solely to show that the application of data assimilation improves the simulation of three convective variables in their region. The authors obviously do not want to dig deeper and scrutinize possible reasons for the changes nor do they intend to discuss reasons of the high spatial variability – even if this would be of interest to many readers. With this limitation, I'm not sure whether the paper is suitable to be published in HESS. I leave the decision to the editor.

>> It seems that from our replies, the reviewer inferred that we do not want to "dig deeper". For instance, in the previous version of the manuscript we included substantial new material addressing his/her comments. In the current version we include further results in order to determine why we obtained different values of the parameters in different regions of the Iberian Peninsula (new Figures A6 and A8 in the Appendix, and new panels in Figures A1-A3).

Other points that have to be considered are:

1.) Your answer to my major revision point 1 is not satisfactory. At least a statement about the reasons of the increased reliability (more realistic temperature profile or humidity profile) is required. And reasons **why** the parameters at some locations show a larger difference that at other locations must be given.

>> The observed values of TT, CAPE and CIN are reproduced correctly by the experiment including data assimilation (D) as shown by Figures 2 and 5 for TT, Figures 3 and 6 for CAPE and Figures 4 and 7 for CIN. This is not the case for the experiment without data assimilation (N), as it can be shown in the same Figures, where poor correlations or some seasonal biases can be observed.

>> Additionally, these results agree with previous evaluations of these two WRF experiments. As already stated in the manuscript at the end of section 2.1, these two simulations were fully validated in previous studies of the authors (already cited there). Precipitation, Evaporation, Integrated Water Vapour or Soil Moisture from the experiment with data assimilation are similar (or even better in some cases) than the ones produced by the driving reanalysis when they are compared against observational datasets (both in-situ or gridded observations). The main difference between both experiments is only the data assimilation approach as both of them include the same physics parameterization schemes, so that is the only reason to produce more reliable results.

>> The effect of data assimilation in humidity and temperature in our simulation was already discussed in the paper González-Rojí et al. (2018) (see their Figure 13), and we gave further details to the reviewer in the last reply we wrote. As stated there, data assimilation is important at 12 UTC for moisture, and at 00 and 12 UTC for temperature, and their effects are relevant in the southeastern IP and both Guadalquivir and Ebro basins. This pattern is consistent along the seasons, but its intensity varies seasonally (stronger during summer than in winter). As presented in González-Rojí et al. (2020), the soil moisture content is also different in both simulations as a result of the data assimilation (this variable is not assimilated, and data assimilation is the only difference in the configuration of both N and D model runs).

>> Regarding why the parameters show larger differences at some locations, we already included the details about it in the previous submitted manuscript (first iteration), particularly after presenting Figures 5-7. We also included further plots in the annexes (Figures A1-A6) to show that the differences observed in the studied indices were mainly due to the differences in the vertical profiles of the atmosphere. So, we consider we already did this by providing a comprehensive explanation.

>> We thought that all this information was clearly added to the manuscript in our resubmitted version, but it seems that it was not. Consequently, we have expanded the information included from previous studies at the end of section 2.1, and we have added a summary about the reasons that cause the differences between experiments in the conclusions. Beyond that, we have added new panels to figures A1, A2 and A3 to be able to discuss differences observed during spring, and

we have calculated the differences of virtual temperature and mixing ratio from both model simulations (D and N) for some particularly interesting seasons of the year and some sounding stations in order to explain in detail some particular results.

2.) My former major revision point 2: Why should one expect a lower model prediction skill without assimilation? It's right that "the impact of data assimilation is not limited to the grid cells close to the location of the soundings…. The changes extend over large areas". But from that you cannot conclude that the model in general performs better when you restrict your evaluation only to points where you assimilated data.

>> We think that the reviewer is talking about two different "problems" in a single statement. On the one side, we have the effect of data assimilation in the prediction skill of the model, and on the other one, the problem of validating the results against assimilated observations.

>> Regarding the first one, the main objective of the data assimilation scheme is to produce more reliable and accurate initial conditions for the regional models. Thus, it is clear that the results from the model should be closer to real measurements than the outputs from a standard run with a regional model without data assimilation. The improvement in the results is obtained as the data assimilation scheme is performed only after considering as first guess the simple forecast from the model, and once the effect of the observations is used to modify the fields of temperature, wind and pressure in order to make them closer to the observations. However, as is clear from the literature (Wang et al., 2008; Bollmeyer et al., 2015; Shen et al., 2016) and our own papers already cited (Figure 13 in González-Rojí et al., 2018), analysis increments do not affect only to the grid point closest to the place where data (such as soundings) are being assimilated. The fact that the optimization of the cost function involves all the domains implies that analysis increments are not only affecting the grid point where data are assimilated. The fact that the assimilation improves the temperature or moisture fields at the beginning also means that the advection of these fields is acting during the six hours from assimilation cycle to the next assimilation cycle. Thus, the improvement imposed by assimilation extends to other areas, and also other fields which are physically related to the assimilated variables. Data assimilation implies that the boundary conditions must be adapted after the data is assimilated, so that the information ingested through the boundaries is also affected. As a result, data assimilation improves variables such as vertically integrated water vapour which are not assimilated but which can be verified against independent data from satellites, as we did in Figures 3 and 4 in González-Rojí et al. (2018) or alternative datasets such as evaporation, which we already tested for the full Iberian Peninsula in Figures 8 and 9 in González-Rojí et al. (2018). These are fields that we have verified in the past against independent sources at grid points different from the ones affected by the sounding and we are sure beyond any doubt (and have already proved it quantitatively) that the improvement is real and not limited to the place where the sounding balloons are released. We have added some sentences making this clear in the paper (lines 104-111 in the current version, end of Introduction).

>> Regarding the validation of the results against assimilated observations, as stated already in our previous reply to the reviewer, it is true that the results could be considered as biased.

However, we can not discard any of these observations when preparing the simulations without performing a damage to the study that we want to perform, particularly when only eight radiosondes are available over the Iberian Peninsula. Thus, in order to get the most accurate results out from the model, it is clear that we should use all the available measurements. Mainly because with such a reduced amount of data, it would make no sense to include for example four radiosondes in the data assimilation scheme, and the remaining four radiosondes for validation. Moreover, as stated already in the previous reply to the reviewer, we do not assimilate directly any of these indices or precipitation, as we assimilate pressure, temperature, humidity and wind.

>> Most of this information is already included in the manuscript, but we have developed further the paragraphs associated with these statements in sections 2.1 and 2.2 in the new version.

>> -------------------------------------

>> Wang, X., Barker, D. M., Snyder, C., & Hamill, T. M. (2008). A Hybrid ETKF–3DVAR Data Assimilation Scheme for the WRF Model. Part II: Real Observation Experiments, *Monthly Weather Review*, *136*(12), 5132-5147.

>> Bollmeyer, C., Keller, J.D., Ohlwein, C., Wahl, S., Crewell, S., Friederichs, P., Hense, A., Keune, J., Kneifel, S., Pscheidt, I., Redl, S. and Steinke, S. (2015), Towards a high-resolution regional reanalysis for the European CORDEX domain. Q.J.R. Meteorol. Soc., 141: 1-15.

>> Feifei Shen, Jinzhong Min, Dongmei Xu (2016) Assimilation of radar radial velocity data with the WRF Hybrid ETKF–3DVAR system for the prediction of Hurricane Ike (2008), Atmospheric Research, 169: 127-138.

3.) Your answer to my former 3rd revision point: even though if I'm not fully convinced, you should at least comment on that point in the paper.

>> We have included in the new version of the paper a few sentences about the fact that ERA-Interim also assimilates these soundings as suggested by the reviewer.

>> According to section 4.3 from Dee et al., (2011), only a few stations are excluded from the data assimilation process, which include near surface wind measurements, surface pressure or relative humidity measurements in high terrain, specific humidity in extremely cold regions or radiosondes below the model surface. Consequently, as in our case, mainly all the radiosonde stations over land must be included in it. Even with that, many studies still use assimilated radiosondes for the validation of their own simulations or even the reanalysis (e.g., Vergados et al., 2014; Simmons et al., 2014; Zhao et al., 2019).

>> -------------------------------------

>> Simmons, A. J., Poli, P., Dee, D. P., Berrisford, P., Hersbach, H., Kobayashi, S., & Peubey, C. (2014). Estimating low-frequency variability and trends in atmospheric temperature using ERA-Interim. *Quarterly Journal of the Royal Meteorological Society*, *140*(679), 329-353.

>> Vergados, P., Mannucci, A. J., & Ao, C. O. (2014). Assessing the performance of GPS radio occultation measurements in retrieving tropospheric humidity in cloudiness: A comparison study with radiosondes, ERA-Interim, and AIRS data sets. *Journal of Geophysical Research: Atmospheres*, *119*(12), 7718-7731.

>> Zhao, Q., Yao, Y., Yao, W., & Zhang, S. (2019). GNSS-derived PWV and comparison with radiosonde and ECMWF ERA-Interim data over mainland China. *Journal of Atmospheric and Solar-Terrestrial Physics*, *182*, 85-92.

4.) Your answer to my former 6nd revision point: Please give a comment also in the manuscript.

>> The information about why we considered the nearest point to the station and not an averaged area was already stated in the reviewed manuscript (beginning of section 2.3.1). However, we have included more details to that paragraph.

5.) Your answer to my former 8nd revision point: The conclusion is intended to help the reader understand why your research should matter to them; it should not simply reiterate your results or the discussion; recommend a specific course or courses of action; critically refer on the relevance of your research, and also refer your work to other references. Try at least to consider a few points of these points. Otherwise rename this section into "Summary".

>> As we already stated in the reply to your comment 1, we have added some sentences to the Conclusions in order to make it suitable and interesting for the readers.

**Minor revision points:**

1. My former minor revision point 1: The references you have cited also used other indices and considered other regions. Still: Why did you selected these three parameters, why not, e.g., SWEAT, LI (which, according to several papers, has the highest predictive skill) or others? Simply because these were available and other not? A simple statement on that in one sentence is sufficient. (Note that the second paragraph of your answer (5 vs. 30 years) is very speculative).

>> As already stated in the manuscript at the end of section 2.3.1, other indices could be calculated. However, some of them are defined in a similar way (e.g., TT and K are calculated based only on temperature at two pressure levels) or previous studies (Blanchard, 1998; López et al., 2001 - Both included already in the manuscript) showed strong correlations between them (e.g., CAPE and LI). Additionally, our main objective is to evaluate the performance of both WRF simulations, so we needed to compare our results to those obtained from different observational sources as the University of Wyoming radiosounding database (also later against IGRA dataset). In

that database, only Showalter index (S), LI, SWEAT, K, TT, CAPE and CIN are available. In the case of IGRA, the same parameters are provided, with the exception of SWEAT.

>> The R package that we created and that is essential for our study, allows us to calculate directly from the pseudosoundings extracted from the model at each grid point the values of different variables and indices such as CAPE, CIN, TT (already used in this paper), LI, S and K. Based on this set of indices, we decided to choose only three of them calculated by different methodologies and not to include all of them as that would make the paper long and repetitive. Based on what we stated in the previous paragraph and in the manuscript, we decided to focus on CAPE, CIN and TT, which are indices that show interesting but not redundant results.

>> Additionally, the last paragraph of section 2.3.1 was extended and further information is given.

2. My former point 6: CAPE is a measure of convective instability, but not a convective **index**.

>> Our former reply to point 6: We know that CAPE and CIN are measurements of energies available (CAPE) or inhibition (CIN) in the column of the atmosphere. However, even if they represent an amount of energy, they can also be considered as indices of instability in the atmosphere. As already stated in our previous reply, many authors before (such as Tsonis, Djuric and Bohren and Albrecht) have defined them as indices in their books for Atmospheric Thermodynamics.

>> In order to avoid further comments about this, we have modified all the sentences where we refer to CAPE or CIN as instability indices, even if we do not agree with the reviewer here.

3. My former point 10: it still reads "…convective precipitation is usually associated with extreme events…" and this is wrong; change "usually" by "frequently", but also define "extreme events" (do you refer to rainfall, hail, or wind?); "due to high intensity and short duration"; the short duration is related to convective processes and does not make convective precipitation an extreme event per se; change into "due to high intensity over a short duration".

>> As suggested by the reviewer, the sentence was adapted. We hope this time it fits the standards of the reviewer. It can be read now:
"In general, convective precipitation is frequently  associated with precipitation extreme events due to high intensity over a short duration"

4. My former point 16: I know, but this is a statement not supported by their research. Please change this reference into a more appropriate, e.g., Graf et al., 2011. Graf M. A., Sprenger M., and R. W. Moore, 2011: Central European tornado environments as viewed from a potential vorticity and Lagrangian perspective. #are#, 101, 31–45, https://doi.org/10.1016/j.atmosres.2011.01.007.

>> We have changed the citation associated with that statement as suggested by the reviewer.

5. My former point 23: Basically, I agree; but please explain here in one clause or sub-clause **why** you want to evaluate that.

>> We have developed further the reasons why the evaluation of the convective field is of interest for the readers in the new version of the manuscript. In that part of the Introduction, we have stated that its importance lies in the fact that the study of these parameters makes it possible to determine with enough precision the regions in which conditions conducive to atmospheric instability can be fulfilled.

6. My former point 47 (and 49): Even though you are not directly referring to convective precipitation here, you are discussing TT in relation to rainfall. If precipitation is dominated by other processes such as lifting associated with positive PV-anomalies, TT doesn't matter.

>> Well, to our understanding we are only pointing out that the strongest values of TT and CAPE (according to point 49 raised by the reviewer) are observed in the same regions where the largest amounts of precipitation are measured in each season. Our intention with those statements is only to show the readers that our results fit the observed patterns of precipitation over the Iberian Peninsula. We do that comparison with the precipitation patterns as we know that, to some extent, unstable conditions in the atmosphere are associated with precipitation.

>> In any case, in order to avoid further problems with those statements, we have deleted them from the manuscript.

7. My former points 50 and 51: Have you included a short statement about this in the manuscript?

>> Most of the details about these results were included already in the manuscript after the first revision, along with the plots included also in the reply to the comments made by reviewer 2. Particularly, they are included in lines 372-388. Regarding the comment 51, also a new paragraph about it was included in previous version (lines 459-467).

8. My former point 52: I'm not really convinced by your reply not to consider the convective situation. It would be much more interesting to see how well CIN is modeled in convective situations and not on days, where CIN has no meaning. Besides, consideration of the relation between CIN and CAPE would make the content of the paper much more interesting for a wider community (also for me as a meteorologist ).

>> We agree with the reviewer in the fact that considering the convective situation only can be of interest to meteorologists or forecasters, but that is out of the scope of our paper as our main objective is only to evaluate the performance of both WRF simulations at simulating unstable conditions in the atmosphere. We have clearly stated this goal in the abstract, introduction, and finally again in the conclusions. Generally speaking, we want to see if all the convective situations observed are well captured by the simulations, and if the values produced by the experiments are similar to those measured. In order to do that, we must consider all the data available in our

simulated period, and we cannot restrict the evaluation only to the convective situations. If we followed the reviewer's suggestion, we would only choose the convective situations observed in reality and we would only evaluate the corresponding days in the simulations. However, we would be missing the convective situations from the model that are not observed in reality, an important point to detect problems in the simulations.

>> The reviewer also asks about the relationship between the studied variables. As we showed the reviewer in our previous response, CAPE and TT indices were not related by a simple linear relationship as the $R^2$ was below 0.2 for all the stations and seasons. In this case, we have calculated the relationship between CAPE and CIN in all the stations available and during summer, as the values are larger during this season. Figure 1 shows that the relationship between these two variables is again negligible according to the $R^2$ values obtained (below 0.1 always). Thus, as in the case of TT and CAPE, even if these two variables are related to the thermodynamic conditions in the atmosphere, they are not related by a simple linear relationship because of the multiple phenomenology that might exist in the real atmosphere. Thus, we do not see the point in separating CIN for different values of CAPE. Additionally, CIN is a measurement of inhibition, so even if it is evaluated by itself, it can also provide important information in the analysis, and it must be validated independently and not restricted according to the values of CAPE.

[Figure]

Figure 1 : Scatterplots for the values of CAPE and CIN as included in IGRA for all the stations during summer. The values of CAPE over the 60th percentile are in blue, and the values below that value are in red. The value of the 60th percentile is marked with a grey line, and the linear models are also included with the corresponding colors.

9. There are still some linguistic corrections to be made, but I think these will be fixed by the journal's edit.

>> A detailed revision and edition of the language was conducted in the new version of the manuscript.

10. The introduction broadly describing convective activity across Europe and other features related to convection in general is well written. However, it does not really fit to the main content of the paper focusing on "the performance of two simulations created by using WRF model". At least the topic of data assimilation as the central point of the paper has to be introduced here.

Further minor points:

1. L18-19: Mean CAPE is always higher at 12 UTC that at 00 UTC; this is not worth mentioning in the abstract.

2. L30: include "…complex topography, **insufficient assimilated observations, and** forecast errors…"

3. L33: delete the words "extreme" as you do not separate among the precipitation intensity.

4. L39: The deep convection…

5. L42: of the atmospheric convection

6. L49: precipitation extreme events convective precipitation

7. L60: most unstable region region with highest instability

8. L67: There is something missing in this sentence (verb and subjective)

9. L86: the mean CAPE

10. L102: extreme events convective situations

11. L115: both wind components **horizontal** wind; geopotential geopotential **height**

12. L125: presents the same prarameterizations relies on the same setup (the parameterizations used are introduced later)

13. L140/163: write out NOAA at first use

14. L167: …12 UTC, 02 and… 12 UTC, corresponding to 02 and 14 LT, respectively)

15. L168: in L167 12 UTC was 14 and not 13 LT

16. L173-179 is somewhat cumbersome (two almost identical sentences)

17. L186 highlighted highlight

18. L187: This… refer to what?

19. L188: in **our** WRF simulations

20. L196 as we would be taking into account as we take into account (you did that, didn't you?)

21. L197-198: This sentence is unclear. "levels are already measured for a drifting distance"? What is the distance of 4.5 km? How is the drifting related to the cloud cover??

22. L200: "This procedure…" What do you refer to?

23. L204: still, it's not minutes rather more than one hour for an entire profile of the atmosphere

24. L205: "…because of wind"; already discussed above.

25. L222: it **was** not it **is** not

26. L233: "Lifted Condensation Level" "**Lifting** C…" is more appropriate (cf. AMS Glossary)

27. L235 trigger cause

28. L237-238: "provide similar information"; I fully disagree with this statement, see the bunch of literature on the various convective indices quantifying conditional, latent, potential instability, or a combination thereof!

29. L262-264 and elsewhere: Spearman / Persons's correlation **coefficient**

30. L288/L306: worsening decrease (avoid personal assessment)

31. L293: are **most** remarkable in Murcia

32. L295-96 and **elsewhere** (e.g., 299): **in** those stations **at** those stations; **at** the Mediterranean coast; you cannot say in a station; this was already explained in my 1st review.

33. L300 **at/for** Barcelona

34. L308: **S**ection 2.3.1; **of** TT index

35. L332 (and elsewhere): trigger cause

36. L346: and Gibraltar) and Gibraltar

37. L347/48 shorter smaller

38. L363-64: "warmer sounding levels" is strange; Reference the reference

39. L362-64: that's of course not a trajectory! lifted air parcel; the sentence "which produces that the lifted trajectory crosses earlier than D the sounding" is very strange

40. L374: "most active ones"?? You mean highest CAPE values?

41. L399: "…system, so the lifting that can trigger convection can appear" "…system that can trigger convection by orographically induced lifting"

42. L406-407: replace the sentence beginning with "…are originated…" "are a consequence of low dew point temperature mainly due to dry air."

43. L412-13: This sentence is a fragment (no Verb and Subjective); again, you cannot say "in the slope"

44. L417/418 and elsewhere: **on** the Atlantic coast; **on** the western coast

45. L419: that  than

46. L438 results **from** TT results **for** TT

47. L440 highly convective events: this is now very confusing. You are investigation mean values, aren't you? Why are you speaking of events? And why "highly"? Did you separate among different intensity classes? I think you did not…

48. L441: "TT and CAPE are indices for atmospheric instability"; No. The one is an index (TT), the other is an integral bulk of convective energy and **not** an index. Besides, CAPE solely estimates latent instability, whereas TT combines conditional (VT), latent, and potential instability.

49. L443-44: "since CAPE and CIN are dependent on the entire profile of the atmosphere.." There are two flaws: you mean the profile below the level of neutral buoyancy and not the whole atmosphere (which is unbounded); and CIN depends only on the layers below the LFC. Furthermore, several studies have shown that – at least in Europe – the Lifted Index has a higher prediction skill than CAPE, even if considers only a parcel lifted to 500 hPa.

50. L446: on the Atlantic coast

51. L449: what do you mean by intensity? When you refer to that, you need to consider CAPE in combination with CIN (as you know, in case of high CIN, instability cannot be released and the intensity is low no matter of CAPE).

52. L451: delete §dynamics"

53. L469: the correlation correlation coefficients according to Pearson and Spearman

54. L470: "…of them" of what?

55. L489: delete "develop"

56. L490: **convective** inhibition

>> We thank the reviewer for highlighting these minor points. They were included in the new version of the manuscript, except some comments:

>> 9: We don't understand what the reviewer is remarking here, as that is what is already stated in the text. In any case, we have modified it to "the mean of CAPE".

>> 15: In Lisbon (Portugal), 12 UTC is 12 LT and 13 LT during winter and summer times respectively. In contrast, for the Spanish stations and Gibraltar, 12 UTC is 13 LT and 14 LT during winter and summer times respectively. The fact that the local times stated in the test were chosen during the summer times has been added to those lines, but the local time for Portugal has not been modified as suggested by the reviewer.

>> 20. As replied to the reviewer in the last revision and as stated in the previous line to that highlighted by the reviewer, we are NOT considering the averaged value of several grid points to be compared against radiosonde data. If we do that, with the spatial resolution of 15 km used in our experiments, we would be considering an area of 2025 km2, which is not suitable to be compared against in situ data. Then, that sentence was not modified in the new version.

>> 23. The main point of that sentence is to show that the vertical profile of the atmosphere is not measured instantaneously (as already stated in that line), and not about the exact amount of time needed for it. In any case, we have modified the sentence again.

>> 31: "are most remarkable" as suggested by the reviewer is not what we mean. Consequently, we have changed it to: "These differences are more remarkable in Murcia." In any case, we think that this kind of editions are adequate for the journal's style editor after acceptance.

---

## Author Response (AR3)

Dear Editor,

Please find enclosed the revised version of the manuscript entitled "Changes in the simulation of atmospheric instability over the Iberian Peninsula due to the use of 3DVAR data assimilation" by S. J. González-Rojí, S. Carreno-Madinabeitia, J. Sáenz and G. Ibarra-Berastegi, that we resubmit to the journal Hydrology and Earth System Sciences.

In this version, all the points raised by the reviewer have been addressed. To do so, we have updated figures 9 and 10 (also Figure 8 for consistency with the new ones), and we have added some new text in the results section. Additionally, we have also corrected some minor inaccuracies included in the manuscript that we submitted last time. These corrections do not modify the final outcomes from the paper. Apart from that, with the aim of reducing the length of the main manuscript, we decided to move the figures included in the annexes to the Supplementary Materials online. Finally, we have also modified the affiliations of most of the authors in order to be consistent with the new names given recently to the Departments by the University of the Basque Country (UPV/EHU).

As always, attached to this cover letter you will find the new version of the manuscript, the version with the changes tracked where all the modifications are highlighted, and a detailed response to the reviewer's points. We consider that we have successfully addressed all the points raised by the reviewer and, as such, we hope that the manuscript can be accepted this time.

Yours faithfully,
Santos J. González-Rojí

**Reply to the review made by the Anonymous Referee #2**
**Received on: 30 April 2021**

The authors have done a great job. I think that the paper has improved considerably. Therefore, I suggest to accept the paper, but to please consider the few points below.
>> Thanks for taking into consideration all the effort that we made to improve the paper according to your comments.

1. L107: "precipitation can be developed" --> "precipitation may occur"
>> We have corrected it as suggested.

2. Sentence L107-109: There's something missing in this sentence (verb?)
>> The reviewer is right and something was missing in the sentence. We have rephrased it to:
"As shown before, atmospheric instability is a highly demanding feature in model simulations and a topic with great importance nowadays due to the large damage that extreme convective events can cause to society, and which frequency will be increased in the future. Thus, it is of great interest to diagnose the ability of particular configurations of a model to properly simulate the structure of temperature and moisture at low levels, which lead to unstable atmospheric instability"

3. L224: a extremely --> an extremely
>> We have corrected that typo.

4. L552: in the pressure levels --> at the pressure levels
>> We changed it as suggested in the new version of the manuscript.

5. 574-576: Be careful with that statement about CAPE/CIN as their calculation is highly sensitive to the low-level values even though when considering mixed values. This is not the case for TT. Or what do you mean by "reliable"?
>> We meant with "reliable" that because of employing more pressure levels during its calculation, the information provided by CAPE and CIN should be more physically meaningful than the information obtained by TT, as this index is only calculated from two pressure levels (500 and 850 hPa). In order to make this clear, we have changed the sentence to: "Thus, since the calculation of CAPE and CIN takes into consideration the vertical profile of the atmosphere until the Level of Neutral Buoyancy, these two variables should be considered first to evaluate atmospheric conditions and not TT, as this index only takes into account two pressure levels (500 and 850 hPa)."

6. My point 8: This is a general misunderstanding. You say that your main objective is to evaluate the WRF results during unstable conditions. Absolutely understood. But CIN against to CAPE and TT is not a measure of instability. For example, during a very stable situation, CAPE is = 0, but CIN could be in the convective range of ~50-200 J/kg K. So if you consider CIN on all days, your focus is not on convective conditions as stated several times. That's the point.

With the relation between CAPE and CIN in my comment I didn't refer to their correlation, but to consider CIN only during unstable conditions, i.e. when CAPE is > 0 J / kg K.

>> We see the point made by the reviewer now. We didn't understand it the previous time. We do not dispute (never did) that evaluating CIN only during unstable conditions is interesting for weather forecasting. However, we believe that if we want to evaluate the performance of the model at reproducing the observed atmospheric conditions with TT, CAPE and CIN, we need to evaluate the entire period (2010-2014) and not restrict the evaluation to only the days when CAPE is above a given threshold.

>> However, it is true that when we comment on the maps that show the regions with larger values of TT, CAPE and CIN, we talk about the most unstable regions over the IP, and to do so, we say that we need to evaluate CAPE and CIN in combination. So, in order to ease the interpretation of that for the reader and to answer this point by reviewer, we have calculated the mean maps for CAPE and CIN only for the days in which CAPE is above the 75th percentile (calculated for each point, season and time - 00 or 12 UTC). We decided not to apply a fixed threshold such as 0 or 50 J/Kg as suggested by the reviewer in previous comments to each point of the domain since each of them presents a different behaviour, and consequently, the number of data computed in the mean would not be the same in each point depending on that threshold. By using the 75th percentile, the amount of data employed for the calculation of the mean at each point would be the same, and the evaluation will be fair for all the points of the domain. Besides that, the third quartile will always be representative of conditions at the unstable range of the distribution for every grid point.

>> The results show that the spatial distribution of CAPE is rather similar to that obtained for the mean of the entire period, and only the values are larger (Figure 1 in this response). However, some differences can be observed for CIN, particularly at night during both winter and summer (Figure 2 in this response). During winter, it can be seen that the highest values of CIN appear at night at the Mediterranean coast. This area is usually where nocturnal radiation cooling is large because of a lower cloud cover, although this result is highly dependent on the dataset used for the analysis (Calbo and Sanchez-Lorenzo, 2009), and consequently, it should be further analysed for our simulations in the future. During summer, large areas of the southern IP are affected by the stabilizing effect due to radiational surface cooling during the frequent clear nights beyond the Mediterranean regions which were also apparent in the average summer night (00 UTC) map. In addition,

the same differences between both WRF experiments are observed in these new maps as in the case of the mean maps for the entire period 2010-2014. Consequently, N shows larger CIN values than D during winter, but this is reversed in summer when larger values are shown by D.

>> In order to include in the manuscript the most relevant results, we have decided to include only the maps for CAPE and CIN above the 75th percentile of CAPE for the D experiment in a new column in Figures 9 and 10 (CAPE and CIN maps respectively in the manuscript). Moreover, we have included some new text in section 2.3.2 (Analysis) explaining how these maps were calculated, in the results section explaining and interpreting these new maps, and finally also in the conclusions. These new paragraphs are highlighted as changed in the uploaded version with changes marked.
* * *
Calbó, J. and Sanchez-Lorenzo, A.: Cloudiness climatology in the Iberian Peninsula from three global gridded datasets (ISCCP, CRU TS 2.1, ERA-40), Theoretical and Applied Climatology, 96, 105–115, https://doi.org/10.1007/s00704-008-0039-z, 2009.

[Figure]

Figure 1: Spatial distribution of mean CAPE for period 2010-2014 over the IP as computed from N (first column) and D (second column) for those days in which CAPE is higher to the third quartile of the sample of CAPE at every grid point. The maps for winter (rows 1 and 2) and summer (rows 3 and 4) at 00 (rows 1 and 3) and 12 UTC (rows 2 and 4) are shown. The median value (J/kg) of each map is presented in the bottom right corner of the plots.

[Figure]

Figure 2: Same ad Figure 1, but for CIN in those days in which CAPE is higher to the third quartile of the sample of CAPE at every grid point.